# On the Double Descent of Random Features Models Trained with SGD

**Fanghui Liu**[*]
LIONS, EPFL
fanghui.liu@epfl.ch

**Johan A.K. Suykens**
ESAT-STADIUS, KU Leuven
johan.suykens@esat.kuleuven.be

**Volkan Cevher**
LIONS, EPFL
volkan.cevher@epfl.ch

## Abstract

We study generalization properties of random features (RF) regression in high dimensions optimized by stochastic gradient descent (SGD) in under-/over-parameterized regime. In this work, we derive precise non-asymptotic error bounds of RF regression under both constant and polynomial-decay step-size SGD setting, and observe the double descent phenomenon both theoretically and empirically. Our analysis shows how to cope with multiple randomness sources of initialization, label noise, and data sampling (as well as stochastic gradients) with no closed-form solution, and also goes beyond the commonly-used Gaussian/spherical data assumption. Our theoretical results demonstrate that, with SGD training, RF regression still generalizes well for interpolation learning, and is able to characterize the double descent behavior by the unimodality of variance and monotonic decrease of bias. Besides, we also prove that the constant step-size SGD setting incurs no loss in convergence rate when compared to the exact minimum-norm interpolator, as a theoretical justification of using SGD in practice.

## 1 Introduction

Over-parameterized models, e.g., linear/kernel regression [1, 2, 3, 4] and neural networks [5, 6, 7], still generalize well even if the labels are pure noise [8]. Such high-capacity models have received significant attention recently as they go against with classical generalization theory. A paradigm for understanding this important phenomenon is *double descent* [9], in which the test error first decreases with increasing number of model parameters in the under-parameterized regime. They large error is yielded until interpolating the data, which is called the interpolation threshold. Finally, the test error decreases again in the over-parameterized regime.

Our work partakes in this research vein and studies the random features (RF) model [10], as a simplified version of neural networks, in the context of double descent phenomenon. Briefly, RF model samples random features $\{\omega_i\}_{i=1}^m$ from a specific distribution, corresponding to a kernel function. We then construct an explicit map: $x \in \mathbb{R}^d \mapsto \sigma(Wx) \in \mathbb{R}^m$, where $W = [\omega_1, \cdots, \omega_m]^\top \in \mathbb{R}^{m \times d}$ is the random features matrix and $\sigma(\cdot)$ is the nonlinear (activation) function determined by the kernel. As a result, the RF model can be viewed as training a two-layer neural network where the weights in the first layer are chosen randomly and then fixed (a.k.a. the random features) and only the output layer is optimized, striking a trade-off between practical performance and accessibility to analysis

---

[*]Most of this work was done when Fanghui was at KU Leuven. Correspondence to: Fanghui Liu <fanghui.liu@epfl.ch>.

36th Conference on Neural Information Processing Systems (NeurIPS 2022).

[4, 11]. An RF model becomes an over-parameterized model if we take the number of random features $m$ larger than that of training data $n$. The literature on RF under the over-parameterized regime can be split into various camps according to different assumptions on the formulation of target function, data distribution, and activation functions [4, 12, 11, 13, 14, 15] (see comparisons in Table 1 in Appendix A). The existing theoretical results demonstrate that the excess risk curve exhibits double descent.

Nevertheless, the analysis framework of previous work on RF regression mainly relies on the least-squares closed-form solution, including *minimum-norm* interpolator and ridge regressor. Besides, they often assume the data with specific distribution, e.g., to be Gaussian or uniformly spread on a sphere. Such dependency on the analytic solution and relatively strong data assumption in fact mismatches practical neural networks optimized by stochastic gradient descent (SGD) based algorithms. Our work precisely bridges this gap: We provide a new analysis framework for the generalization properties of RF models trained with SGD and general activation functions, also accommodating adaptive (i.e., polynomial decay) step-size selection, and provide non-asymptotic results in under-/over-parameterized regimes. We make the following contributions and findings:

First, we characterize statistical properties of covariance operators/matrices in RF, including $\Sigma_m := \frac{1}{m}\mathbb{E}_{\boldsymbol{x}}[\sigma(\boldsymbol{W}\boldsymbol{x}/\sqrt{d})\sigma(\boldsymbol{W}\boldsymbol{x}/\sqrt{d})^\top]$ and its expectation version $\widetilde{\Sigma}_m := \mathbb{E}_{\boldsymbol{W}}[\Sigma_m]$. We demonstrate that, under Gaussian initialization, if the activation function $\sigma(\cdot) : \mathbb{R} \mapsto \mathbb{R}$ is Lipschitz continuous, $\mathrm{Tr}(\Sigma_m)$ is a sub-exponential random variable with $\mathcal{O}(1)$ sub-exponential norm; $\widetilde{\Sigma}_m$ has only two distinct eigenvalues at $\mathcal{O}(1)$ and $\mathcal{O}(1/m)$ order, respectively. Such analysis on the spectra of $\Sigma_m$ and $\widetilde{\Sigma}_m$ (without spectral decay assumption) is helpful to obtain sharp error bounds for excess risk. This is different from the least squares setting based on effective dimension [2, 16].

Second, based on the bias-variance decomposition in stochastic approximation, we take into account multiple randomness sources of initialization, label noise, and data sampling as well as stochastic gradients. We (partly) disentangle these randomness sources and derive non-asymptotic error bounds under the optimization effect: the error bounds for bias and variance as a function of the radio $m/n$ are monotonic decreasing and unimodal, respectively. Importantly, our analysis holds for both constant and polynomial-decay step-size SGD setting, and is valid under sub-Gaussian data and general activation functions.

Third, our non-asymptotic results show that, RF regression trained with SGD still generalizes well for interpolation learning, and is able to capture the double descent behavior. In addition, we demonstrate that the constant step-size SGD setting incurs no loss on the convergence rate of excess risk when compared to the exact least-squares closed form solution. Our empirical evaluations support our theoretical results and findings.

Our analysis (technical challenges are discussed in Section 4) sheds light on the effect of SGD on high dimensional RF models in under-/over-parameterized regimes, and bridges the gap between the minimum-norm solution and numerical iteration solution in terms of optimization and generalization on double descent. It would be helpful for understanding large dimensional machine learning and neural network models more generally.

## 2  Related work and problem setting

This section reviews relevant works and introduces our problem setting of RF regression with SGD.

**Notation:** The notation $\boldsymbol{a} \otimes \boldsymbol{a}$ denotes the tensor product of a vector $\boldsymbol{a}$. For two operators/matrices, $A \preccurlyeq B$ means $B - A$ is positive semi-definite (PSD). For any two positive sequences $\{a_t\}_{t=1}^s$ and $\{b_t\}_{t=1}^s$, the notation $a_t \lesssim b_t$ means that there exists a positive constant $C$ independent of $s$ such that $a_t \leq Cb_t$, and analogously for $\sim, \gtrsim$, and $\precsim$. For any $a, b \in \mathbb{R}$, $a \wedge b$ denotes the minimum of $a$ and $b$.

### 2.1  Related work

A flurry of research papers are devoted to analysis of over-parameterized models on optimization [17, 18, 19], generalization (or their combination) under neural tangent kernel [20, 21, 22] and mean-field analysis regime [23, 24]. We take a unified perspective on optimization and generalization but work in the high-dimensional setting to fully capture the double descent behavior. By high-dimensional setting, we mean that $m$, $n$, and $d$ increase proportionally, large and comparable [4, 12, 13, 11].

**Double descent in random features model:** Characterizing the double descent of the RF model often derives from random matrix theory (RMT) in high dimensional statistics [1, 4, 12, 13, 25] and from the replica method [11, 26, 14]. Under specific assumptions on data distribution, activation functions, target function, and initialization, these results show that the generalization error/excess risk increase when $m/n < 1$, diverge when $m/n \to 1$, and then decrease when $m/n > 1$. Further, refined results are developed on the *analysis of variance* due to multiple randomness sources [11, 27, 15]. We refer to comparisons in Table 1 in Appendix A for further details. Technically speaking, since RF (least-squares) regression involves with inverse random matrices, these two classes of methods attempt to achieve a similar target: how to disentangle the nonlinear activation function by the Gaussian equivalence conjecture. RMT utilizes calculus of deterministic equivalents (or resolvents) for random matrices and replica methods focus on some specific scalar parameters that allows for circumventing the expectation computation. In fact, most of the above methods can be asymptotically equivalent to the Gaussian covariate model [28].

**Non-asymptotic stochastic approximation:** Many papers on linear least-squares regression [29, 30], kernel regression [31, 32], random features [33] with SGD often work in the under-parameterized regime, where $d$ is finite and much smaller than $n$. In the over-parameterized regime, under GD setting, the excess risk of least squares is controlled by the smallest positive eigenvalue in [34] via random matrix theory. Under the averaged constant step-size SGD setting, the excess risk in [35] on least squares in high dimensions can be independent of $d$, and the convergence rate is built in [16]. This convergence rate is also demonstrated under the minimal-iterate [36] or last-iterate [37] setting in step-size SGD for noiseless least squares. We also notice a concurrent work [38] on last-iterate SGD with decaying step-size on least squares. Besides, the existence of multiple descent [39, 40] beyond double descent and SGD as implicit regularizer [41, 42] can be traced to the above two lines of work. Our work shares some similar technical tools with [31] and [16] but differs from them in several aspects. We detail the differences in Section 4.

## 2.2 Problem setting

We study the standard problem setting for RF least-squares regression and adopt the relevant terminologies from learning theory: *cf.*, [43, 31, 33, 25] for details. Let $X \subseteq \mathbb{R}^d$ be a metric space and $Y \subseteq \mathbb{R}$. The training data $\{(\boldsymbol{x}_i, y_i)\}_{i=1}^n$ are assumed to be independently drawn from a non-degenerate unknown Borel probability measure $\rho$ on $X \times Y$. The *target function* of $\rho$ is defined by $f_\rho(\boldsymbol{x}) = \int_Y y \, d\rho(y \mid \boldsymbol{x})$, where $\rho(\cdot \mid \boldsymbol{x})$ is the conditional distribution of $\rho$ at $\boldsymbol{x} \in X$.

**RF least squares regression:** We study the RF regression problem with the squared loss as follows:

$$\min_{f \in \mathcal{H}} \mathcal{E}(f), \quad \mathcal{E}(f) := \int (f(\boldsymbol{x}) - y)^2 d\rho(\boldsymbol{x}, y) = \|f - f_\rho\|_{L^2_{\rho_X}}^2, \quad \text{with } f(\boldsymbol{x}) = \langle \boldsymbol{\theta}, \varphi(\boldsymbol{x}) \rangle,$$

where the optimization vector $\boldsymbol{\theta} \in \mathbb{R}^m$ and the feature mapping $\varphi(\boldsymbol{x})$ is defined as

$$\varphi(\boldsymbol{x}) := \frac{1}{\sqrt{m}} \left[ \sigma(\boldsymbol{\omega}_1^\top \boldsymbol{x}/\sqrt{d}), \cdots, \sigma(\boldsymbol{\omega}_m^\top \boldsymbol{x}/\sqrt{d}) \right]^\top = \frac{1}{\sqrt{m}} \sigma(\boldsymbol{W} \boldsymbol{x}/\sqrt{d}) \in \mathbb{R}^m, \quad (1)$$

where $\boldsymbol{W} = [\boldsymbol{\omega}_1, \cdots, \boldsymbol{\omega}_m]^\top \in \mathbb{R}^{m \times d}$ with $W_{ij} \sim \mathcal{N}(0, 1)$ corresponds to such two-layer neural network initialized with random Gaussian weights. Then, the corresponding hypothesis space $\mathcal{H}$ is a reproducing kernel Hilbert space

$$\mathcal{H} := \left\{ f \in L^2_{\rho_X} \,\middle|\, f(\boldsymbol{x}) = \frac{1}{\sqrt{m}} \langle \boldsymbol{\theta}, \sigma(\boldsymbol{W} \boldsymbol{x}/\sqrt{d}) \rangle \right\}, \quad (2)$$

with $\|f\|_{L^2_{\rho_X}}^2 = \int_X |f(\boldsymbol{x})|^2 d\rho_X(\boldsymbol{x}) = \langle f, \Sigma_m f \rangle_{\mathcal{H}}$ with the *covariance* operator $\Sigma_m : \mathbb{R}^m \to \mathbb{R}^m$

$$\Sigma_m = \int_X \varphi(\boldsymbol{x}) \otimes \varphi(\boldsymbol{x}) d\rho_X(\boldsymbol{x}), \quad (3)$$

actually defined in $\mathcal{H}$ that is isomorphic to $\mathbb{R}^m$. This is the usually (uncentered) covariance matrix in finite dimensions,[2] i.e., $\boldsymbol{\Sigma}_m = \mathbb{E}_{\boldsymbol{x}}[\varphi(\boldsymbol{x}) \otimes \varphi(\boldsymbol{x})]$. Define $J_m : \mathbb{R}^m \to L^2_{\rho_X}$ such that $(J_m \boldsymbol{v})(\cdot) = \langle \boldsymbol{v}, \varphi(\cdot) \rangle$, $\forall \boldsymbol{v} \in \mathbb{R}^m$, we have $\Sigma_m = J_m^* J_m$, where $J_m^*$ denotes the adjoint operator of $J_m$.

---

[2]In this paper, we do not distinguish the notations $\Sigma_m$ and $\boldsymbol{\Sigma}_m$. This is also suitable to other operators/matrices, e.g., $\widetilde{\Sigma}_m$.

Clearly, $\Sigma_m$ is random with respect to $\boldsymbol{W}$, and thus its deterministic version is defined as $\widetilde{\Sigma}_m = \mathbb{E}_{\boldsymbol{x},\boldsymbol{W}}[\varphi(\boldsymbol{x}) \otimes \varphi(\boldsymbol{x})]$.

**SGD with averaging:** Regarding the stochastic approximation, we consider the one pass SGD with iterate averaging and adaptive step-size at each iteration $t$: after a training sample $(\boldsymbol{x}_t, y_t) \sim \rho$ is observed, we update the decision variable as below (initialized at $\boldsymbol{\theta}_0$)

$$\boldsymbol{\theta}_t = \boldsymbol{\theta}_{t-1} + \gamma_t[y_t - \langle \boldsymbol{\theta}_{t-1}, \varphi(\boldsymbol{x}_t)\rangle]\varphi(\boldsymbol{x}_t), \qquad t = 1, 2, \ldots, n\,, \qquad (4)$$

where we use the polynomial decay step size $\gamma_t := \gamma_0 t^{-\zeta}$ with $\zeta \in [0, 1)$, following [31]. This setting also holds for the constant step-size case by taking $\zeta = 0$. Besides, we employ the bath size $= 1$ in an online setting style, which is commonly used in theory [31, 16, 44] for ease of analysis, which captures the key idea of SGD by combining stochastic gradients and data sampling.

The final output is defined as the average of the iterates: $\bar{\boldsymbol{\theta}}_n := \frac{1}{n}\sum_{t=0}^{n-1} \boldsymbol{\theta}_t$. Here we sum up $\{\theta_t\}_{t=0}^{n-1}$ with $n$ terms for notational simplicity. The optimality condition for Eq. (4) implies $\mathbb{E}_{(\boldsymbol{x},y)\sim\rho}[(y - \langle \boldsymbol{\theta}^*, \varphi(\boldsymbol{x})\rangle)\varphi(\boldsymbol{x})] = \boldsymbol{0}$, which corresponds to $f^* = J_m\boldsymbol{\theta}^*$ if we assume that $f^* = \operatorname{argmin}_{f\in\mathcal{H}} \mathcal{E}(f)$ exists (see Assumption 2 in the next section). Likewise, we have $f_t = J_m\boldsymbol{\theta}_t$ and $\bar{f}_n = J_m\bar{\boldsymbol{\theta}}_n$.

In this paper, we study the averaged excess risk $\mathbb{E}\|\bar{f}_n - f^*\|_{L^2_{\rho_X}}^2$ instead of $\mathbb{E}\|\bar{f}_n - f_\rho\|_{L^2_{\rho_X}}^2$, that follows [31, 45, 33, 25], as $f^*$ is the best possible solution in $\mathcal{H}$ and the mis-specification error $\|f^* - f_\rho\|_{L^2_{\rho_X}}^2$ pales into insignificance. Note that the expectation used here is considered with respect to the random features matrix $\boldsymbol{W}$, and the distribution of the training data $\{(\boldsymbol{x}_t, y_t)\}_{t=1}^n$ (note that $\|\bar{f}_n - f^*\|_{L^2_{\rho_X}}^2$ is itself a different expectation over $\rho_X$).

## 3 Main results

In this section, we present our main theoretical results on the generalization properties employing error bounds for bias and variance of RF regression in high dimensions optimized by averaged SGD.

### 3.1 Assumptions

Before we present our result, we list the assumptions used in this paper, refer to Appendix B for more discussions.

**Assumption 1.** *[46, 1, high dimensional setting] We work in the large $d, n, m$ regime with $c \leqslant \{d/n, m/n\} \leqslant C$ for some constants $c, C > 0$ such that $m, n, d$ are large and comparable. The data point $\boldsymbol{x} \in \mathbb{R}^d$ is assumed to satisfy $\|\boldsymbol{x}\|_2^2 \sim \mathcal{O}(d)$ and the sample covariance operator $\Sigma_d := \mathbb{E}_{\boldsymbol{x}}[\boldsymbol{x} \otimes \boldsymbol{x}]$ with bounded spectral norm $\|\Sigma_d\|_2$ (finite and independent of $d$).*

**Assumption 2.** *There exists $f^* \in \mathcal{H}$ such that $f^* = \operatorname{argmin}_{f\in\mathcal{H}} \mathcal{E}(f)$ with bounded Hilbert norm.*

**Remark:** This bounded Hilbert norm assumption is commonly used in [47, 40, 48] even though $n$ and $d$ tend to infinity. It holds true for linear functions with $\|f\|_{\mathcal{H}} \leqslant 4\pi$ [49], see Appendix B for details.

**Assumption 3.** *The activation function $\sigma(\cdot)$ is assumed to be Lipschitz continuous.*

**Remark:** This assumption is quite general to cover commonly-used activation functions used in random features and neural networks, e.g., ReLU, Sigmoid, Logistic, and sine/cosine functions.

Recall $\Sigma_m := \mathbb{E}_{\boldsymbol{x}}[\varphi(\boldsymbol{x}) \otimes \varphi(\boldsymbol{x})]$ in Eq. (3) and its expectation $\widetilde{\Sigma}_m := \mathbb{E}_{\boldsymbol{W}}[\Sigma_m]$, we make the following fourth moment assumption that follows [29, 16, 37] to analyse SGD for least squares.

**Assumption 4** (Fourth moment condition). *Assume there exists some positive constants $r', r \geqslant 1$, such that for any PSD operator $A$, it holds that*

$$\mathbb{E}_{\boldsymbol{W}}[\Sigma_m A \Sigma_m] \preccurlyeq \mathbb{E}_{\boldsymbol{W}}\left(\mathbb{E}_{\boldsymbol{x}}\left([\varphi(\boldsymbol{x}) \otimes \varphi(\boldsymbol{x})]A[\varphi(\boldsymbol{x}) \otimes \varphi(\boldsymbol{x})]\right)\right) \preccurlyeq r'\mathbb{E}_{\boldsymbol{W}}[\operatorname{Tr}(\Sigma_m A)\Sigma_m] \preccurlyeq r\operatorname{Tr}(\widetilde{\Sigma}_m A)\widetilde{\Sigma}_m.$$

**Remark:** This assumption requires the data are drawn from some not-too-heavy-tailed distribution, e.g., $\Sigma_m^{-\frac{1}{2}}\boldsymbol{x}$ has sub-Gaussian tail, common in high dimensional statistics. This condition is weaker than most previous work on double descent that requires the data to be Gaussian [1, 11, 27, 12], or uniformly spread on a sphere [4, 50], see comparisons in Table 1 in Appendix A. Note that the

assumption for any PSD operator is just for ease of description. In fact some certain PSD operators satisfying this assumption are enough for our proof. Besides, a special case of this assumption with $A := I$ is proved by Lemma 3, and thus this assumption can be regarded as a natural extension, with more discussions in Appendix B.

**Assumption 5** (Noise condition). *There exists $\tau > 0$ such that $\Xi := \mathbb{E}_{\boldsymbol{x}}[\varepsilon^2 \varphi(\boldsymbol{x}) \otimes \varphi(\boldsymbol{x})] \preccurlyeq \tau^2 \Sigma_m$, where the noise $\varepsilon := y - f^*(\boldsymbol{x})$.*

**Remark:** This noise assumption is standard in [31, 16] and holds for the standard noise model $y = f^*(\boldsymbol{x}) + \varepsilon$ with $\mathbb{E}[\varepsilon] = 0$ and $\mathbb{V}[\varepsilon] < \infty$ [1].

### 3.2 Properties of covariance operators

Before we present the main results, we study statistical properties of $\Sigma_m$ and $\widetilde{\Sigma}_m$ by the following lemmas (with proof deferred to Appendix C), that will be needed for our main result. This is different from the least squares setting [2, 16] that introduces the effective dimension to separate the entire space into a "head" subspace where the error decays more quickly than the complement "tail" subspace. Instead, the following lemma shows that $\widetilde{\Sigma}_m$ has only two distinct eigenvalues at $\mathcal{O}(1)$ and $\mathcal{O}(1/m)$ order, respectively. Such fast eigenvalue decay can avoid extra data spectrum assumption for tight bound. For description simplicity, we consider the single-output activation function: $\sigma(\cdot) : \mathbb{R} \to \mathbb{R}$. Our results can be extended to multiple-output activation functions, see Appendix C.1.2 for details.

**Lemma 1.** *Under Assumption 1 and 3, the expected covariance operator $\widetilde{\Sigma}_m := \mathbb{E}_{\boldsymbol{x}, \boldsymbol{W}}[\varphi(\boldsymbol{x}) \otimes \varphi(\boldsymbol{x})] \in \mathbb{R}^{m \times m}$ has the same diagonal elements and the same non-diagonal element*

$$(\widetilde{\Sigma}_m)_{ii} = \frac{1}{m} \mathbb{E}_{\boldsymbol{x}} \mathbb{E}_{z \sim \mathcal{N}(0, \|\boldsymbol{x}\|_2^2/d)}[\sigma(z)]^2 \sim \mathcal{O}(1/m), \quad (\widetilde{\Sigma}_m)_{ij} = \frac{1}{m} \mathbb{E}_{\boldsymbol{x}} \Big( \mathbb{E}_{z \sim \mathcal{N}(0, \|\boldsymbol{x}\|_2^2/d)}[\sigma(z)] \Big)^2 \sim \mathcal{O}(1/m).$$

*Accordingly, $\widetilde{\Sigma}_m$ has only two distinct eigenvalues*

$$\widetilde{\lambda}_1 = (\widetilde{\Sigma}_m)_{ii} + (m-1)(\widetilde{\Sigma}_m)_{ij} \sim \mathcal{O}(1), \quad \widetilde{\lambda}_2 = (\widetilde{\Sigma}_m)_{ii} - (\widetilde{\Sigma}_m)_{ij} = \frac{1}{m} \mathbb{E}_{\boldsymbol{x}} \mathbb{V}[\sigma(z)] \sim \mathcal{O}(1/m).$$

**Remark:** Lemma 1 implies $\text{tr}(\widetilde{\Sigma}_m) < \infty$. In fact, $\mathbb{E}_{\boldsymbol{x}} \mathbb{V}[\sigma(z)] > 0$ holds almost surely as $\sigma(\cdot)$ is not a constant, and thus $\widetilde{\Sigma}_m$ is positive definite.

Here we take the ReLU activation $\sigma(x) = \max\{x, 0\}$ as one example, RF actually approximates the first-order arc-cosine kernel [51] with $\varphi(\boldsymbol{x}) \in \mathbb{R}^m$. We have $(\widetilde{\Sigma}_m)_{ii} = \frac{1}{2md} \text{Tr}(\Sigma_d)$ and $(\widetilde{\Sigma}_m)_{ij} = \frac{1}{2md\pi} \text{Tr}(\Sigma_d)$ by recalling $\Sigma_d := \mathbb{E}_{\boldsymbol{x}}[\boldsymbol{x}\boldsymbol{x}^\top]$ and $\text{Tr}(\Sigma_d)/d \sim \mathcal{O}(1)$. More examples can be found in Appendix C.1.2.

**Lemma 2.** *Under Assumptions 1 and 3, random variables $\|\Sigma_m\|_2$, $\|\Sigma_m - \widetilde{\Sigma}_m\|_2$, and $\text{Tr}(\Sigma_m)$ are sub-exponential, and have sub-exponential norm at $\mathcal{O}(1)$ order.*

**Remark:** This lemma characterizes the sub-exponential property of covariance operator $\Sigma_m$, which is a fundamental result for our proof since the bias and variance involve them.

The following lemma demonstrates that the behavior of the fourth moment can be bounded.

**Lemma 3.** *Under Assumptions 1, and 3, there exists a constant $r > 0$ such that $\mathbb{E}_{\boldsymbol{W}}(\Sigma_m^2) \preccurlyeq \mathbb{E}_{\boldsymbol{x}, \boldsymbol{W}}[\varphi(\boldsymbol{x}) \otimes \varphi(\boldsymbol{x}) \otimes \varphi(\boldsymbol{x}) \otimes \varphi(\boldsymbol{x})] \preccurlyeq r\text{Tr}(\widetilde{\Sigma}_m)\widetilde{\Sigma}_m$.*

**Lemma 4.** *Under Assumptions 1 and 3, we have $\text{Tr}[\widetilde{\Sigma}_m^{-1} \mathbb{E}_{\boldsymbol{W}}(\Sigma_m^2)] \sim \mathcal{O}(1)$.*

We remark here that Lemma 3 is a special case of Assumption 4 if we take $A := I$ and $r := 1 + \mathcal{O}\left(\frac{1}{m}\right)$; and Lemma 4 is a direct corollary of Lemma 3.

### 3.3 Results for error bounds

Recall the definition of the noise $\boldsymbol{\varepsilon} = [\varepsilon_1, \cdots, \varepsilon_n]^\top$ with $\varepsilon_t = y_t - f^*(\boldsymbol{x}_t)$, $t = 1, 2, \ldots, n$, the averaged excess risk can be expressed as

$$\mathbb{E}\|\bar{f}_n - f^*\|_{L^2_{\rho_X}}^2 := \mathbb{E}_{\boldsymbol{X}, \boldsymbol{W}, \boldsymbol{\varepsilon}}\|\bar{f}_n - f^*\|_{L^2_{\rho_X}}^2 = \mathbb{E}_{\boldsymbol{X}, \boldsymbol{W}, \boldsymbol{\varepsilon}}\langle\bar{f}_n - f^*, \Sigma_m(\bar{f}_n - f^*)\rangle = \mathbb{E}_{\boldsymbol{X}, \boldsymbol{W}, \boldsymbol{\varepsilon}}\langle\bar{\eta}_n, \Sigma_m\bar{\eta}_n\rangle,$$

where $\bar{\eta}_n := \frac{1}{n}\sum_{t=0}^{n-1}\eta_t$ with the centered SGD iterate $\eta_t := f_t - f^*$. Following the standard bias-variance decomposition in stochastic approximation [31, 30, 16], it admits

$$\eta_t = [I - \gamma_t\varphi(\boldsymbol{x}_t)\otimes\varphi(\boldsymbol{x}_t)](f_{t-1} - f^*) + \gamma_t\varepsilon_t\varphi(\boldsymbol{x}_t),$$

where the first term corresponds to the bias

$$\eta_t^{\texttt{bias}} = [I - \gamma_t\varphi(\boldsymbol{x}_t)\otimes\varphi(\boldsymbol{x}_t)]\eta_{t-1}^{\texttt{bias}}, \quad \eta_0^{\texttt{bias}} = f^*, \tag{5}$$

and the second term corresponds to the variance

$$\eta_t^{\texttt{var}} = [I - \gamma_t\varphi(\boldsymbol{x}_t)\otimes\varphi(\boldsymbol{x}_t)]\eta_{t-1}^{\texttt{var}} + \gamma_t\varepsilon_t\varphi(\boldsymbol{x}_t), \quad \eta_0^{\texttt{var}} = 0. \tag{6}$$

Accordingly, we have $f_t = \eta_t^{\texttt{bias}} + \eta_t^{\texttt{var}} + f^*$ due to $\mathbb{E}_{\boldsymbol{\varepsilon}}\bar{f}_n = \bar{\eta}_n^{\texttt{bias}} + f^*$ and $\|f\|_{L^2_{\rho_X}}^2 = \langle f, \Sigma_m f\rangle$.

**Proposition 1.** *Based on the above setting, the averaged excess risk admits the following bias-variance decomposition*

$$\mathbb{E}\|\bar{f}_n - f^*\|_{L^2_{\rho_X}}^2 = \mathbb{E}_{\boldsymbol{X},\boldsymbol{W},\boldsymbol{\varepsilon}}\|\bar{f}_n - \mathbb{E}_{\boldsymbol{\varepsilon}}\bar{f}_n + \mathbb{E}_{\boldsymbol{\varepsilon}}\bar{f}_n - f^*\|_{L^2_{\rho_X}}^2 = \underbrace{\mathbb{E}_{\boldsymbol{X},\boldsymbol{W}}\langle\bar{\eta}_n^{\texttt{bias}}, \Sigma_m\bar{\eta}_n^{\texttt{bias}}\rangle}_{:=\texttt{Bias}} + \underbrace{\mathbb{E}_{\boldsymbol{X},\boldsymbol{W},\boldsymbol{\varepsilon}}\langle\bar{\eta}_n^{\texttt{var}}, \Sigma_m\bar{\eta}_n^{\texttt{var}}\rangle}_{:=\texttt{Variance}}.$$

By (partly) decoupling the multiple randomness sources of initialization, label noise, and data sampling (as well as stochastic gradients), we give precise non-asymptotic error bounds for bias and variance as below.

**Theorem 1.** *(Error bound for bias) Under Assumptions 1, 2, 3, 4 with $r' \geqslant 1$, if the step-size $\gamma_t := \gamma_0 t^{-\zeta}$ with $\zeta \in [0, 1)$ satisfies $\gamma_0 \lesssim \frac{1}{r'\mathrm{Tr}(\widetilde{\Sigma}_m)} \sim \mathcal{O}(1)$, the* `Bias` *in Proposition 1 holds by*

$$\texttt{Bias} \lesssim \gamma_0 r' n^{\zeta-1}\|f^*\|^2 \sim \mathcal{O}\left(n^{\zeta-1}\right).$$

**Remark:** The error bound for `Bias` is monotonically decreasing at $\mathcal{O}(n^{\zeta-1})$ rate. For the constant step-size setting, it converges at $\mathcal{O}(1/n)$ rate, which is better than $\mathcal{O}(\sqrt{\log n/n})$ in [25] relying on closed-form solution under correlated features with polynomial decay on $\Sigma_d$. Besides, our result on bias matches the exact formulation in [11] under the closed-form solution, i.e., monotonically decreasing bias. One slight difference is, their result on bias tends to a constant under the over-parameterized regime while our bias result can converge to zero.

**Theorem 2.** *(Error bound for variance) Under Assumptions 1, 3, 4 with $r' \geqslant 1$, and Assumption 5 with $\tau > 0$, if the step-size $\gamma_t := \gamma_0 t^{-\zeta}$ with $\zeta \in [0, 1)$ satisfies $\gamma_0 \lesssim \frac{1}{r'\mathrm{Tr}(\widetilde{\Sigma}_m)} \sim \mathcal{O}(1)$, the* `Variance` *defined in Proposition 1 holds*

$$\texttt{Variance} \lesssim \gamma_0 r'\tau^2 \begin{cases} mn^{\zeta-1}, & \text{if } m \leqslant n \\ 1 + n^{\zeta-1} + \dfrac{n}{m}, & \text{if } m > n \end{cases}$$

**Remark:** We make the following remarks:
*i)* The error bound for `Variance` is demonstrated to be unimodal: increasing with $m$ in the under-parameterized regime and decreasing with $m$ in the over-parameterized regime, and finally converge to a constant order (that depends on noise parameter $\tau^2$), which matches recent results relying on closed-form solution for (refined) variance, e.g., [11, 27, 15].
*ii)* When compared to least squares, our result can degenerate to this setting by choosing $m := d$. Our upper bound is able to match the lower bound in [1, Corollary 1] with the same order, which demonstrates the tightness of our upper bound. Besides, our results can recover the result of [16] by taking the effective dimension $k^* = \min\{n, d\}$ (no data spectrum assumption is required here). More discussion on our derived results refers to Appendix A.

## 4 Proof outline and discussion

In this section, we first introduce the structure of the proofs with high level ideas, and then discuss our work with previous literature in terms of the used techniques and the obtained results.

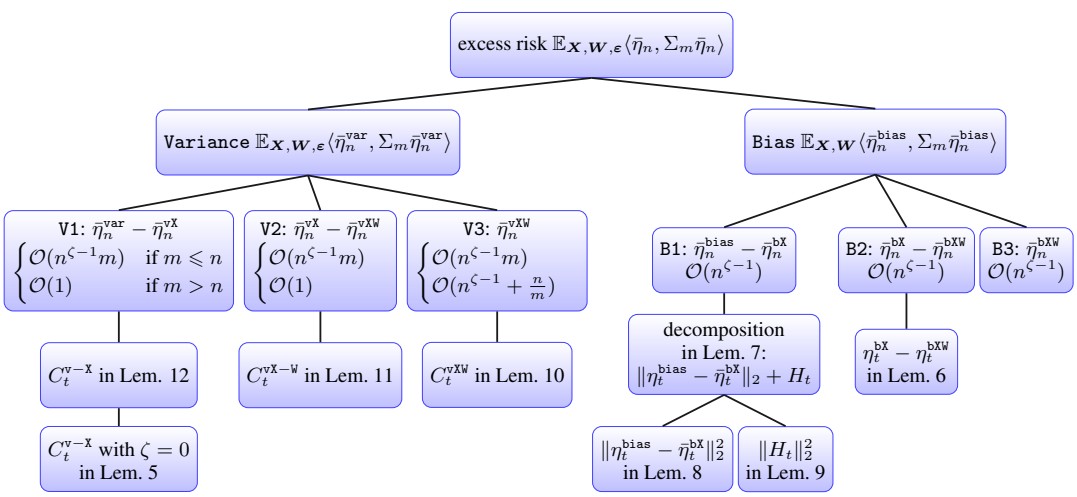

Figure 1: The roadmap of proofs.

## 4.1 Proof outline

We (partly) disentangle the multiple randomness sources on the data $X$, the random features matrix $W$, the noise $\varepsilon$, make full use of statistical properties of covariance operators $\Sigma_m$ and $\widetilde{\Sigma}_m$ in Section 3.2, and provide the respective (bias and variance) upper bounds in terms of multiple randomness sources, as shown in Figure 1.

**Bias**: To bound `Bias`, we need some auxiliary notations. Recall $\Sigma_m = \mathbb{E}_{\boldsymbol{x}}[\varphi(\boldsymbol{x}) \otimes \varphi(\boldsymbol{x})]$ and $\widetilde{\Sigma}_m = \mathbb{E}_{\boldsymbol{x},\boldsymbol{W}}[\varphi(\boldsymbol{x}) \otimes \varphi(\boldsymbol{x})]$, define

$$\eta_t^{\texttt{bX}} = (I - \gamma_t \Sigma_m)\eta_{t-1}^{\texttt{bX}}, \quad \eta_0^{\texttt{bX}} = f^*, \qquad \eta_t^{\texttt{bXW}} = (I - \gamma_t \widetilde{\Sigma}_m)\eta_{t-1}^{\texttt{bXW}}, \quad \eta_0^{\texttt{bXW}} = f^*, \qquad (7)$$

with the average $\bar{\eta}_n^{\texttt{bX}} := \frac{1}{n}\sum_{t=0}^{n-1} \bar{\eta}_t^{\texttt{bX}}$ and $\bar{\eta}_n^{\texttt{bXW}} := \frac{1}{n}\sum_{t=0}^{n-1} \bar{\eta}_t^{\texttt{bXW}}$. Accordingly, $\eta_t^{\texttt{bX}}$ can be regarded as a "deterministic" version of $\eta_t^{\texttt{bias}}$: we omit the randomness on $X$ (data sampling, stochastic gradients) by replacing $[\varphi(\boldsymbol{x})\varphi(\boldsymbol{x})^\top]$ with its expectation $\Sigma_m$. Likewise, $\eta_t^{\texttt{bXW}}$ is a deterministic version of $\eta_t^{\texttt{vX}}$ by replacing $\Sigma_m$ with its expectation $\widetilde{\Sigma}_m$ (randomness on initialization).

By Minkowski inequality, the `Bias` can be decomposed as `Bias` $\lesssim$ `B1` + `B2` + `B3`, where `B1` := $\mathbb{E}_{\boldsymbol{X},\boldsymbol{W}}\big[\langle \bar{\eta}_n^{\texttt{bias}} - \bar{\eta}_n^{\texttt{bX}}, \Sigma_m(\bar{\eta}_n^{\texttt{bias}} - \bar{\eta}_n^{\texttt{bX}})\rangle\big]$ and `B2` := $\mathbb{E}_{\boldsymbol{W}}\big[\langle \bar{\eta}_n^{\texttt{bX}} - \bar{\eta}_n^{\texttt{bXW}}, \Sigma_m(\bar{\eta}_n^{\texttt{bX}} - \bar{\eta}_n^{\texttt{bXW}})\rangle\big]$ and `B3` := $\langle \bar{\eta}_n^{\texttt{bXW}}, \widetilde{\Sigma}_m \bar{\eta}_n^{\texttt{bXW}}\rangle$. Here B3 is a deterministic quantity that is closely connected to model (intrinsic) bias without any randomness; while B1 and B2 evaluate the effect of randomness from $X$ and $W$ on the bias, respectively. The error bounds for them can be directly found in Figure 1.

To bound B3, we directly focus on its formulation by virtue of spectrum decomposition and integral estimation. To bound B2, we have `B2` $= \frac{1}{n^2}\mathbb{E}_{\boldsymbol{W}}\big\|\Sigma_m^{\frac{1}{2}}\sum_{t=0}^{n-1}(\eta_t^{\texttt{bX}} - \eta_t^{\texttt{bXW}})\big\|^2$, where the key part $\eta_t^{\texttt{bX}} - \eta_t^{\texttt{bXW}}$ can be estimated by Lemma 6. To bound B1, it can be further decomposed as (here we use inaccurate expression for description simplicity) `B1` $\lesssim \sum_t \|\eta_t^{\texttt{bX}} - \eta_t^{\texttt{bXW}}\|_2^2 + \sum_t \mathbb{E}_{\boldsymbol{X}}\|H_t\|^2$ in Lemma 7, where $H_{t-1} := [\Sigma_m - \varphi(\boldsymbol{x}_t) \otimes \varphi(\boldsymbol{x}_t)]\eta_{t-1}^{\texttt{bX}}$. The first term can be upper bounded by $\sum_t \|\eta_t^{\texttt{bX}} - \eta_t^{\texttt{bXW}}\|_2^2 \lesssim \text{Tr}(\Sigma_m)n^\zeta \|f^*\|^2$ in Lemma 8, and the second term admits $\sum_t \mathbb{E}_{\boldsymbol{X}}\|H_t\|^2 \lesssim \text{Tr}(\Sigma_m)\|f^*\|^2$ in Lemma 9.

**Variance**: To bound `Variance`, we need some auxiliary notations.

$$\eta_t^{\texttt{vX}} := (I - \gamma_t \Sigma_m)\eta_{t-1}^{\texttt{vX}} + \gamma_t \varepsilon_t \varphi(\boldsymbol{x}_t), \quad \eta_0^{\texttt{vX}} = 0, \qquad (8)$$

$$\eta_t^{\texttt{vXW}} := (I - \gamma_t \widetilde{\Sigma}_m)\eta_{t-1}^{\texttt{vXW}} + \gamma_t \varepsilon_t \varphi(\boldsymbol{x}_t), \quad \eta_0^{\texttt{vXW}} = 0, \qquad (9)$$

with the averaged quantities $\bar{\eta}_n^{\texttt{vX}} := \frac{1}{n}\sum_{t=0}^{n-1} \bar{\eta}_t^{\texttt{vX}}$, $\bar{\eta}_n^{\texttt{vXW}} := \frac{1}{n}\sum_{t=0}^{n-1} \bar{\eta}_t^{\texttt{vXW}}$. Accordingly, $\eta_t^{\texttt{vX}}$ can be regarded as a "semi-stochastic" version of $\eta_t^{\texttt{var}}$: we keep the randomness due to the noise $\varepsilon_t$ but omit

the randomness on $\boldsymbol{X}$ (data sampling) by replacing $[\varphi(\boldsymbol{x})\varphi(\boldsymbol{x})^\top]$ with its expectation $\Sigma_m$. Likewise, $\eta_t^{\mathtt{vXW}}$ can be regarded as a "semi-stochastic" version of $\eta_t^{\mathtt{vX}}$ by replacing $\Sigma_m$ with its expectation $\widetilde{\Sigma}_m$ (randomness on initialization).

By virtue of Minkowski inequality, the $\mathtt{Variance}$ can be decomposed as $\mathtt{Variance} \lesssim \mathtt{V1} + \mathtt{V2} + \mathtt{V3}$, where $\mathtt{V1} := \mathbb{E}_{\boldsymbol{X},\boldsymbol{W},\boldsymbol{\varepsilon}}\big[\langle \bar{\eta}_n^{\mathtt{var}} - \bar{\eta}_n^{\mathtt{vX}}, \Sigma_m(\bar{\eta}_n^{\mathtt{var}} - \bar{\eta}_n^{\mathtt{vX}})\rangle\big]$, $\mathtt{V2} := \mathbb{E}_{\boldsymbol{X},\boldsymbol{W},\boldsymbol{\varepsilon}}\big[\langle \bar{\eta}_n^{\mathtt{vX}} - \bar{\eta}_n^{\mathtt{vXW}}, \Sigma_m(\bar{\eta}_n^{\mathtt{vX}} - \bar{\eta}_n^{\mathtt{vXW}})\rangle\big]$, and $\mathtt{V3} := \mathbb{E}_{\boldsymbol{X},\boldsymbol{W},\boldsymbol{\varepsilon}}\langle \bar{\eta}_n^{\mathtt{vXW}}, \Sigma_m\bar{\eta}_n^{\mathtt{vXW}}\rangle$. Though $\mathtt{V1}$, $\mathtt{V2}$, $\mathtt{V3}$ still interact the multiple randomness, $\mathtt{V1}$ disentangles some randomness on data sampling, $\mathtt{V2}$ discards some randomness on initialization, and $\mathtt{V3}$ focuses on the "minimal" interaction between data sampling, label noise, and initialization. The error bounds for them can be found in Figure 1.

To bound $\mathtt{V3}$, we focus on the formulation of the covariance operator $C_t^{\mathtt{vXW}} := \mathbb{E}_{\boldsymbol{X},\boldsymbol{\varepsilon}}[\eta_t^{\mathtt{vXW}} \otimes \eta_t^{\mathtt{vXW}}]$ in Lemma 10 and the statistical properties of $\widetilde{\Sigma}_m$ and $\Sigma_m$. To bound $\mathtt{V2}$, we need study the covariance operator $C_t^{\mathtt{vX-W}} := \mathbb{E}_{\boldsymbol{X},\boldsymbol{\varepsilon}}[(\eta_t^{\mathtt{vX}} - \eta_t^{\mathtt{vXW}}) \otimes (\eta_t^{\mathtt{vX}} - \eta_t^{\mathtt{vXW}})]$ admitting $\|C_t^{\mathtt{vX-W}}\| \lesssim \|\Sigma_m^2\|_2\|\widetilde{\Sigma}_m\|_2$ in Lemma 11. To bound $\mathtt{V1}$, we need study the covariance operator $C_t^{\mathtt{v-X}} := \mathbb{E}_{\boldsymbol{X},\boldsymbol{\varepsilon}}[(\eta_t^{\mathtt{var}} - \eta_t^{\mathtt{vX}}) \otimes (\eta_t^{\mathtt{var}} - \eta_t^{\mathtt{vX}})]$, as a function of $\zeta \in [0,1)$, admitting $\mathrm{Tr}[C_t^{\mathtt{v-X}}(\zeta)] \lesssim \mathrm{Tr}[C_t^{\mathtt{v-X}}(0)]$ in Lemma 5, and further $C_t^{\mathtt{v-X}} \lesssim \mathrm{Tr}(\Sigma_m)I$ in Lemma 12.

## 4.2 Discussion on techniques

Our proof framework follows [31] that focuses on kernel regression with stochastic approximation in the under-parameterized regimes ($d$ is regarded as finite and much smaller than $n$). Nevertheless, even in the under-parameterized regime, their results can not be directly extended to random features model due to the extra randomness on $\boldsymbol{W}$. For instance, their results depend on [29, Lemma 1] by taking conditional expectation to bridge the connection between $\mathbb{E}[\|\alpha_t\|_2]$ and $\mathbb{E}\langle\alpha_t, \Sigma_m\alpha_t\rangle$. This is valid for $\mathtt{B1}$ but expires on other quantities.

Some technical tools used in this paper follow [16] that focuses on linear regression with constant step-size SGD for benign overfitting. However, our results differ from it in 1) tackling multiple randomness, e.g., stochastic gradients, random features (Gaussian initialization), by introducing another type of error decomposition and several deterministic/randomness covariance operators. We prove nice statistical properties of them for proof, which gets rid of data spectrum assumption in [16]. 2) tackling non-constant step-size SGD setting by introducing new integral estimation techniques. Original techniques on constant step-size in [16] are invalid due to non-homogeneous update rules. The above two points make our proof relatively more intractable and largely different. Besides, their results demonstrate that linear regression with SGD generalizes well (converges with $n$) but has few findings on double descent. Instead, our result depends on $n$ and $m$ (where $d$ is implicitly included in $m$), and is able to explain double descent.

Here we take the estimation for the variance in [16] under the least squares setting as an example to illustrate this.

$$\mathtt{Variance} \lesssim \sum_{t=0}^{n-1} \left\langle I - (I - \gamma\Sigma_d)^{n-t}, I - (I - \gamma\Sigma_d)^t \right\rangle \qquad \text{[Eq. (4.10) in [16]]}$$

In this setting, the effective dimension to tackle $I - (I - \gamma\Sigma_d)^{n-t}$; while our result is based on fast eigenvalue decay of $\widetilde{\Sigma}_m$ in Lemma 1 can direct to bound this. Besides, the homogeneous markov chain under the constant step-size setting is employed [16] for $(I - \gamma\Sigma_d)^{n-t}$, which is naturally invalid under our decaying step-size setting. Instead, we introduce integral estimation techniques to tackle adaptive step-size, see Appendix E for details.

## 5 Numerical Validation

In this section, we provide some numerical experiments in Figure 2 to support our theoretical results and findings. Note that our results go beyond Gaussian data assumption and can be empirically validated on real-world datasets. More experiments can be found in Appendix H.

### 5.1 Behavior of RF for interpolation learning

Here we evaluate the test mean square error (MSE) of RFF regression on the MNIST data set [52], following the experimental setting of [13, 53], to study the generalization performance of minimum-norm solution, see Figure 2(a). More results on regression dataset refer to Appendix H.

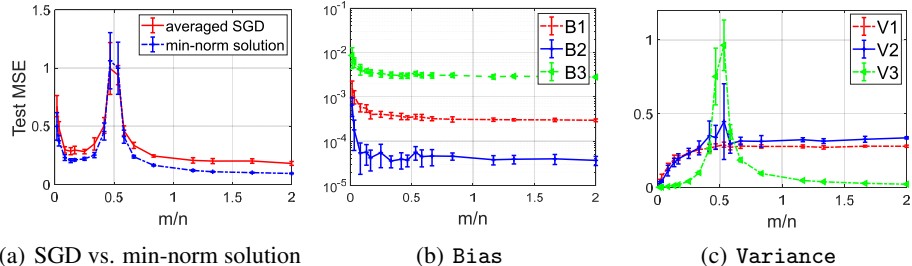

| (a) SGD vs. min-norm solution | (b) Bias | (c) Variance |

Figure 2: Test MSE (mean±std.) of RF regression as a function of the ratio $m/n$ on MNIST data set (digit 3 vs. 7) across the Gaussian kernel, for $d = 784$ and $n = 600$ in (a). The interpolation threshold occurs at $m/n = 0.5$ as the Gaussian kernel outputs the $2m$-feature mapping (instead of $m$), i.e., $\sigma(\boldsymbol{W}\boldsymbol{x}) \in \mathbb{R}^{2m}$. Under this setting, the trends of Bias and Variance are empirically given in (b) and (c).

**Experimental settings:** We take digit 3 vs. 7 as an example, and randomly select 300 training data in these two classes, resulting in $n = 600$ for training. Hence, our setting with $n = 600$, $d = 784$, and tuning $m$ satisfies our realistic high dimensional assumption. The Gaussian kernel $k(\boldsymbol{x}, \boldsymbol{x}') = \exp(-\|\boldsymbol{x} - \boldsymbol{x}'\|_2^2/(2\sigma_0^2))$ is used, where the kernel width $\sigma_0$ is chosen as $\sigma_0^2 = d$ in high dimensional settings such that $\|\boldsymbol{x}\|_2^2/d \sim \mathcal{O}(1)$ in Assumption 1. In our experiment, the initial step-size is set to $\gamma_0 = 1$ and we take the initial point $\boldsymbol{\theta}_0$ near the min-norm solution[3] corrupted with zero-mean, unit-variance Gaussian noise. The experiments are repeated 10 times and the test MSE (mean±std.) can be regarded as a function of the ratio $m/n$ by tuning $m$. Results on different initialization and more epochs of SGD refer to Appendix H.

**SGD vs. minimal-norm solution:** Figure 2(a) shows the test MSE of RF regression with averaged SGD (we take $\zeta = 0.5$ as an example; red line) and minimal-norm solution (blue line). First, we observe the double descent phenomenon: a phase transition on the two sides of the interpolation threshold at $2m = n$ when these two algorithms are employed. Second, in terms of test error, RF with averaged SGD is slightly inferior to that with min-norm solution, but still generalizes well.

### 5.2 Behavior of our error bounds

We have experimentally validated the phase transition and corresponding double descent in the previous section, and here we aim to semi-quantitatively assess our derived bounds for Bias and Variance, see Figure 2(b) and 2(c), respectively. Results of these quantities on different step-size refer to Appendix H.

**Experimental settings:** Since the target function $f^*$, the covariance operators $\Sigma_d$, $\Sigma_m$, and the noise $\varepsilon$ are unknown on the MNIST data set, our experimental evaluation need some assumptions to calculate Bias and Variance. First, we assume the label noise $\varepsilon \sim \mathcal{N}(0, 1)$, which can in turn obtain $f^*(\boldsymbol{x})$ on both training and test data due to $f^*(\boldsymbol{x}) = y - \varepsilon$. Second, the covariance matrices $\Sigma_d$ and $\Sigma_m$ are estimated by the related sample covariance matrices. When using the Gaussian kernel, the covariance matrix $\widetilde{\Sigma}_m$ can be directly computed, see the remark in Lemma 1, where the expectation on $\boldsymbol{x}$ is approximated by Monte Carlo sampling with $n$ training samples. Accordingly, based on the above results, we are ready to calculate $\eta_t^{\texttt{bias}}$ in Eq. (5), $\eta_t^{\texttt{bX}}$, and $\eta_t^{\texttt{bXW}}$ in Eq. (7), respectively, which is further used to approximately compute B1 $:= \mathbb{E}_{\boldsymbol{X}, \boldsymbol{W}}\left[\langle \bar{\eta}_n^{\texttt{bias}} - \bar{\eta}_n^{\texttt{bX}}, \Sigma_m(\bar{\eta}_n^{\texttt{bias}} - \bar{\eta}_n^{\texttt{bX}})\rangle\right]$ (red line) and B2 $:= \mathbb{E}_{\boldsymbol{W}}\left[\langle \bar{\eta}_n^{\texttt{bX}} - \bar{\eta}_n^{\texttt{bXW}}, \Sigma_m(\bar{\eta}_n^{\texttt{bX}} - \bar{\eta}_n^{\texttt{bXW}})\rangle\right]$ (blue line) and B3 $:= \langle \bar{\eta}_n^{\texttt{bXW}}, \widetilde{\Sigma}_m \bar{\eta}_n^{\texttt{bXW}}\rangle$ (green line). The (approximate) computation for Variance can be similar achieved by this process.

**Error bounds for bias:** Figure 2(b) shows the trends of (scaled) B1, B2, and B3. Recall our error bound: B1, B2, B3 $\sim \mathcal{O}(n^{\zeta-1})$, we find that, all of them monotonically decreases at a certain convergence rate when $m$ increases from the under-parameterized regime to the over-parameterized regime. These experimental results coincide with our error bound on them.

---

[3]In our numerical experiments, we only employ single-pass SGD, and thus the initialization is chosen close to minimum norm solution, with more discussion in Appendix H.

**Error bounds for variance:** Figure 2(c) shows the trends of (scaled) `V1`, `V2`, and `V3`. Recall our error bound: in the under-parameterized regime, `V1`, `V2`, and `V3` increase with $m$ at a certain $\mathcal{O}(n^{\zeta-1}m)$ rate; and in the over-parameterized regime, `V1` and `V2` are in $\mathcal{O}(1)$ order while `V3` decreases with $m$. Figure 2(c) shows that, when $2m < n$, `V1` and `V2` monotonically increases with $m$ and then remain unchanged when $2m > n$. Besides, `V3` is observed to be unimodal: firstly increasing when $2m < n$, reaching to the peak at $2m = n$, and then decreasing when $2m > n$, which admits the phase transition at $2m = n$. Accordingly, these findings accord with our theoretical results, and also matches refined results in [11, 27, 15]: the unimodality of variance is a prevalent phenomenon.

## 6 Conclusion

We present non-asymptotic results for RF regression under the averaged SGD setting for understanding double descent under the optimization effect. Our theoretical and empirical results demonstrate that, the error bounds for variance and bias can be unimodal and monotonically decreasing, respectively, which is able to recover the double descent phenomenon. Regarding to constant/adaptive step-size setting, there is no difference between the constant step-size case and the exact minimal-norm solution on the convergence rate; while the polynomial-decay step-size case will slow down the learning rate, but does not change the error bound for variance in over-parameterized regime that converges to $\mathcal{O}(1)$ order, that depends on noise parameter(s).

Our work centers around the RF model, which is still a bit far away from practical neural networks. Theoretical understanding the generalization properties of over-parameterized neural networks is a fundamental but difficult problem. We believe that a comprehensive and thorough understanding of shallow neural networks, e.g., the RF model, is a necessary first step. Besides, we consider the single-pass SGD in our work for simplicity rather than multiple-pass SGD used in practice. This is also an interesting direction for understanding the optimization effect of SGD in the double descent.

Besides, our results obtain the dimension-free bound under both *non-asymptotic* and *asymptotic* regimes. We also need to mention that, our results are also valid under the fixed $d$ setting (which can be larger or smaller than $n$). This is more practical for real-world applications.

## Acknowledgment

The research leading to these results has received funding from the European Research Council under the European Union's Horizon 2020 research and innovation program: ERC Advanced Grant E-DUALITY (787960) and grant agreement n° 725594 - time-data. This paper reflects only the authors' views and the Union is not liable for any use that may be made of the contained information. This work was supported by SNF project – Deep Optimisation of the Swiss National Science Foundation (SNSF) under grant number 200021_205011; Research Council KU Leuven: Optimization frameworks for deep kernel machines C14/18/068; Flemish Government: FWO projects: GOA4917N (Deep Restricted Kernel Machines: Methods and Foundations), PhD/Postdoc grant. This research received funding from the Flemish Government (AI Research Program). This work was supported in part by Ford KU Leuven Research Alliance Project KUL0076 (Stability analysis and performance improvement of deep reinforcement learning algorithms), EU H2020 ICT-48 Network TAILOR (Foundations of Trustworthy AI - Integrating Reasoning, Learning and Optimization), Leuven.AI Institute.
We also thank Zhenyu Liao and Leello Dadi for their helpful discussions on this work.

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
