^{\mathtt{bias}} - \bar{\eta}_n^{\mathtt{bX}}, \Sigma_m(\bar{\eta}_n^{\mathtt{bias}} - \bar{\eta}_n^{\mathtt{bX}})\rangle\big]$ and B2 := $\mathbb{E}_{\boldsymbol{W}}\big[\langle \bar{\eta}_n^{\mathtt{bX}} - \bar{\eta}_n^{\mathtt{bXW}}, \Sigma_m(\bar{\eta}_n^{\mathtt{bX}} - \bar{\eta}_n^{\mathtt{bXW}})\rangle\big]$ and B3 := $\langle \bar{\eta}_n^{\mathtt{bXW}}, \widetilde{\Sigma}_m \bar{\eta}_n^{\mathtt{bXW}}\rangle$. Here B3 is a deterministic quantity that is closely connected to model (intrinsic) bias without any randomness; while B1 and B2 evaluate the effect of randomness from $\boldsymbol{X}$ and $\boldsymbol{W}$ on the bias, respectively. The error bounds for them can be directly found in Figure 1.

To bound B3, we directly focus on its formulation by virtue of spectrum decomposition and integral estimation. To bound B2, we have B2 $= \frac{1}{n^2}\mathbb{E}_{\boldsymbol{W}}\left\|\Sigma_m^{\frac{1}{2}}\sum_{t=0}^{n-1}(\eta_t^{\mathtt{bX}} - \eta_t^{\mathtt{bXW}})\right\|^2$, where the key part $\eta_t^{\mathtt{bX}} - \eta_t^{\mathtt{

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

## A    Comparisons with previous work

### A.1    Problem settings

Here we summarize various representative approaches in Table 1 according to the used data assumption, the type of solution, and the derived results.

Table 1: Comparison of problem settings on analysis of high dimensional random features on double descent.

| | data assumption | solution | result |
|---|---|---|---|
| [1] | Gaussian | closed-form | variance $\nearrow \searrow$ |
| [12] | Gaussian | GD | variance $\nearrow \searrow$ |
| [4] | i.i.d on sphere | closed-form | variance, bias $\nearrow \searrow$ |
| [11] | Gaussian | closed-form | refined [2] |
| [14] | Gaussian | closed-form | $\nearrow \searrow$ |
| [27] | Gaussian | closed-form | refined |
| [54] | Gaussian | closed-form | $\nearrow \searrow$ |
| [28] | Gaussian | closed-form | $\nearrow \searrow$ |
| [13] | general | closed-form | $\nearrow \searrow$ |
| [15] | isotropic features with finite moments | closed form | refined |
| [25] | correlated features with polynomial decay on $\Sigma_d$ | closed form | interpolation learning |
| Ours | sub-Gaussian data | SGD | variance $\nearrow \searrow$, bias $\searrow$ |

[1] A refined decomposition on variance is conducted by sources of randomness on data sampling, initialization, label noise to possess each term [11] or their full decomposition in [27, 15].

Here we discuss the used assumption on data distribution and the discussion on other assumptions is deferred to Appendix B. It can be found that, most papers assume the data to be Gaussian or uniformly distributed on the sphere. The following papers admit weaker assumption on data. Given a correlated features model that is commonly used in high dimensional statistics [1]

$$\boldsymbol{x} = \Sigma_d^{\frac{1}{2}} \boldsymbol{t}, \quad \mathbb{E}[t_i] = 0, \mathbb{V}[t_i] = 1, \quad \text{with } \Sigma_d := \mathbb{E}_{\boldsymbol{x}}[\boldsymbol{x}\boldsymbol{x}^\top], \tag{10}$$

where $\boldsymbol{t} \in \mathbb{R}^d$ has i.i.d entries $t_i$ ($i = 1, 2, \ldots, d$) with zero mean and unit variance. In [25], they further require that each entry is i.i.d sub-Gaussian and $\Sigma_d$ admits polynomial decay on eigenvalues. In [15], the authors consider isotropic features with finite moment, i.e., taking $\Sigma_d := I$ in Eq. (10) and $\mathbb{E}[t_i^{8+\eta}] < \infty$ for any arbitrary positive constant $\eta > 0$. Our model holds for sub-Gaussian, and thus the used data assumption 4 is weaker than them. We also remark that, no assumption on data

distribution is employed [13] but they require that test data "behave" statistically like the training data by concentrated random vectors. Indeed, their data assumption is weaker than ours, but their analysis framework builds on the exact closed-form solution from random matrix theory. Instead, we focus on the SGD setting and thus take a unified perspective on optimization and generalization.

Here we briefly discuss our result with previous work. Compared to [12] on RF optimized by gradient descent under the Gaussian data in an asymptotic view, our non-asymptotic result holds for more general data distribution under the SGD setting. In fact, our data assumption is weaker than most previous work assuming the data to be Gaussian, uniformly spread on a sphere, or isotropic/correlated features (with spectral decay assumption), except [13]. Nevertheless, we extend their asymptotic results relying on the least-squares closed-form solution to non-asymptotic results under the SGD setting, which takes the effect of optimization into consideration. Besides, our result coincides several findings with refined variance decomposition in [11, 27, 15], e.g., the interaction effect can dominate the variance (between samples and initialization); the unimodality of variance is a prevalent phenomenon.

## A.2 Discussion on the tightness of our results

We present the upper bounds of excess risk in this work, and it is natural to ask whether the lower bound can be derived by our proof framework. Unfortunately, the first step in our proof is based on Minkowski inequality such that $\texttt{Bias} \leqslant 3(\texttt{B1} + \texttt{B2} + \texttt{B3})$ and $\texttt{Variance} \leqslant 3(\texttt{V1} + \texttt{V2} + \texttt{V3})$. This could be a limitation of this work, but our derived results are still tight when compared to previous work in both under- and over-parameterized regimes.

First, we compare our result with classical random features regression with SGD in the under-parameterized regime [33]. Under the same standard assumptions, e.g., $f^* \in H$ and label noise with bounded variance, without refined assumptions, e.g., source condition describing the smoothness of $f^*$ and capacity condition describing the "size" of the corresponding $\mathcal{H}$ [45], by taking one-pass over the data (the same setting with our result) and the random features $m = \mathcal{O}(\sqrt{n})$, the excess risk [33] achieves at a certain $\mathcal{O}(1/\sqrt{n})$ rate. Under the same setting with the constant-step size, i.e., $\gamma = 0$, we have

$$\mathbb{E}\|\bar{f}_n - f^*\|^2_{L^2_{\rho_X}} = \underbrace{\texttt{Bias}}_{\mathcal{O}(\frac{1}{n})} + \underbrace{\texttt{Variance}}_{\mathcal{O}(\frac{1}{\sqrt{n}})} \lesssim \frac{1}{\sqrt{n}}\,,$$

which achieves the same learning rate with [33], and has been proved to be optimal in a minimax sense [55] under the standard assumptions. That means, the constant step-size SGD setting incurs no loss in convergence rate when compared to the exact kernel ridge regression.

Second, in the over-parameterized regime, previous work using random matrix theory and replica method provide an exact formulation of the excess risk. Nevertheless, it appears difficult to compare the specific convergence rate due to their complex formulations, and thus we in turn study the tendency. Here we take [11] as an example for comparison. They use conditional expectations to split the variance into label noise, initialization, and data sampling, and the first two terms dominates the variance.
($i$) Our result on bias matches their exact formulation, i.e., monotonically decreasing bias. One slight difference is, their result on bias tends to a constant under the over-parameterized regime while our bias result can converge to zero.
($ii$) Our result on variance admits the same tendency with their result, leading to unimodal variance, where some part(s) are with phase transition and some part(s) firstly monotonically increase during the under-parameterized regime and then remain unchanged during the over-parameterized regime. More importantly, both of the above two results demonstrate that, the variance will finally converge to a constant order, that depends on the variance of label noise $\tau^2$. That means, our (upper bound) result is tight to describe phase transition and the final convergence state (depending on the noise level) when compared to the exact formulation results.

Third, though convergence rates of random features for double descent is non-easy to compare, results on least squares [2, 16] under the over-parameterized regime or interpolation are possible for comparison. Here we take our result by choosing $m := d$ and the constant step-size for least squares setting, and compare their lower bound results to demonstrate the tightness of our result. By virtue of

Lemma 1, we can reformulate our result as

$$\mathbb{E}\|\bar{f}_n - f^*\|_{L^2_{\rho_X}}^2 \lesssim \gamma_0 \tau^2 \begin{cases} \dfrac{1}{n} + \dfrac{d}{n}, & \text{if } d \leqslant n \\ 1 + \dfrac{1}{n} + \dfrac{n}{d}, & \text{if } d > n \end{cases}$$

which matches the same order with [1, Corollary 1].

Besides, when compared to [16], if taking the effective dimension $k^* = \min\{n, d\}$ (no data spectrum assumption is required here), we can recover their result. In fact, our result is able to match their lower bound [2, 16]: `excess risk` $\gtrsim \tau^2(\frac{1}{n} + \frac{n}{d})$ with only one difference on an extra constant when $d > n$.

Based on the above discussion, our upper bound matches previous work with exact formulation or lower bound under various settings, which demonstrates the tightness of our upper bound, and accordingly, our result is able to recover the double descent phenomenon.

## B  Discussion on the used assumptions

Here we give more discussion on the used assumptions, especially Assumptions 2 and 4, which are fair and attainable.

**Discussion on Assumption 2:** $i$) *bounded Hilbert norm:* In high-dimensional asymptotics, this bounded Hilbert norm assumption is commonly used in kernel regression [47, 40, 56], and RF model [48] even though $n$ and $d$ tend to infinity. Here we give an example satisfying this assumption, which is provided by [49, Proposition 4], i.e., linear functions on the sphere can have bounded Hilbert norm for all $d$.

To be specific, assume $f : \mathbb{S}^d \to \mathbb{R}$ such that $f(\boldsymbol{x}) = \boldsymbol{v}^\top \boldsymbol{x}$ for a certain $\boldsymbol{v} \in \mathbb{S}^d$, if we consider the following reproducing kernel

$$k(\boldsymbol{x}, \boldsymbol{x}') = \int_{\mathbb{S}^d} 1_{\{\boldsymbol{\omega}^\top \boldsymbol{x} \geq 0\}} 1_{\{\boldsymbol{\omega}^\top \boldsymbol{x}' \geq 0\}} \mathrm{d}\mu(\boldsymbol{\omega}),$$

where $\mu$ is the probability measure of $\boldsymbol{\omega}$, leading to a zero-order arc-cosine kernel [51] by taking Gaussian measure. Then we have

$$\|f\|_{\mathcal{H}} = \frac{2d\pi}{d - 1} \leqslant 4\pi,$$

which verifies that our assumption on bounded Hilbert norm is attainable.

We also need to remark that, unbounded Hilbert norm of functions can be achieved [49, 57] when $d \to \infty$ in some cases. For example, if we consider the above problem setting but employ the first-order arc-cosine kernel, we have $\|f\|_{\mathcal{H}} \asymp C\sqrt{d}$ for some constant $C$ independent of $d$.

Accordingly, apart from directly regarding it as an assumption, we also give an example such that a function can have bounded Hilbert norm. In fact, in practice $d$ is fixed (larger or smaller than $n$), and accordingly it is reasonable for a fixed ground truth with bounded Hilbert norm.

$ii$) *optimal solution:* We assume that $\mathcal{E}(f)$ admits a unique global optimum. If multiple solutions exist, we choose the minimum norm solution of $\mathbb{E}(f)$, i.e.,

$$f^* = \operatorname*{argmin}_{f \in \mathcal{H}} \|f\|_{\mathcal{H}} \quad \text{s.t. } f \in \operatorname*{argmin}_{f \in \mathcal{H}} \mathcal{E}(f),$$

which follows the setting [16, 38].

**Discussion on Assumption 4:** This assumption follows the spirit of [16, Assumption 2.2]. According to [58, Theorem 5.2.15], assume $\boldsymbol{x}$ is a sub-Gaussian random vector with density of the form $p(\boldsymbol{x}) = \exp(-U(\boldsymbol{x}))$ for the strongly convex function $U : \mathbb{R}^d \to \mathbb{R}$. Accordingly, $\Sigma_m^{-\frac{1}{2}} \boldsymbol{x}$ is sub-Gaussian, and then for any fixed $\boldsymbol{W}$ and PSD operator $A$, we have

$$\mathbb{E}_{\boldsymbol{x}}[\varphi^\top(\boldsymbol{x}) A \varphi(\boldsymbol{x}) \Sigma_m] \lesssim \mathrm{Tr}(A\Sigma_m)\Sigma_m.$$

The proof is similar to [16, Lemma A.1], and thus we omit the proof for simplicity. The sub-Gaussian assumption is common in high dimensional statistics [2], which is weaker than most previous work

on double descent that requires the data to be Gaussian [1, 11, 27, 12], or uniformly spread on a sphere [4, 50], as discussed in Appendix A.

In fact, our proof only requires Assumption 4 valid to some specific PSD operators, e.g., $S^{\mathsf{W}}$, $\mathbb{E}_{\boldsymbol{X}}[\Sigma_m - \varphi(\boldsymbol{x}_t) \otimes \varphi(\boldsymbol{x}_t)]^2$ defined in Appendix D. For description simplicity, we present the requirement on all PSD operators in Assumption 4. Besides, one special case of Assumption 4 by taking $A := I$ is proved by Lemma 3, and accordingly this assumption can be regarded as a natural extension.

This assumption is also similar to the bounded fourth moment in stochastic approximation, see [29, 31, 30, 36, 37] for details.

## C   Results on covariance operators

In this section, we present the proofs of Lemmas 1, 2, 3, 4 on statistical properties of $\Sigma_m$ and $\widetilde{\Sigma}_m$.

### C.1   Proof of Lemma 1 and examples

Here we present the proof of Lemma 1 and then give two examples by taking different activation functions.

#### C.1.1   Proof of Lemma 1

*Proof.* Recall the definition of $\widetilde{\Sigma}_m$, we have

$$\widetilde{\Sigma}_m := \mathbb{E}_{\boldsymbol{x}, \boldsymbol{W}}[\varphi(\boldsymbol{x}) \otimes \varphi(\boldsymbol{x})] = \frac{1}{m}\mathbb{E}_{\boldsymbol{x}, W_{ij} \sim \mathcal{N}(0,1)}\left[\sigma\left(\frac{\boldsymbol{W}\boldsymbol{x}}{\sqrt{d}}\right)\sigma\left(\frac{\boldsymbol{W}\boldsymbol{x}}{\sqrt{d}}\right)^{\top}\right] \in \mathbb{R}^{m \times m}.$$

We consider the diagonal and non-diagonal elements of $\widetilde{\Sigma}_m$ separately.

**Diagonal element:** The diagonal entry $(\widetilde{\Sigma}_m)_{ii} = \frac{1}{m}\mathbb{E}_{\boldsymbol{x}, \boldsymbol{\omega}_i}[\sigma(\frac{\boldsymbol{\omega}_i^{\top}\boldsymbol{x}}{\sqrt{d}})\sigma(\frac{\boldsymbol{\omega}_i^{\top}\boldsymbol{x}}{\sqrt{d}})] = \frac{1}{m}\mathbb{E}_{\boldsymbol{x}}\mathbb{E}_{\boldsymbol{\omega}}[\sigma(\frac{\boldsymbol{\omega}^{\top}\boldsymbol{x}}{\sqrt{d}})]^2$
is the same. In fact, $\mathbb{E}_{\boldsymbol{\omega}}\left[\sigma\left(\frac{\boldsymbol{\omega}^{\top}\boldsymbol{x}}{\sqrt{d}}\right)\right]^2$ is actually a one-dimensional integration by considering the basis $(\boldsymbol{e}_1, \boldsymbol{e}_2, \cdots, \boldsymbol{e}_d)$ with $\boldsymbol{e}_1 = \boldsymbol{x}/\|\boldsymbol{x}\|_2$, and $\boldsymbol{e}_2, \cdots, \boldsymbol{e}_d$ any completion of the basis. This technique is commonly used in [59, 60]. The random feature $\boldsymbol{\omega}$ admits the coordinate representation $\boldsymbol{\omega} = \bar{\omega}_1\boldsymbol{e}_1 + \bar{\omega}_2\boldsymbol{e}_2 + \cdots + \bar{\omega}_d\boldsymbol{e}_d$, and thus

$$\boldsymbol{\omega}^{\top}\boldsymbol{x} = (\bar{\omega}_1\boldsymbol{e}_1 + \bar{\omega}_2\boldsymbol{e}_2 + \cdots + \bar{\omega}_d\boldsymbol{e}_d)^{\top}(\|\boldsymbol{x}\|\boldsymbol{e}_1) = \|\boldsymbol{x}\|\bar{\omega}_1,$$

which implies

$$\begin{aligned}
\mathbb{E}_{\boldsymbol{\omega}}\left[\sigma\left(\frac{\boldsymbol{\omega}^{\top}\boldsymbol{x}}{\sqrt{d}}\right)\right]^2 &= (2\pi)^{-\frac{d}{2}}\int_{\mathbb{R}^d}\left[\sigma\left(\frac{\boldsymbol{\omega}^{\top}\boldsymbol{x}}{\sqrt{d}}\right)\right]^2\exp\left(-\frac{1}{2}\|\boldsymbol{\omega}\|_2^2\right)\mathrm{d}\boldsymbol{\omega} \\
&= \frac{1}{\sqrt{2\pi}}\int_{\mathbb{R}}\left[\sigma\left(\frac{\bar{\omega}_1\|\boldsymbol{x}\|_2}{\sqrt{d}}\right)\right]^2\exp\left(-\frac{\bar{\omega}_1^2}{2}\right)\mathrm{d}\bar{\omega}_1 \\
&= \frac{1}{\sqrt{2\pi}}\int_{\mathbb{R}}[\sigma(z)]^2\exp\left(-\frac{z^2}{2\|\boldsymbol{x}\|^2/d}\right)\frac{\sqrt{d}}{\|\boldsymbol{x}\|_2}\mathrm{d}z \\
&= \mathbb{E}_{z \sim \mathcal{N}(0, \|\boldsymbol{x}\|_2^2/d)}[\sigma(z)]^2,
\end{aligned}$$

where we change the integral variable $z := \frac{\bar{\omega}_1\|\boldsymbol{x}\|_2}{\sqrt{d}}$. Hence we have $(\widetilde{\Sigma}_m)_{ii} = \frac{1}{m}\mathbb{E}_{\boldsymbol{x}}\mathbb{E}_{z \sim \mathcal{N}(0, \|\boldsymbol{x}\|_2^2/d)}[\sigma(z)]^2$.

**Non-diagonal element:** The non-diagonal entry $(\widetilde{\Sigma}_m)_{ij} = \frac{1}{m}\mathbb{E}_{\boldsymbol{x}, \boldsymbol{\omega}_i, \boldsymbol{\omega}_j}[\sigma(\frac{\boldsymbol{\omega}_i^{\top}\boldsymbol{x}}{\sqrt{d}})\sigma(\frac{\boldsymbol{\omega}_j^{\top}\boldsymbol{x}}{\sqrt{d}})^{\top}] = \frac{1}{m}\mathbb{E}_{\boldsymbol{x}}[\mathbb{E}_{\boldsymbol{\omega}}\sigma(\frac{\boldsymbol{\omega}^{\top}\boldsymbol{x}}{\sqrt{d}})]^2$ is the same due to the independence between $\boldsymbol{\omega}_i$ and $\boldsymbol{\omega}_j$. Likewise, it can be represented as a one-dimensional integration

$$(\widetilde{\Sigma}_m)_{ij} = \frac{1}{m}\mathbb{E}_{\boldsymbol{x}}\left[\mathbb{E}_{\boldsymbol{\omega}}\sigma\left(\frac{\boldsymbol{\omega}^{\top}\boldsymbol{x}}{\sqrt{d}}\right)\right]^2 = \frac{1}{m}\mathbb{E}_{\boldsymbol{x}}\left[\mathbb{E}_{z \sim \mathcal{N}(0,1)}\sigma\left(\frac{z\|\boldsymbol{x}\|}{\sqrt{d}}\right)\right]^2 = \frac{1}{m}\mathbb{E}_{\boldsymbol{x}}\left(\mathbb{E}_{z \sim \mathcal{N}(0, \|\boldsymbol{x}\|_2^2/d)}[\sigma(z)]\right)^2.$$

Accordingly, by denoting $a := (\widetilde{\Sigma}_m)_{ii}$ and $b := (\widetilde{\Sigma}_m)_{ij}$, the covariance operator $\widetilde{\Sigma}_m$ can be represented as

$$\widetilde{\Sigma}_m = (a - b)I_m + b\mathbf{1}\mathbf{1}^\top \in \mathbb{R}^{m \times m}\,, \tag{11}$$

with its determinant $\det(\widetilde{\Sigma}_m) = (1 + \frac{mb}{a-b})(a - b)^m$. Hence, the eigenvalues of $\widetilde{\Sigma}_m$ can be naturally obtained by the matrix determinant lemma: $\widetilde{\lambda}_1(\widetilde{\Sigma}_m) = a - b + bm$ and the remaining eigenvalues are $a - b$.

According to [61, Theorem 2.26], by virtue of the Lipschitz function $\sigma(\cdot)$ of Gaussian variables, we have

$$\mathbb{P}\left[ \left| \sigma\left(\frac{\boldsymbol{\omega}^\top \boldsymbol{x}}{\sqrt{d}}\right) - \mathbb{E}_{\boldsymbol{\omega} \sim \mathcal{N}(\mathbf{0}, \boldsymbol{I}_d)} \sigma\left(\frac{\boldsymbol{\omega}^\top \boldsymbol{x}}{\sqrt{d}}\right) \right| \geqslant t \right] \leqslant c \exp(-t^2)\,, \ \ \forall t \geqslant 0\,,$$

which implies that $\sigma\left(\frac{\boldsymbol{\omega}^\top \boldsymbol{x}}{\sqrt{d}}\right)$ is a sub-Gaussian random variable due to its expectation in the $\mathcal{O}(1)$ order. Accordingly, for $z \sim \mathcal{N}(0, \|\boldsymbol{x}\|_2^2/d)$, we have $\mathbb{E}_{\boldsymbol{x}} \mathbb{V}[\sigma(z)] \sim \mathcal{O}(1)$ as $\sigma(z)$ is sub-Gaussian with $\mathcal{O}(1)$ norm and its finite second moment, i.e., $\mathbb{V}[\sigma(z)] \sim \mathcal{O}(1)$, which implies $\widetilde{\lambda}_2 = \frac{1}{m}\mathbb{E}_{\boldsymbol{x}}\mathbb{V}[\sigma(z)] \sim \mathcal{O}(1/m)$. Finally, we conclude the proof. $\square$

### C.1.2 Examples

In our analysis, we assume $\sigma(\cdot) : \mathbb{R} \to \mathbb{R}$ with single-output for description simplicity. In fact, our results can be easily extended to multiple-output cases, e.g., the Gaussian kernel corresponding to $\sigma(x) = [\cos(x), \sin(x)]^\top$. Here we give two examples, including single- and multiple-output: arc-cosine kernel that corresponds to the ReLU function $\sigma(x) = \max\{0, x\}$; and the Gaussian kernel.

**Arc-cosine kernel:** We begin with calculation of arc-cosine kernels due to its related single-output activation function. Denote $\widetilde{z} := \max\{0, z\}$ with $z \sim \mathcal{N}(0, \|\boldsymbol{x}\|_2^2/d)$, it is subject to the Rectified Gaussian distribution admitting (refer to [25])

$$\mathbb{E}[\widetilde{z}] = \frac{\|\boldsymbol{x}\|_2}{\sqrt{2d\pi}}\,, \quad \mathbb{E}[\widetilde{z}]^2 = \frac{\|\boldsymbol{x}\|_2^2}{2d}\,, \quad \mathbb{V}[\widetilde{z}] = \frac{\|\boldsymbol{x}\|_2^2}{2d}\left(1 - \frac{1}{\pi}\right)\,.$$

Accordingly, recall the sample covariance operator $\Sigma_d := \mathbb{E}_{\boldsymbol{x}}[\boldsymbol{x}\boldsymbol{x}^\top]$, the diagonal elements are the same

$$(\widetilde{\Sigma}_m)_{ii} = \frac{1}{m}\mathbb{E}_{\boldsymbol{x}}\mathbb{E}_{z \sim \mathcal{N}(0, \|\boldsymbol{x}\|_2^2/d)}[\sigma(z)]^2 = \frac{1}{2md}\mathbb{E}_{\boldsymbol{x}}\|\boldsymbol{x}\|_2^2 = \frac{1}{2md}\mathrm{Tr}(\Sigma_d)\,, \quad i = 1, 2, \ldots, m\,,$$

and the non-diagonal elements are the same

$$(\widetilde{\Sigma}_m)_{ij} = \frac{1}{m}\mathbb{E}_{\boldsymbol{x}}\left(\mathbb{E}_{z \sim \mathcal{N}(0, \|\boldsymbol{x}\|_2^2/d)}[\sigma(z)]\right)^2 = \frac{1}{2md\pi}\mathrm{Tr}(\Sigma_d)\,, \quad i, j = 1, 2, \ldots, m \text{ with } i \neq j\,.$$

**Gaussian kernels:** Briefly, if we choose $\sigma(x) = [\cos(x), \sin(x)]^\top$, a multiple-output version, RF actually approximates the Gaussian kernel with $\varphi(\boldsymbol{x}) \in \mathbb{R}^{2m}$ in Eq. (1), resulting in $\widetilde{\Sigma}_m \in \mathbb{R}^{2m \times 2m}$. In this case, $\widetilde{\Sigma}_m = S_1 \oplus S_2$ is a block diagonal matrix, where $\oplus$ is the direct sum. By denoting $\vartheta := \|\boldsymbol{x}\|_2^2/d$, the matrix $S_1 \in \mathbb{R}^{m \times m}$ has the same diagonal elements $[S_1]_{ii} = \frac{1}{2m}\mathbb{E}_{\boldsymbol{x}}\left[1 + \exp(-2\vartheta)\right]$, and the same non-diagonal elements $\frac{1}{m}\mathbb{E}_{\boldsymbol{x}}\left[\exp(-\vartheta)\right]$. The matrix $S_2$ is diagonal with $[S_2]_{ii} = \frac{1}{2m}\mathbb{E}_{\boldsymbol{x}}\left[1 - \exp(-2\vartheta)\right]$. In this case, $\widetilde{\Sigma}_m$ admits three distinct eigenvalues: the largest eigenvalue at $\mathcal{O}(1)$ order, and the remaining two eigenvalues at $\mathcal{O}(1/m)$ order.

According to Bochner's theorem [62], we have $\mathbb{E}[\cos(\boldsymbol{\omega}^\top \boldsymbol{z})] = \exp(-z^2/2)$ and $\mathbb{E}[\cos^2(\boldsymbol{\omega}^\top \boldsymbol{z})] = \frac{1+\exp(-2z^2)}{2}$ for $\boldsymbol{\omega} \sim \mathcal{N}(\mathbf{0}, \boldsymbol{I}_d)$ and $z := \|\boldsymbol{z}\|_2$. In fact, this can be computed by two steps: first transforming the $d$-dimensional integration to a one-dimensional integral as discussed before; and then computing the integral by virtue of the Euler's formula $\exp(-\mathrm{i}x) = \cos x + \mathrm{i}\sin x$. For instance,

$$\mathbb{E}[\cos(\boldsymbol{\omega}^\top \boldsymbol{z})] = \mathbb{E}_{x \sim \mathcal{N}(0, \|\boldsymbol{z}\|_2^2)} \cos x = \frac{1}{\sqrt{2\pi}\|\boldsymbol{z}\|_2}\mathrm{Re}\left[\int_{\mathbb{R}} \exp\left(-\frac{x^2}{2\|\boldsymbol{z}\|_2^2}\right)\exp(\mathrm{i}x)\mathrm{d}x\right]$$

$$= \exp\left(-\frac{\|\boldsymbol{z}\|_2^2}{2}\right)\mathrm{Re}\left[\frac{1}{\sqrt{2\pi}\|\boldsymbol{z}\|_2}\int_{\mathbb{R}}\exp\left(-\frac{(x - \mathrm{i}\|\boldsymbol{z}\|_2^2)}{2\|\boldsymbol{z}\|_2^2}\right)\mathrm{d}x\right]$$

$$= \exp\left(-\frac{\|\boldsymbol{z}\|_2^2}{2}\right)\,.$$

Similarly, we have $\mathbb{E}[\sin(\boldsymbol{\omega}^\top \boldsymbol{z})] = 0$ and $\mathbb{E}[\sin^2(\boldsymbol{\omega}^\top \boldsymbol{z})] = \frac{1-\exp(-2z^2)}{2}$ for $\boldsymbol{\omega} \sim \mathcal{N}(\boldsymbol{0}, \boldsymbol{I}_d)$ and $z := \|\boldsymbol{z}\|_2$.

Based on the above results, for the Gaussian kernel, the expected covariance operator $\widetilde{\Sigma}_m$ is a block diagonal matrix

$$\widetilde{\Sigma}_m = \left[ \begin{array}{c|c} \boldsymbol{S}_1 & \boldsymbol{0} \\ \hline \boldsymbol{0} & \boldsymbol{S}_2 \end{array} \right] \in \mathbb{R}^{2m \times 2m} \,,$$

where $\boldsymbol{S}_1 \in \mathbb{R}^{m \times m}$ has the same diagonal elements and the same non-diagonal elements:

$$[\boldsymbol{S}_1]_{ii} = \frac{1}{m} \mathbb{E}_{\boldsymbol{x}, \boldsymbol{\omega}} \left[ \cos\left( \frac{\boldsymbol{\omega}^\top \boldsymbol{x}}{\sqrt{d}} \right) \right]^2 = \frac{1}{2m} \mathbb{E}_{\boldsymbol{x}} \left[ 1 + \exp\left( -2 \frac{\|\boldsymbol{x}\|_2^2}{d} \right) \right] \,, \quad i = 1, 2, \ldots, m \,,$$

$$[\boldsymbol{S}_1]_{ij} = \frac{1}{m} \mathbb{E}_{\boldsymbol{x}} \left[ \mathbb{E}_{\boldsymbol{\omega}} \cos\left( \frac{\boldsymbol{\omega}^\top \boldsymbol{x}}{\sqrt{d}} \right) \right]^2 = \frac{1}{m} \mathbb{E}_{\boldsymbol{x}} \left[ \exp\left( -\frac{\|\boldsymbol{x}\|_2^2}{d} \right) \right] \,, \quad i, j = 1, 2, \ldots, m \text{ with } i \neq j \,.$$

The matrix $\boldsymbol{S}_2 \in \mathbb{R}^{m \times m}$ is diagonal with

$$[\boldsymbol{S}_2]_{ii} = \frac{1}{m} \mathbb{E}_{\boldsymbol{x}, \boldsymbol{\omega}} \left[ \sin\left( \frac{\boldsymbol{\omega}^\top \boldsymbol{x}}{\sqrt{d}} \right) \right]^2 = \frac{1}{2m} \mathbb{E}_{\boldsymbol{x}} \left[ 1 - \exp\left( -2 \frac{\|\boldsymbol{x}\|_2^2}{d} \right) \right] \,, \quad i = 1, 2, \ldots, m \,.$$

Accordingly, $\widetilde{\Sigma}_m$ has three distinct eigenvalues

$$\widetilde{\lambda}_1 = \mathbb{E}_{\boldsymbol{x}} \left[ \exp\left( -\frac{\|\boldsymbol{x}\|_2^2}{d} \right) \right] + \frac{1}{2m} \mathbb{E}_{\boldsymbol{x}} \left[ 1 - \exp\left( -\frac{\|\boldsymbol{x}\|_2^2}{d} \right) \right]^2 \sim \mathcal{O}(1) \,,$$

$$\widetilde{\lambda}_2 = \frac{1}{2m} \mathbb{E}_{\boldsymbol{x}} \left[ 1 - \exp\left( -2 \frac{\|\boldsymbol{x}\|_2^2}{d} \right) \right] \sim \mathcal{O}\left( \frac{1}{m} \right) \,,$$

$$\widetilde{\lambda}_3 = \frac{1}{2m} \mathbb{E}_{\boldsymbol{x}} \left[ 1 - \exp\left( -\frac{\|\boldsymbol{x}\|_2^2}{d} \right) \right]^2 \sim \mathcal{O}\left( \frac{1}{m} \right) \,.$$

In this case, we can also get the similar claim on spectra of $\widetilde{\Sigma}_m$ with the single-output version: $\widetilde{\Sigma}_m$ admits the largest eigenvalue at $\mathcal{O}(1)$ order, and the remaining eigenvalues are at $\mathcal{O}(1/m)$ order.

## C.2 Proof of Lemma 2

*Proof.* As discussed before, $\sigma\left( \frac{\boldsymbol{\omega}^\top \boldsymbol{x}}{\sqrt{d}} \right)$ is a sub-Gaussian random variable with the $\mathcal{O}(1)$ sub-Gaussian norm order. Hence, $\|\Sigma_m - \widetilde{\Sigma}_m\|_2$ is a sub-exponential random variable with

$$\|\Sigma_m - \widetilde{\Sigma}_m\|_2 \leqslant \|\Sigma_m\|_2 + \|\widetilde{\Sigma}_m\|_2 = \frac{1}{m} \left\| \mathbb{E}_{\boldsymbol{x}} \left[ \sigma\left( \frac{\boldsymbol{W}\boldsymbol{x}}{\sqrt{d}} \right) \sigma\left( \frac{\boldsymbol{W}\boldsymbol{x}}{\sqrt{d}} \right)^\top \right] \right\|_2 + \mathcal{O}(1) \quad \text{[using Lemma 1]}$$

$$\leqslant \frac{1}{m} \mathbb{E}_{\boldsymbol{x}} \left\| \sigma\left( \frac{\boldsymbol{W}\boldsymbol{x}}{\sqrt{d}} \right) \right\|_2^2 + \mathcal{O}(1) \quad \text{[Jensen's inequality]}$$

$$\lesssim \frac{1}{m} \left( \mathbb{E}_{\boldsymbol{x}} \|\sigma(\boldsymbol{0}_m)\|_2^2 + \mathbb{E}_{\boldsymbol{x}} \left\| \frac{\boldsymbol{W}\boldsymbol{x}}{\sqrt{d}} \right\|_2^2 \right) + \mathcal{O}(1) \quad [\sigma: \text{Lipschitz continuous}]$$

$$\lesssim \mathcal{O}(1) + \frac{1}{md} \sum_{i=1}^m \boldsymbol{\omega}_i^\top \mathbb{E}_{\boldsymbol{x}}[\boldsymbol{x}\boldsymbol{x}^\top] \boldsymbol{\omega}_i \quad [\text{using } \|\Sigma_d\|_2 < \infty]$$

$$\lesssim \frac{1}{d} \|\boldsymbol{\omega}\|_2^2 \quad [\text{here } \boldsymbol{\omega} \sim \mathcal{N}(\boldsymbol{0}, \boldsymbol{I}_d)] \,,$$

where $\|\boldsymbol{\omega}\|_2^2$ is a $\chi^2(d)$ random variable, and thus $\|\Sigma_m - \widetilde{\Sigma}_m\|_2$ has sub-exponential norm at $\mathcal{O}(1)$ order. Accordingly, the high moment $\mathbb{E}\|\Sigma_m\|_2^p < \infty$ holds for finite $p$. Following the above derivation, we can also conclude that $\text{Tr}(\Sigma_m)$ has the sub-exponential norm $\mathcal{O}(1)$, i.e.

$$\text{Tr}(\Sigma_m) = \frac{1}{m} \mathbb{E}_{\boldsymbol{x}} \text{Tr} \left[ \sigma\left( \frac{\boldsymbol{W}\boldsymbol{x}}{\sqrt{d}} \right) \sigma\left( \frac{\boldsymbol{W}\boldsymbol{x}}{\sqrt{d}} \right)^\top \right] = \frac{1}{m} \mathbb{E}_{\boldsymbol{x}} \left\| \sigma\left( \frac{\boldsymbol{W}\boldsymbol{x}}{\sqrt{d}} \right) \right\|_2^2 \lesssim \frac{1}{d} \|\boldsymbol{\omega}\|_2^2 \,.$$

Likewise, we can derive $\text{Tr}(\Sigma_m^2) < \infty$ in the similar fashion. $\qquad \square$

## C.3 Proof of Lemma 3

*Proof.* The first inequality naturally holds, and so we focus on the second inequality. Denote $\Phi := \mathbb{E}_{\boldsymbol{x}, \boldsymbol{W}}[\varphi(\boldsymbol{x}) \otimes \varphi(\boldsymbol{x}) \otimes \varphi(\boldsymbol{x}) \otimes \varphi(\boldsymbol{x})]$, its diagonal elements are the same

$$\Phi_{ii} = \frac{m-1}{m^2} \mathbb{E}_{\boldsymbol{x}} \left( \mathbb{E}_{z \sim \mathcal{N}(0, \|\boldsymbol{x}\|_2^2/d)}[\sigma(z)]^2 \right)^2 + \frac{1}{m^2} \mathbb{E}_{\boldsymbol{x}} \mathbb{E}_{z \sim \mathcal{N}(0, \|\boldsymbol{x}\|_2^2/d)}[\sigma(z)]^4 \sim \mathcal{O}\left( \frac{1}{m} \right) .$$

Its non-diagonal elements $\Phi_{ij}$ with $i \neq j$ are the same

$$\Phi_{ij} = \frac{m-3}{m^2} \mathbb{E}_{\boldsymbol{x}} \left[ \left( \mathbb{E}_{z \sim \mathcal{N}(0, \|\boldsymbol{x}\|_2^2/d)}[\sigma(z)] \right)^2 \mathbb{E}_{z \sim \mathcal{N}(0, \|\boldsymbol{x}\|_2^2/d)}[\sigma(z)]^2 \right]$$

$$+ \frac{2}{m^2} \mathbb{E}_{\boldsymbol{x}} \left[ \mathbb{E}_{z \sim \mathcal{N}(0, \|\boldsymbol{x}\|_2^2/d)}[\sigma(z)]^3 \mathbb{E}_{z \sim \mathcal{N}(0, \|\boldsymbol{x}\|_2^2/d)}[\sigma(z)] \right] ,$$

where the first term is in $\mathcal{O}(\frac{1}{m})$ order and the second term is in $\mathcal{O}(\frac{1}{m^2})$ order. By denoting $a := (\widetilde{\Sigma}_m)_{ii}$, $b := (\widetilde{\Sigma}_m)_{ij}$ as given by Lemma 1, $A := \Phi_{ii}$, and $B := \Phi_{ij}$, the operator $r\mathrm{Tr}(\widetilde{\Sigma}_m)\widetilde{\Sigma}_m - \Phi$ can be represented as

$$r\mathrm{Tr}(\widetilde{\Sigma}_m)\Sigma_m - \Phi = [rm(a-b) - A + B] I_m + (rmab - B)\mathbf{1}\mathbf{1}^\top ,$$

of which the smallest eigenvalue is $rma(a-b) - A + B$. Accordingly, to ensure the positive definiteness of $r\mathrm{Tr}(\widetilde{\Sigma}_m)\widetilde{\Sigma}_m - \Phi$, which implies $\mathbb{E}_{\boldsymbol{W}} \left( \mathbb{E}_{\boldsymbol{x}} \left( [\varphi(\boldsymbol{x}) \otimes \varphi(\boldsymbol{x})]A[\varphi(\boldsymbol{x}) \otimes \varphi(\boldsymbol{x})] \right) \right) \preccurlyeq r\mathrm{Tr}(\widetilde{\Sigma}_m)\widetilde{\Sigma}_m$, we require its smallest eigenvalue is non-negative, i.e., $rma(a-b) - A + B \geqslant 0$. That means, $r$ should satisfies

$$r \geqslant \frac{A-B}{ma(a-b)} = \frac{A-B}{\frac{1}{m}\mathbb{E}_{\boldsymbol{x}}\mathbb{E}_{z \sim \mathcal{N}(0, \|\boldsymbol{x}\|_2^2/d)}[\sigma(z)]^2 \mathbb{E}_{\boldsymbol{x}}\mathbb{V}[\sigma(z)]} . \tag{12}$$

Since $A - B$ admits

$$A - B \leqslant \frac{1}{m}\mathbb{E}_{\boldsymbol{x}}\mathbb{E}_z[\sigma(z)]^2 \mathbb{E}_{\boldsymbol{x}}\mathbb{V}[\sigma(z)] + \mathcal{O}\left( \frac{1}{m^2} \right) ,$$

then by taking $r := 1 + \mathcal{O}\left( \frac{1}{m} \right)$, the condition in Eq. (12) satisfies, and thus $r\mathrm{Tr}(\widetilde{\Sigma}_m)\widetilde{\Sigma}_m - \Phi$ is positive definite, which concludes the proof. $\qquad\square$

# D  Preliminaries on PSD operators

In this section, we first define some stochastic/deterministic PSD operators that follow [63, 16] in stochastic approximation, and then present Lemma 5 that is based on PSD operators and is needed to estimate B1 and V1. Note that, the PSD operators will make the notation in our proof simple and clarity but do not change the proof itself.

Following [63, 16], we define several stochastic PSD operators as below. Given the random features matrix $\boldsymbol{W}$ and any PSD operator $A$, define

$$S^{\mathtt{W}} := \mathbb{E}_{\boldsymbol{x}}[\varphi(\boldsymbol{x}) \otimes \varphi(\boldsymbol{x}) \otimes \varphi(\boldsymbol{x}) \otimes \varphi(\boldsymbol{x})], \quad \widetilde{S}^{\mathtt{W}} := \Sigma_m \otimes \Sigma_m ,$$

$$S^{\mathtt{W}} \circ A := \mathbb{E}_{\boldsymbol{x}} \left[ \varphi(\boldsymbol{x})^\top \varphi(\boldsymbol{x}) A \varphi(\boldsymbol{x}) \otimes \varphi(\boldsymbol{x}) \right], \quad \widetilde{S}^{\mathtt{W}} \circ A := \Sigma_m A \Sigma_m ,$$

where the superscript $\mathtt{W}$ denotes the randomness dependency on the random feature matrix $\boldsymbol{W}$. Besides, for any $\gamma_i$ ($i = 1, 2, \ldots, n$), define the following operators

$$(I - \gamma_i T^{\mathtt{W}}) \circ A := \mathbb{E}_{\boldsymbol{x}} \left( [I - \gamma_i \varphi(\boldsymbol{x}) \otimes \varphi(\boldsymbol{x})]A[I - \gamma_i \varphi(\boldsymbol{x}) \otimes \varphi(\boldsymbol{x})] \right)$$

$$(I - \gamma_i \widetilde{T}^{\mathtt{W}}) \circ A := (I - \gamma_i \Sigma_m) A (I - \gamma_i \Sigma_m) ,$$

associated with two corresponding operators (that depend on $\gamma_i$)

$$T^{\mathtt{W}} := \Sigma_m \otimes I + I \otimes \Sigma_m - \gamma_i S^{\mathtt{W}}, \quad \widetilde{T}^{\mathtt{W}} := \Sigma_m \otimes I + I \otimes \Sigma_m - \gamma_i \widetilde{S}^{\mathtt{W}} .$$

Clearly, the above operators $S^{\mathtt{W}}$, $\widetilde{S}^{\mathtt{W}}$, $(I - \gamma_i T^{\mathtt{W}})$, $(I - \gamma_i \widetilde{T}^{\mathtt{W}})$, $T^{\mathtt{W}}$, and $\widetilde{T}^{\mathtt{W}}$ are PSD, and $S^{\mathtt{W}} \succcurlyeq \widetilde{S}^{\mathtt{W}}$. The proof is similar to [16, Lemma B.1] and thus we omit it here.

Further, if $\gamma_0 < 1/\mathrm{Tr}(\Sigma_m)$, the PSD operator $I - \gamma_i \Sigma_m$ $(i = 1, 2, \ldots, n)$ is a contraction map, and thus for any PSD operator $A$ and step-size $\gamma_i$, the following exists

$$\sum_{t=0}^{\infty} (I - \gamma_i \widetilde{T}^{\mathtt{W}})^t \circ A = \sum_{t=0}^{\infty} (I - \gamma_i \Sigma_m)^t A (I - \gamma_i \Sigma_m)^t \,.$$

Hence, $(\widetilde{T}^{\mathtt{W}})^{-1} := \gamma_i \sum_{t=0}^{\infty} (I - \gamma_i \widetilde{T}^{\mathtt{W}})^t$ exists and is PSD. We need to remark that, though $\mathrm{Tr}(\Sigma_m)$ is a random variable, it is with a sub-exponential $\mathcal{O}(1)$ norm. That means, this holds with exponentially high probability.

Based on the above stochastic operators, we define several deterministic PSD ones by taking the expectation over $\boldsymbol{W}$ as below. For any given $\gamma_i$ $(i = 1, 2, \ldots, n)$, we have the following PSD operators

$$S := \mathbb{E}_{\boldsymbol{W}}[\Sigma_m \otimes \Sigma_m], \quad \widetilde{S} := \widetilde{\Sigma}_m \otimes \widetilde{\Sigma}_m \,,$$

$$T := \widetilde{\Sigma}_m \otimes I + I \otimes \widetilde{\Sigma}_m - \gamma_i S, \quad \widetilde{T} := \widetilde{\Sigma}_m \otimes I + I \otimes \widetilde{\Sigma}_m - \gamma_i \widetilde{S} \,,$$

$$S \circ A := \mathbb{E}_{\boldsymbol{W}}[\Sigma_m A \Sigma_m], \quad \widetilde{S} \circ A := \widetilde{\Sigma}_m A \widetilde{\Sigma}_m \,,$$

$$(I - \gamma_i T) \circ A := \mathbb{E}_{\boldsymbol{W}}[(I - \gamma_i \Sigma_m) A (I - \gamma_i \Sigma_m)], \quad (I - \gamma_i \widetilde{T}) \circ A := (I - \gamma_i \widetilde{\Sigma}_m) A (I - \gamma_i \widetilde{\Sigma}_m) \,,$$

which implies $\widetilde{T} - T = \gamma_i (S - \widetilde{S})$.

Based on the above PSD operators, we present a lemma here that is used to estimate B1 and V1.[4]

**Lemma 5.** *Under Assumptions 1, 2, 3, 4 with $r' \geqslant 1$, denote*

$$D_t^{\mathtt{v-x}} := \sum_{s=1}^{t} \prod_{i=s+1}^{t} (I - \gamma_i T^{\mathtt{W}}) \circ \gamma_s^2 B \Sigma_m \,, \tag{13}$$

*with a scalar $B$ independent of $k$, if the step-size $\gamma_t := \gamma_0 t^{-\zeta}$ with $\zeta \in [0, 1)$ satisfies*

$$\gamma_0 < \min \left\{ \frac{1}{r' \mathrm{Tr}(\Sigma_m)}, \frac{1}{c' \mathrm{Tr}(\Sigma_m)} \right\} \,,$$

*where the constant $c'$ is defined as*

$$c' := \begin{cases} 1, & \text{if } \zeta = 0 \,, \\ \dfrac{1}{1 - 2^{-\zeta}}, & \text{if } \zeta \in (0, 1) \,. \end{cases} \tag{14}$$

*Then $D_t^{\mathtt{v-x}}$ can be upper bounded by*

$$D_t^{\mathtt{v-x}} \preccurlyeq \frac{\gamma_0 B}{1 - \gamma_0 r' \mathrm{Tr}(\Sigma_m)} I \,.$$

**Remark:** The PSD operator $I - \gamma_i T^{\mathtt{W}}$ cannot be guaranteed as a contraction map since we cannot directly choose $\gamma_0 < \frac{1}{\mathrm{Tr}[\varphi(\boldsymbol{x}) \varphi(\boldsymbol{x})^{\top}]}$ for general data $\boldsymbol{x}$. However, its summation in Eq. (13) can be still bounded by our lemma. In our work, we set $B := r' \mathrm{Tr}(\Sigma_m)$ for estimate B1, and $B := \tau^2 r' \gamma_0 [\mathrm{Tr}(\Sigma_m) + \gamma_0 \mathrm{Tr}(\Sigma_m^2)]$ to bound V1, respectively.

*Proof.* Our proof can be divided into two parts: one is to prove $\mathrm{Tr}[D_t^{\mathtt{v-x}}(\zeta)] \leqslant \mathrm{Tr}[D_t^{\mathtt{v-x}}(0)]$ for any $\zeta \in [0, 1)$; the other is to provide the upper bound of $D_t^{\mathtt{v-x}}(0)$. We focus on the first part and the proof in the second part follows [63, Lemmas 3 and 5] and [16, Lemma B.4].

---

[4]Our proofs on the remaining quantities including V2, V3, B2, B3 do not use PSD operators.

The quantity $\mathrm{Tr}[D_t^{\mathtt{v}-\mathtt{x}}(\zeta)]$ admits the following representation by the definition of $I - \gamma_i T^{\mathtt{W}}$

$$\mathrm{Tr}[D_t^{\mathtt{v}-\mathtt{x}}(\zeta)] = \sum_{s=1}^{t} \prod_{i=s+1}^{t} \mathrm{Tr}\left[(I - \gamma_i T^{\mathtt{W}}) \circ \gamma_s^2 B \Sigma_m\right]$$

$$= \sum_{s=1}^{t} B \gamma_s^2 \prod_{i=s+1}^{t} \mathrm{Tr}\left(\mathbb{E}_{\boldsymbol{x}}[I - \gamma_i \varphi(\boldsymbol{x}) \otimes \varphi(\boldsymbol{x})] \Sigma_m [I - \gamma_i \varphi(\boldsymbol{x}) \otimes \varphi(\boldsymbol{x})]\right)$$

$$= B \sum_{s=1}^{t} \gamma_s^2 \prod_{i=s+1}^{t} \mathrm{Tr}\left(\Sigma_m - 2\gamma_i \Sigma_m^2 + \gamma_i^2 \Sigma_m \mathbb{E}_{\boldsymbol{x}}\left[\varphi(\boldsymbol{x}) \otimes \varphi(\boldsymbol{x}) \otimes \varphi(\boldsymbol{x}) \otimes \varphi(\boldsymbol{x})\right]\right).$$

Based on the above results, we have

$$\mathrm{Tr}[D_t^{\mathtt{v}-\mathtt{x}}(0)] - \mathrm{Tr}[D_t^{\mathtt{v}-\mathtt{x}}(\zeta)] = B \sum_{s=1}^{t} \prod_{i=s+1}^{t} \mathrm{Tr}\left(\Sigma_m\left[(\gamma_0^2 - \gamma_s^2)I - 2(\gamma_0^3 - \gamma_s^2\gamma_i)\Sigma_m\right.\right.$$

$$\left.\left. + (\gamma_0^4 - \gamma_i^2\gamma_s^2)\mathbb{E}_{\boldsymbol{x}}\left[\varphi(\boldsymbol{x}) \otimes \varphi(\boldsymbol{x}) \otimes \varphi(\boldsymbol{x}) \otimes \varphi(\boldsymbol{x})\right]\right]\right)$$

$$\geqslant B \sum_{s=1}^{t} \prod_{i=s+1}^{t} \mathrm{Tr}\left(\Sigma_m\left[(\gamma_0^2 - \gamma_s^2)I - 2(\gamma_0^3 - \gamma_s^2\gamma_i)\Sigma_m + (\gamma_0^4 - \gamma_i^2\gamma_s^2)\Sigma_m^2\right]\right)$$

$$= B \sum_{s=1}^{t} \prod_{i=s+1}^{t} \sum_{j=1}^{m} \left(\lambda_j\left[(\gamma_0^2 - \gamma_s^2) - 2(\gamma_0^3 - \gamma_s^2\gamma_i)\lambda_j + (\gamma_0^4 - \gamma_i^2\gamma_s^2)\lambda_j^2\right]\right)$$

$$= B \sum_{s=1}^{t} \prod_{i=s+1}^{t} \sum_{j=1}^{m} \left(\lambda_j\left[(\gamma_0^4 - \gamma_i^2\gamma_s^2)\left(\lambda_j - \frac{\gamma_0^3 - \gamma_s^2\gamma_i}{\gamma_0^4 - \gamma_i^2\gamma_s^2}\right)^2 - \frac{\gamma_0^2\gamma_s^2(\gamma_0 - \gamma_i)^2}{\gamma_0^4 - \gamma_i^2\gamma_s^2}\right]\right).$$

Accordingly, $\mathrm{Tr}[D_t^{\mathtt{v}-\mathtt{x}}(0)] - \mathrm{Tr}[D_t^{\mathtt{v}-\mathtt{x}}(\zeta)] \geqslant 0$ naturally holds when $\zeta = 0$. When $\zeta \in (0, 1)$, it holds if $\lambda_j \leqslant \frac{\gamma_0^3 - \gamma_s^2\gamma_i - \gamma_0^2\gamma_s + \gamma_0\gamma_s\gamma_i}{\gamma_0^4 - \gamma_s^2\gamma_i^2}$ with $j = 1, 2, \ldots, m$. This condition can be satisfied by

**Case 1** (if $s = 1$). *In this case, $\gamma_1 = \gamma_0$ and we have*

$$\lambda_j \leqslant \mathrm{Tr}(\Sigma_m) \leqslant \frac{1}{2\gamma_0} \leqslant \frac{1}{\gamma_0 + \gamma_i} = \frac{\gamma_0^3 - \gamma_s^2\gamma_i - \gamma_0^2\gamma_s + \gamma_0\gamma_s^2}{\gamma_0^4 - \gamma_s^2\gamma_i^2}, \quad \text{when } s = 1.$$

**Case 2** (if $s = 2, 3, \ldots$). *In this case, notice*

$$\frac{\gamma_0^4 - \gamma_s^2\gamma_i^2}{\gamma_0^3 - \gamma_s^2\gamma_i - \gamma_0^2\gamma_s + \gamma_0\gamma_s^2} \leqslant \frac{\gamma_0^4}{\gamma_0(\gamma_0 - \gamma_s)(\gamma_0^2 + \gamma_s\gamma_i)} \leqslant \frac{\gamma_0^3}{(\gamma_0 - \gamma_2)(\gamma_0^2 + \gamma_2\gamma_3)} = \frac{1}{1 - 2^{-\zeta}},$$

Accordingly, we have

$$\lambda_j \leqslant \mathrm{Tr}(\Sigma_m) \leqslant \frac{1 - 2^{-\zeta}}{\gamma_0} \leqslant \frac{\gamma_0^3 - \gamma_s^2\gamma_i - \gamma_0^2\gamma_s + \gamma_0\gamma_s^2}{\gamma_0^4 - \gamma_s^2\gamma_i^2},$$

where the second inequality holds by Eq. (14). Accordingly, combining the above two cases, if we choose

$$\gamma_0 \leqslant \frac{1}{\frac{1}{1-2^{-\zeta}}\mathrm{Tr}(\Sigma_m)}, \quad \text{for } \zeta \in (0, 1),$$

we have $\mathrm{Tr}[D_t^{\mathtt{v}-\mathtt{x}}(0)] - \mathrm{Tr}[D_t^{\mathtt{v}-\mathtt{x}}(\zeta)] \geqslant 0$.

In the next, we give the upper bound for $D_t^{\mathtt{v}-\mathtt{x}}(0)$. The proof follows [63, Lemmas 3 and 5] and [16, Lemma B.4]. We just present it here for completeness. We firstly demonstrate that $D_t^{\mathtt{v}-\mathtt{x}}(0)$ is increasing and bounded, which implies that the limit $D_\infty^{\mathtt{v}-\mathtt{x}}(0)$ exists, and then we seek for the upper bound of this limit. To be specific, $D_t^{\mathtt{v}-\mathtt{x}}(0)$ admits the following expression

$$D_t^{\mathtt{v}-\mathtt{x}}(0) := \sum_{k=1}^{t} (I - \gamma_0 T^{\mathtt{W}})^{k-1} \circ \gamma_0^2 B \Sigma_m = D_{t-1}^{\mathtt{v}-\mathtt{x}}(0) + (I - \gamma_0 T^{\mathtt{W}})^{t-1} \circ \gamma_0^2 B \Sigma_m \succcurlyeq D_{t-1}^{\mathtt{v}-\mathtt{x}}(0),$$

which implies that $D_t^{\mathtt{v}-\mathtt{x}}(0)$ is increasing.

Let $A_t := (I - \gamma_0 T^{\mathtt{W}})^{t-1} \circ B\Sigma_m$, and then $A_t = (I - \gamma_0 T^{\mathtt{W}}) \circ A_{t-1}$. We have

$$
\begin{aligned}
\mathrm{Tr}(A_t) &= \mathrm{Tr}[(I - \gamma_0 T^{\mathtt{W}}) \circ A_{t-1}] = \mathrm{Tr}(A_{t-1}) - 2\gamma_0 \mathrm{Tr}(\Sigma_m A_{t-1}) + \gamma_0^2 \mathrm{Tr}(S^{\mathtt{W}} \circ A_{t-1}) \\
&\leqslant \mathrm{Tr}(A_{t-1}) - 2\gamma_0 \mathrm{Tr}(\Sigma_m A_{t-1}) + \gamma_0^2 r' \mathrm{Tr}(\Sigma_m A_{t-1})\mathrm{Tr}(\Sigma_m) \quad \text{[using Assumption 4]} \\
&\leqslant \mathrm{Tr}[(I - \gamma_0 \Sigma_m)A_{t-1}] \leqslant (1 - \gamma_0 \lambda_m)\mathrm{Tr}(A_{t-1}), \quad \text{[using } \gamma_0 \leqslant \tfrac{1}{r'\mathrm{Tr}(\Sigma_m)}]
\end{aligned}
$$

which implies

$$
\mathrm{Tr}[D_t^{\mathtt{v}-\mathtt{x}}(0)] \leqslant \gamma_0^2 \sum_{t=0}^{\infty} \mathrm{Tr}\left((I - \gamma_0 T^{\mathtt{W}})^t \circ B\Sigma_m\right) \leqslant \mathrm{Tr}(B\Sigma_m) \sum_{t=0}^{\infty}(1 - \gamma_0 \lambda_m)^t \leqslant \frac{\gamma_0 \mathrm{Tr}(B\Sigma_m)}{\lambda_m} < \infty \, .
$$

Accordingly, the monotonicity and boundedness of $\{D_t^{\mathtt{v}-\mathtt{x}}(0)\}_{t=0}^{\infty}$ implies that the limit exists, denoted as $D_\infty^{\mathtt{v}-\mathtt{x}}(0)$ with

$$
D_\infty^{\mathtt{v}-\mathtt{x}}(0) = (I - \gamma_0 T^{\mathtt{W}}) \circ D_\infty^{\mathtt{v}-\mathtt{x}}(0) + \gamma_0^2 B\Sigma_m \, ,
$$

which implies $D_\infty^{\mathtt{v}-\mathtt{x}}(0) = \gamma_0 (T^{\mathtt{W}})^{-1} \circ B\Sigma_m$ Further, we have

$$
\begin{aligned}
\widetilde{T}^{\mathtt{W}} \circ D_\infty^{\mathtt{v}-\mathtt{x}}(0) &= T^{\mathtt{W}} \circ D_\infty^{\mathtt{v}-\mathtt{x}}(0) + \gamma_0 S^{\mathtt{W}} \circ D_\infty^{\mathtt{v}-\mathtt{x}}(0) - \gamma_0 \widetilde{S}^{\mathtt{W}} \circ D_\infty^{\mathtt{v}-\mathtt{x}}(0) \quad \text{[definition of } \widetilde{T}^{\mathtt{W}}] \\
&= \gamma_0 B\Sigma_m + \gamma_0 S^{\mathtt{W}} \circ D_\infty^{\mathtt{v}-\mathtt{x}}(0) - \gamma_0 \widetilde{S}^{\mathtt{W}} \circ D_\infty^{\mathtt{v}-\mathtt{x}}(0) \quad\quad\quad (15) \\
&\preccurlyeq \gamma_0 B\Sigma_m + \gamma_0 S^{\mathtt{W}} \circ D_\infty^{\mathtt{v}-\mathtt{x}}(0) \, . \quad \text{[using } S^{\mathtt{W}} \succcurlyeq \widetilde{S}^{\mathtt{W}}]
\end{aligned}
$$

Besides, $(\widetilde{T}^{\mathtt{W}})^{-1} \circ \Sigma_m$ can be bounded by

$$
\begin{aligned}
(\widetilde{T}^{\mathtt{W}})^{-1} \circ \Sigma_m &= \gamma_0 \sum_{t=0}^{\infty}(I - \gamma_0 \widetilde{T}^{\mathtt{W}}) \circ \Sigma_m = \gamma_0 \sum_{t=0}^{\infty}(I - \gamma_0 \Sigma_m)^t \Sigma_m (I - \gamma_0 \Sigma_m)^t \\
&\preccurlyeq \gamma_0 \sum_{t=0}^{\infty}(I - \gamma_0 \Sigma_m)^t \Sigma_m = I \, . \quad \text{[using } \gamma_0 \leqslant 1/\mathrm{Tr}(\Sigma_m)]
\end{aligned}
\quad (16)
$$

Therefore, $D_\infty^{\mathtt{v}-\mathtt{x}}(0)$ can be further upper bounded by

$$
\begin{aligned}
D_\infty^{\mathtt{v}-\mathtt{x}}(0) &\preccurlyeq \gamma_0 (\widetilde{T}^{\mathtt{W}})^{-1} \circ B\Sigma_m + \gamma_0 (\widetilde{T}^{\mathtt{W}})^{-1} \circ S^{\mathtt{W}} \circ D_\infty^{\mathtt{v}-\mathtt{x}}(0) \quad \text{[using Eq. (15)]} \\
&\preccurlyeq \gamma_0 B + \gamma_0 (\widetilde{T}^{\mathtt{W}})^{-1} \circ S^{\mathtt{W}} \circ D_\infty^{\mathtt{v}-\mathtt{x}}(0) \quad \text{[using Eq. (16)]} \\
&= \gamma_0 B \sum_{t=0}^{\infty}[\gamma_0 (\widetilde{T}^{\mathtt{W}})^{-1} \circ S^{\mathtt{W}}]^t \circ I \quad \text{[solving the recursion]} \\
&\preccurlyeq \gamma_0 B \sum_{t=0}^{\infty}\left(\gamma_0 (\widetilde{T}^{\mathtt{W}})^{-1} \circ S^{\mathtt{W}}\right)^{t-1} \circ \gamma_0 (\widetilde{T}^{\mathtt{W}})^{-1} \circ S^{\mathtt{W}} \circ I \\
&\preccurlyeq \gamma_0 B \sum_{t=0}^{\infty}\left(\gamma_0 (\widetilde{T}^{\mathtt{W}})^{-1} \circ S^{\mathtt{W}}\right)^{t-1} \circ \gamma_0 (\widetilde{T}^{\mathtt{W}})^{-1} \circ \mathrm{Tr}(\Sigma_m)\Sigma_m \quad \text{[using Assumption 4]} \\
&\preccurlyeq \gamma_0 B \sum_{t=0}^{\infty}[\gamma_0 r' \mathrm{Tr}(\Sigma_m)]^t \circ I \quad \text{[using Eq. (16)]} \\
&\preccurlyeq \frac{\gamma_0 B}{1 - \gamma_0 r' \mathrm{Tr}(\Sigma_m)} I \, . \quad \text{[using } \gamma_0 < \tfrac{1}{r'\mathrm{tr}(\Sigma_m)}]
\end{aligned}
$$

$$(17)$$

Hence, based on the above results, $D_t^{\mathtt{v}-\mathtt{X}}(0)$ can be further upper bounded by

$$
\begin{aligned}
D_t^{\mathtt{v}-\mathtt{X}}(0) &= (I - \gamma_0 T^{\mathtt{W}}) \circ D_{t-1}^{\mathtt{v}-\mathtt{X}}(0) + \gamma_0^2 B \Sigma_m \\
&= (I - \gamma_0 \widetilde{T}^{\mathtt{W}}) \circ D_{t-1}^{\mathtt{v}-\mathtt{X}}(0) + \gamma_0^2 (S^{\mathtt{W}} - \widetilde{S}^{\mathtt{W}}) \circ D_{t-1}^{\mathtt{v}-\mathtt{X}} + \gamma_0^2 B \Sigma_m \\
&\preccurlyeq (I - \gamma_0 \widetilde{T}^{\mathtt{W}}) \circ D_{t-1}^{\mathtt{v}-\mathtt{X}}(0) + \gamma_0^2 S^{\mathtt{W}} \circ D_\infty^{\mathtt{v}-\mathtt{X}}(0) + \gamma_0^2 B \Sigma_m \\
&\preccurlyeq (I - \gamma_0 \widetilde{T}^{\mathtt{W}}) \circ D_{t-1}^{\mathtt{v}-\mathtt{X}}(0) + \gamma_0^2 r' \mathrm{Tr}[D_\infty^{\mathtt{v}-\mathtt{X}}(0)] \mathrm{Tr}(\Sigma_m) \Sigma_m + \gamma_0^2 B \Sigma_m \quad \text{[using Assumption 4]} \\
&\preccurlyeq (I - \gamma_0 \widetilde{T}^{\mathtt{W}}) \circ D_{t-1}^{\mathtt{v}-\mathtt{X}}(0) + \gamma_0^2 B \Sigma_m \left( \frac{\mathrm{Tr}(\Sigma_m) r' \gamma_0}{1 - \gamma_0 r' \mathrm{Tr}(\Sigma_m)} + 1 \right) \quad \text{[using Eq. (17)]} \\
&\preccurlyeq \gamma_0^2 B \left( \frac{\mathrm{Tr}(\Sigma_m) r' \gamma_0}{1 - \gamma_0 r' \mathrm{Tr}(\Sigma_m)} + 1 \right) \sum_{k=0}^\infty (I - \gamma_0 \Sigma_m)^k \Sigma_m \\
&\preccurlyeq \gamma_0 B \left( \frac{\mathrm{Tr}(\Sigma_m) r' \gamma_0}{1 - \gamma_0 r' \mathrm{Tr}(\Sigma_m)} + 1 \right) I,
\end{aligned}
\tag{18}
$$

which concludes the proof. $\qquad\square$

# E   Some useful integrals estimation

In this section, we present the estimation for the following integrals that will be needed in our proof by denoting $\kappa := 1 - \zeta \in (0, 1]$.

**Integral 1:** We consider the following integral admitting an exact estimation

$$
\int_1^t u^{-\zeta} \exp \left( -c \frac{u^{1-\zeta} - 1}{1 - \zeta} \right) \mathrm{d}u \leqslant t.
\tag{19}
$$

Besides, we also calculate this integral as below: by changing the integral variable $v^\kappa := c \frac{u^{1-\zeta} - 1}{1 - \zeta}$ and

$$
\frac{\mathrm{d}v}{\mathrm{d}u} = u^{1-\kappa} \left( \frac{\kappa}{c} \right)^{\frac{1}{\kappa}} (u^\kappa - 1)^{\frac{\kappa-1}{\kappa}} = \frac{1}{c} u^{1-\kappa} \kappa v^{\kappa-1},
$$

and then we have

$$
\begin{aligned}
\int_1^t u^{-\zeta} \exp \left( -c \frac{u^{1-\zeta} - 1}{1 - \zeta} \right) \mathrm{d}u &= \frac{1}{c} \int_0^{[\frac{c}{\kappa}(t^\kappa - 1)]^{\frac{1}{\kappa}}} u^{-\zeta} u^{1-\kappa} \kappa v^{\kappa-1} \exp(-v^\kappa) \mathrm{d}v \\
&\leqslant \frac{1}{c} \int_0^\infty \exp(-x) \mathrm{d}x = \left( \frac{1}{c} \wedge t \right),
\end{aligned}
\tag{20}
$$

where the last equality uses Eq. (19) and takes the smaller one via the notation $\wedge$. Accordingly, if we take $\zeta = 0$ in Eq. (20), we have

$$
\int_1^t \exp \left( -c \frac{u^{1-\zeta} - 1}{1 - \zeta} \right) \mathrm{d}u \leqslant \left( \frac{1}{c} t^\zeta \wedge t \right).
\tag{21}
$$

Similar to Eq. (21), we have

$$
\int_t^n \exp \left( -\widetilde{\lambda}_i \gamma_0 \frac{u^{1-\zeta} - t^{1-\zeta}}{1 - \zeta} \right) \mathrm{d}u \leqslant (n - t) \wedge \frac{n^\zeta}{\widetilde{\lambda}_i \gamma_0}.
\tag{22}
$$

**Integral 2:** we consider the following integral

$$\int_1^t u^{-\zeta} \exp\left(-c\frac{(t+1)^{1-\zeta}-(u+1)^{1-\zeta}}{1-\zeta}\right)du$$

$$= \frac{(t+1)^{1-\kappa}}{c}\int_0^C [(t+1)(1-x)^{\frac{1}{\kappa}}-1]^{\kappa-1}(1-x)^{\frac{1-\kappa}{\kappa}}\kappa v^{\kappa-1}\exp(-v^\kappa)dv \quad \text{with } x:=(\frac{v}{t+1})^\kappa\frac{\kappa}{c}$$

$$\leqslant \frac{2^\zeta}{c}\int_0^\infty \kappa v^{\kappa-1}\exp(-v^\kappa)dv$$

$$= \left(\frac{2^\zeta}{c}\wedge t\right),$$

$$(23)$$

where we change the integral variable $v^\kappa := c\frac{(t+1)^{1-\zeta}-(u+1)^{1-\zeta}}{1-\zeta}$ with $\kappa := 1-\zeta$ such that

$$du = -\frac{\kappa^{1/\kappa}}{c^{1/\kappa}}\left(\frac{u+1}{t+1}\right)^{1-\kappa}\left[1-\left(\frac{u+1}{t+1}\right)^\kappa\right]^{1-\frac{1}{\kappa}}dv = -\frac{\kappa}{c}\left[1-\left(\frac{v}{t+1}\right)^\kappa\frac{\kappa}{c}\right]^{\frac{1-\kappa}{\kappa}}\left(\frac{v}{t+1}\right)^{\kappa-1}dv,$$

with $(\frac{u+1}{t+1})^\kappa = 1-(v/(t+1))^\kappa\kappa/c$ and the upper limit of integral is $C := c^{1/\kappa}[(t+1)^\kappa-(u+1)^\kappa]^{1/\kappa}$.
Due to $u = (t+1)(1-x)^{\frac{1}{\kappa}}-1 \in [1,t]$, we have $(1-x)^{\frac{1}{\kappa}} \in [2/(t+1),1]$ and accordingly

$$g(x) := [(t+1)(1-x)^{\frac{1}{\kappa}}-1]^{\kappa-1}(1-x)^{\frac{1-\kappa}{\kappa}} \leqslant 2^{1-\kappa}(t+1)^{\kappa-1} \quad \text{with } x \in \left[0, 1-\left(\frac{2}{t+1}\right)^\kappa\right],$$

as an increasing function of $x$.

Similar to Eq. (23), we have the following estimation

$$\int_1^t \gamma_0^2 u^{-2\zeta}\exp\left(-2\widetilde{\lambda}_i\gamma_0\frac{(t+1)^{1-\zeta}-(u+1)^{1-\zeta}}{1-\zeta}\right)du \lesssim \left(\frac{\gamma_0}{\widetilde{\lambda}_i}\wedge\gamma_0^2 t\right). \quad (24)$$

## F    Proofs for `Bias`

In this section, we present the error bound for `Bias`. By virtue of Minkowski inequality, we have

$$\left(\mathbb{E}_{\boldsymbol{X},\boldsymbol{W}}\left[\langle\bar{\eta}_n^{\texttt{bias}},\Sigma_m\bar{\eta}_n^{\texttt{bias}}\rangle\right]\right)^{\frac{1}{2}} \leqslant \left(\underbrace{\mathbb{E}_{\boldsymbol{X},\boldsymbol{W}}\left[\langle\bar{\eta}_n^{\texttt{bias}}-\bar{\eta}_n^{\texttt{bX}},\Sigma_m(\bar{\eta}_n^{\texttt{bias}}-\bar{\eta}_n^{\texttt{bX}})\rangle\right]}_{\triangleq\texttt{B1}}\right)^{\frac{1}{2}} + \left(\mathbb{E}_{\boldsymbol{W}}\left[\langle\bar{\eta}_n^{\texttt{bX}},\Sigma_m\bar{\eta}_n^{\texttt{bX}}\rangle\right]\right)^{\frac{1}{2}}$$

$$\leqslant (\texttt{B1})^{\frac{1}{2}} + \left(\underbrace{\mathbb{E}_{\boldsymbol{W}}\left[\langle\bar{\eta}_n^{\texttt{bX}}-\bar{\eta}_n^{\texttt{bXW}},\Sigma_m(\bar{\eta}_n^{\texttt{bX}}-\bar{\eta}_n^{\texttt{bXW}})\rangle\right]}_{\triangleq\texttt{B2}}\right)^{\frac{1}{2}} + \underbrace{[\langle\bar{\eta}_n^{\texttt{bXW}},\widetilde{\Sigma}_m\bar{\eta}_n^{\texttt{bXW}}\rangle]^{\frac{1}{2}}}_{\triangleq\texttt{B3}}.$$

$$(25)$$

In the next, we give the error bounds for `B3`, `B2`, and `B1`, respectively.

### F.1    Bound for `B3`

In this section, we aim to bound `B3` := $\langle\bar{\eta}_n^{\texttt{bXW}},\widetilde{\Sigma}_m\bar{\eta}_n^{\texttt{bXW}}\rangle$.

**Proposition 2.** *Under Assumption 1, 2, 3, if the step-size $\gamma_t := \gamma_0 t^{-\zeta}$ with $\zeta \in [0,1)$ satisfies $\gamma_0 \leqslant \frac{1}{\text{Tr}(\widetilde{\Sigma}_m)}$, then* `B3` *can be bounded by*

$$\texttt{B3} \lesssim \frac{n^{\zeta-1}}{\gamma_0}\|f^*\|^2.$$

*Proof.* Due to $\gamma_0 \leqslant \frac{1}{\text{Tr}(\widetilde{\Sigma}_m)}$, the operator $I - \gamma_t\widetilde{\Sigma}_m$ is a contraction map for $t = 1, 2, \ldots, n$. Take spectral decomposition $\widetilde{\Sigma}_m = \widetilde{U}\widetilde{\Lambda}\widetilde{U}^\top$ where $\widetilde{U}$ is an orthogonal matrix and $\widetilde{\Lambda}$ is a diagonal matrix

with $(\widetilde{\Lambda})_{11} = \widetilde{\lambda}_1$ and $(\widetilde{\Lambda})_{ii} = \widetilde{\lambda}_2$ $(i = 2, 3, \ldots, m)$ as $\widetilde{\Sigma}_m$ has only two distinct eigenvalues in Lemma 1. Accordingly, we have

$$
\begin{aligned}
\langle \bar{\eta}_n^{\text{bXW}}, \widetilde{\Sigma}_m \bar{\eta}_n^{\text{bXW}} \rangle &= \frac{1}{n^2} \left\langle \sum_{t=0}^{n-1} \prod_{i=1}^{t} (I - \gamma_i \widetilde{\Sigma}_m) f^*, \widetilde{\Sigma}_m \sum_{t=0}^{n-1} \prod_{i=1}^{t} (I - \gamma_i \widetilde{\Sigma}_m) f^* \right\rangle \\
&= \frac{1}{n^2} \left\| \sum_{t=0}^{n-1} \prod_{i=1}^{t} (I - \gamma_i \widetilde{\Sigma}_m) \widetilde{\Sigma}_m^{\frac{1}{2}} f^* \right\|^2 \\
&\leqslant \frac{1}{n^2} \left\| \sum_{t=0}^{n-1} \prod_{i=1}^{t} (I - \gamma_i \widetilde{\Lambda}) \widetilde{\Lambda}^{\frac{1}{2}} \right\|_2^2 \|f^*\|^2 \quad [\text{using } \widetilde{\Sigma}_m = \widetilde{U}\widetilde{\Lambda}\widetilde{U}^\top] \\
&\leqslant \frac{1}{n} \max_{k=1,2} \sum_{t=0}^{n-1} \prod_{i=1}^{t} (1 - \gamma_i \widetilde{\lambda}_k)^2 \widetilde{\lambda}_k \|f^*\|^2 \\
&\leqslant \frac{1}{n} \sum_{t=0}^{n-1} \prod_{i=1}^{t} (1 - \gamma_i \widetilde{\lambda}_1)^2 \widetilde{\lambda}_1 \|f^*\|^2 + \frac{1}{n} \sum_{t=0}^{n-1} \prod_{i=1}^{t} (1 - \gamma_i \widetilde{\lambda}_2)^2 \widetilde{\lambda}_2 \|f^*\|^2 \,.
\end{aligned}
\tag{26}
$$

Note that

$$
\begin{aligned}
\sum_{t=0}^{n-1} \prod_{i=1}^{t} (1 - \gamma_i \widetilde{\lambda}_j)^2 &\leqslant \sum_{t=0}^{n-1} \exp\left(-2\gamma_0 \widetilde{\lambda}_j \sum_{i=1}^{t} i^{-\zeta}\right) \leqslant \sum_{t=0}^{n-1} \exp\left(-2\gamma_0 \widetilde{\lambda}_j \int_1^{t+1} \frac{1}{x^\zeta} \mathrm{d}x\right) \\
&= \sum_{t=0}^{n-1} \exp\left(-2\gamma_0 \widetilde{\lambda}_j \frac{(t+1)^{1-\zeta} - 1}{1 - \zeta}\right) \\
&\leqslant 1 + \int_0^n \exp\left(-2\gamma_0 \widetilde{\lambda}_j \frac{(t+1)^{1-\zeta} - 1}{1 - \zeta}\right) \mathrm{d}x \\
&\leqslant 1 + \left(\frac{n^\zeta}{2\gamma_0 \widetilde{\lambda}_j} \wedge n\right), \quad [\text{using Eq. (21)}]
\end{aligned}
\tag{27}
$$

here according to Lemma 1, for $\widetilde{\lambda}_1$, the upper bound $\frac{n^\zeta}{2\gamma_0 \widetilde{\lambda}_1}$ is tighter than $n$ due to $\widetilde{\lambda}_1 \sim \mathcal{O}(1)$; while this conclusion might not hold for $\widetilde{\lambda}_2$ due to $\widetilde{\lambda}_2 \sim \mathcal{O}(1/m)$. Then, taking Eq. (27) back to Eq. (26), we have

$$
\begin{aligned}
\langle \bar{\eta}_n^{\text{bXW}}, \widetilde{\Sigma}_m \bar{\eta}_n^{\text{bXW}} \rangle &\lesssim \frac{n^{\zeta-1}}{\gamma_0} \|f^*\|^2 + \frac{\widetilde{\lambda}_2}{n} \left(\frac{n^\zeta}{\gamma_0 \widetilde{\lambda}_2} \wedge n\right) \|f^*\|^2 \\
&\lesssim \frac{n^{\zeta-1}}{\gamma_0} \|f^*\|^2 \sim \mathcal{O}(n^{\zeta-1}) \,,
\end{aligned}
\tag{28}
$$

which concludes the proof. $\qquad\square$

### F.2 Bound for B2

Here we aim to bound $\text{B2} := \mathbb{E}_{\boldsymbol{W}}\left[\langle \bar{\eta}_n^{\text{bX}} - \bar{\eta}_n^{\text{bXW}}, \Sigma_m(\bar{\eta}_n^{\text{bX}} - \bar{\eta}_n^{\text{bXW}})\rangle\right] = \mathbb{E}_{\boldsymbol{W}}\left[\langle \bar{\alpha}_n^{\text{W}}, \widetilde{\Sigma}_m \bar{\alpha}_n^{\text{W}}\rangle\right] + \mathbb{E}_{\boldsymbol{W}}\left[\langle \bar{\alpha}_n^{\text{W}}, (\Sigma_m - \widetilde{\Sigma}_m) \bar{\alpha}_n^{\text{W}}\rangle\right]$, where

$$
\alpha_t^{\text{W}} := \eta_t^{\text{bX}} - \eta_t^{\text{bXW}} = (I - \gamma_t \Sigma_m)(\eta_{t-1}^{\text{bX}} - \eta_{t-1}^{\text{bXW}}) + \gamma_t (\widetilde{\Sigma}_m - \Sigma_m) \eta_{t-1}^{\text{bXW}} \,,
\tag{29}
$$

with $\alpha_0^{\text{W}} = 0$. Here $\alpha_t^{\text{W}}$ can be further formulated as

$$
\alpha_t^{\text{W}} = \sum_{k=1}^{t} \gamma_k \prod_{j=k+1}^{t} (I - \gamma_j \Sigma_m)(\widetilde{\Sigma}_m - \Sigma_m) \prod_{s=1}^{k-1} (I - \gamma_s \widetilde{\Sigma}_m) f^* \,,
\tag{30}
$$

where we use the recursion

$$
A_t := (I - \gamma_t \Sigma_m) A_{t-1} + B_t = \sum_{s=1}^{t} \prod_{i=s+1}^{t} (I - \gamma_i \Sigma_m) B_s \,.
$$

Accordingly, B2 admits

$$B2 = \mathbb{E}_{\boldsymbol{W}}\left[\langle \bar{\alpha}_n^{\mathsf{W}}, \Sigma_m \bar{\alpha}_n^{\mathsf{W}}\rangle\right] = \frac{1}{n^2}\mathbb{E}_{\boldsymbol{W}}\left\langle \sum_{t=0}^{n-1}\alpha_t^{\mathsf{W}}, \Sigma_m \sum_{t=0}^{n-1}\alpha_t^{\mathsf{W}}\right\rangle = \frac{1}{n^2}\mathbb{E}_{\boldsymbol{W}}\left\|\Sigma_m^{\frac{1}{2}}\sum_{t=0}^{n-1}\alpha_t^{\mathsf{W}}\right\|^2, \tag{31}$$

and we have the following error bound for B2.

**Proposition 3.** *Under Assumption 1, 2, 3, if the step-size $\gamma_t := \gamma_0 t^{-\zeta}$ with $\zeta \in [0,1)$ satisfies*

$$\gamma_0 \leqslant \min\left\{\frac{1}{\mathrm{Tr}(\Sigma_m)}, \frac{1}{\mathrm{Tr}(\widetilde{\Sigma}_m)}\right\},$$

*then B2 can be bounded by*

$$B2 \lesssim \frac{\|f^*\|^2}{\gamma_0}n^{\zeta-1}.$$

**Remark:** In our paper, we require $I - \gamma_t\Sigma_m$ $(t = 1, 2, \ldots m)$ to be a contraction map. Though $\mathrm{Tr}(\Sigma_m)$ is a random variable, it is with a sub-exponential $\mathcal{O}(1)$ norm, that means, the condition $\gamma_0 < 1/\mathrm{Tr}(\Sigma_m)$ can be equivalently substituted by $\gamma_0 < 1/[c\mathrm{Tr}(\widetilde{\Sigma}_m)]$ for some large $c$ (independent of $n$, $m$, $d$) with exponentially high probability. This is also used for estimating other quantities.

Before we present the error bounds for B2, we need the following lemma.

**Lemma 6.** *Under Assumption 1, 2, 3, if the step-size $\gamma_t := \gamma_0 t^{-\zeta}$ with $\zeta \in [0,1)$ satisfies*

$$\gamma_0 \leqslant \min\left\{\frac{1}{\mathrm{Tr}(\Sigma_m)}, \frac{1}{\mathrm{Tr}(\widetilde{\Sigma}_m)}\right\},$$

*denote $\Upsilon_i := \sum_{t=0}^{n-1}\sum_{k=1}^{t}\gamma_k(\widetilde{\lambda}_i - \lambda_i)\lambda_i^{\frac{1}{2}}\prod_{j=k+1}^{t}(1-\gamma_j\lambda_i)\prod_{s=1}^{k-1}(1-\gamma_s\widetilde{\lambda}_i), \forall i \in [m], we have*

$$\Upsilon_i \lesssim \lambda_i^{\frac{1}{2}}\left(\frac{n^\zeta}{\gamma_0\lambda_i} \wedge n\right), \quad \text{if } \lambda_i \neq 0; \quad \text{and } \Upsilon_i = 0, \quad \text{if } \lambda_i = 0.$$

*Proof.* Following the derivation in Appendix E, we consider the index $i$ with $\lambda_i \neq 0$ such that

$$\Upsilon_i := \sum_{t=0}^{n-1}\sum_{k=1}^{t}\gamma_k(\widetilde{\lambda}_i - \lambda_i)\lambda_i^{\frac{1}{2}}\prod_{j=k+1}^{t}(1-\gamma_j\lambda_i)\prod_{s=1}^{k-1}(1-\gamma_s\widetilde{\lambda}_i)$$

$$\leqslant \sum_{t=0}^{n-1}(\widetilde{\lambda}_i - \lambda_i)\lambda_i^{\frac{1}{2}}\sum_{k=1}^{t}\gamma_k \exp\left(-\sum_{j=k+1}^{t}\gamma_j\lambda_i\right)\exp\left(-\sum_{s=1}^{k-1}\gamma_s\widetilde{\lambda}_i\right)$$

$$\leqslant \sum_{t=0}^{n-1}(\widetilde{\lambda}_i - \lambda_i)\lambda_i^{\frac{1}{2}}\sum_{k=1}^{t}\gamma_k \exp\left(-\lambda_i\int_{k+1}^{t+1}\frac{\gamma_0}{x^\zeta}\mathrm{d}x\right)\exp\left(-\widetilde{\lambda}_i\int_{1}^{k}\frac{\gamma_0}{x^\zeta}\mathrm{d}x\right)$$

$$= \sum_{t=0}^{n-1}(\widetilde{\lambda}_i - \lambda_i)\lambda_i^{\frac{1}{2}}\sum_{k=1}^{t}\gamma_0 k^{-\zeta} \exp\left(-\lambda_i\gamma_0\frac{(t+1)^{1-\zeta}-(k+1)^{1-\zeta}}{1-\zeta}\right)\exp\left(-\widetilde{\lambda}_i\gamma_0\frac{k^{1-\zeta}-1}{1-\zeta}\right)$$

$$\lesssim \sum_{t=0}^{n-1}\gamma_0(\widetilde{\lambda}_i - \lambda_i)\lambda_i^{\frac{1}{2}}\left[\int_{1}^{t}u^{-\zeta}\exp\left(-\gamma_0\frac{\lambda_i(t+1)^{1-\zeta}-\lambda_iu^{1-\zeta}+\widetilde{\lambda}_iu^{1-\zeta}-\widetilde{\lambda}_i}{1-\zeta}\right)\mathrm{d}u\right.$$

$$\left.+ t^{-\zeta}\exp\left(-\widetilde{\lambda}_i\gamma_0\frac{t^{1-\zeta}-1}{1-\zeta}\right)\right],$$

$$\tag{32}$$

Denote $\kappa := 1 - \zeta$ and

$$v^\kappa := \gamma_0\frac{\lambda_i(t+1)^{1-\zeta}-\lambda_iu^{1-\zeta}+\widetilde{\lambda}_iu^{1-\zeta}-\widetilde{\lambda}_i}{1-\zeta},$$

by changing the integral variable $u$ to $v$, we have

$$\frac{\mathrm{d}u}{\mathrm{d}v} = \frac{u^{1-\kappa}}{\widetilde{\lambda}_i - \lambda_i} \left(\frac{\gamma_0}{\kappa}\right)^{-1/\kappa} [(\widetilde{\lambda}_i - \lambda_i)u^\kappa + \lambda_i(t+1)^\kappa - \widetilde{\lambda}_i] = \frac{u^{1-\kappa}}{\widetilde{\lambda}_i - \lambda_i}\frac{\kappa}{\gamma_0}v^{\kappa-1},$$

Accordingly, Eq. (32) can be upper bounded by

$$\Upsilon_i \lesssim \sum_{t=0}^{n-1} \gamma_0(\widetilde{\lambda}_i - \lambda_i)\lambda_i^{\frac{1}{2}}\left[\int_1^t u^{-\zeta}\exp\left(-\gamma_0\frac{\lambda_i(t+1)^{1-\zeta} - \lambda_i u^{1-\zeta} + \widetilde{\lambda}_i u^{1-\zeta} - \widetilde{\lambda}_i}{1-\zeta}\right)\mathrm{d}u\right.$$

$$\left. + t^{-\zeta}\exp\left(-\widetilde{\lambda}_i\gamma_0\frac{t^{1-\zeta} - 1}{1-\zeta}\right)\right]$$

$$\leqslant \sum_{t=0}^{n-1} \gamma_0(\widetilde{\lambda}_i - \lambda_i)\lambda_i^{\frac{1}{2}}\left[\int_{c_1^{\frac{1}{\kappa}}}^{c_2^{\frac{1}{\kappa}}} u^{-\zeta}\exp(-v^\kappa)\frac{1}{\widetilde{\lambda}_i - \lambda_i}u^{1-\kappa}\frac{\kappa}{\gamma_0}v^{\kappa-1}\mathrm{d}v + t^{-\zeta}\exp\left(-\widetilde{\lambda}_i\gamma_0\frac{t^{1-\zeta} - 1}{1-\zeta}\right)\right]$$

$$= \sum_{t=0}^{n-1}\left[\lambda_i^{\frac{1}{2}}\int_{c_1}^{c_2}\exp(-x)\mathrm{d}x + \gamma_0(\widetilde{\lambda}_i - \lambda_i)\lambda_i^{\frac{1}{2}}t^{-\zeta}\exp\left(-\widetilde{\lambda}_i\gamma_0\frac{t^{1-\zeta} - 1}{1-\zeta}\right)\right]$$

$$\lesssim \lambda_i^{\frac{1}{2}}\int_0^n \exp\left(-\lambda_i\gamma_0\frac{(u+1)^{1-\zeta} - 1}{1-\zeta}\right)\mathrm{d}u$$

$$\leqslant \lambda_i^{\frac{1}{2}}\left(\frac{n^\zeta}{\gamma_0\lambda_i}\wedge n\right), \quad \text{[using Eq. (22)]}$$

where $c_1 := \frac{\gamma_0}{\kappa}\lambda_i[(t+1)^\kappa - 1]$ and $c_2 := \frac{\gamma_0}{\kappa}\widetilde{\lambda}_i[t^\kappa - 1]$. Finally, we conclude the proof. $\qquad\square$

In the next, we are ready to present the error bounds for B2.

*Proof of Proposition 3.* According to Eq. (31), we need estimation for $\left\|\Sigma_m^{\frac{1}{2}}\sum_{t=0}^{n-1}\alpha_t^{\mathtt{W}}\right\|_2$ for estimating B2. By spectrum decomposition, we have $\prod_{j=k+1}^t(I - \gamma_j\Sigma_m) = U\left(\prod_{j=k+1}^t(I - \gamma_j\Lambda)\right)U^\top$ and $\prod_{s=1}^{k-1}(I - \gamma_s\widetilde{\Sigma}_m) = \widetilde{U}\prod_{s=1}^{k-1}(I - \gamma_s\widetilde{\Lambda})\widetilde{U}^\top$. Then we have

$$\left\|\Sigma_m^{1/2}\sum_{t=0}^{n-1}\alpha_t^{\mathtt{W}}\right\|_2 = \left\|\sum_{t=0}^{n-1}\sum_{k=1}^t\gamma_k\prod_{j=k+1}^t(I - \gamma_j\Sigma_m)(\widetilde{\Sigma}_m - \Sigma_m)\prod_{s=1}^{k-1}(I - \gamma_s\widetilde{\Sigma}_m)\Sigma_m^{\frac{1}{2}}f^*\right\|_2$$

$$= \left\|\sum_{t=0}^{n-1}\sum_{k=1}^t\gamma_k\prod_{j=k+1}^t(I - \gamma_j\Lambda_m)(\widetilde{\Lambda}_m - \Lambda_m)\prod_{s=1}^{k-1}(I - \gamma_s\widetilde{\Lambda}_m)\Lambda_m^{\frac{1}{2}}f^*\right\|_2$$

$$\leqslant \max_{i\in\{1,2,\ldots,m\}}\sum_{t=0}^{n-1}\sum_{k=1}^t\gamma_k(\widetilde{\lambda}_i - \lambda_i)\lambda_i^{\frac{1}{2}}\prod_{j=k+1}^t(1 - \gamma_j\lambda_i)\prod_{s=1}^{k-1}(1 - \gamma_s\widetilde{\lambda}_i)\|f^*\|,$$

(33)

where the second equality holds by $\|AB\|_2 = \|BA\|_2$ for any two PSD matrices.

By Lemma 6, we have

$$\text{B2} = \frac{1}{n^2}\mathbb{E}_{\boldsymbol{W}}\left\|\Sigma_m^{\frac{1}{2}}\sum_{t=0}^{n-1}\alpha_t^{\mathtt{W}}\right\|^2 = \frac{1}{n^2}\mathbb{E}_{\boldsymbol{W}}\left\|\max_{i\in\{1,2,\ldots,m\}}\Upsilon_i\right\|^2 \lesssim \frac{1}{n^2}\mathbb{E}_{\boldsymbol{W}}\left[\lambda_i^{\frac{1}{2}}\left(\frac{n^\zeta}{\gamma_0\lambda_i}\wedge n\right)\right]^2\|f^*\|^2$$

$$:= \|f^*\|^2\mathbb{E}_{\boldsymbol{W}}\left[\frac{n^{2(1-\zeta)}}{\gamma_0^2\lambda_{i^*}}\wedge\lambda_{i^*}\right] = \|f^*\|^2\begin{cases}\mathbb{E}_{\boldsymbol{W}}\left[\frac{n^{2(1-\zeta)}}{\gamma_0^2\lambda_{i^*}}\right], & \text{if } \lambda_{i^*}\geqslant\frac{n^{\zeta-1}}{\gamma_0} \\ \mathbb{E}_{\boldsymbol{W}}[\lambda_{i^*}], & \text{if } \lambda_{i^*}\leqslant\frac{n^{\zeta-1}}{\gamma_0}.\end{cases}$$

$$\lesssim \frac{\|f^*\|^2}{\gamma_0}n^{\zeta-1}.$$

(34)

$\square$

### F.3  Bound for B1

Here we aim to bound $\texttt{B1} := \mathbb{E}_{\boldsymbol{X},\boldsymbol{W}}\left[\langle \bar{\eta}_n^{\texttt{bias}} - \bar{\eta}_n^{\texttt{bX}}, \Sigma_m(\bar{\eta}_n^{\texttt{bias}} - \bar{\eta}_n^{\texttt{bX}})\rangle\right]$. Define $\alpha_t^{\texttt{X}} := \eta_t^{\texttt{bias}} - \eta_t^{\texttt{bX}}$, we have

$$\alpha_t^{\texttt{X}} = [I - \gamma_t\varphi(\boldsymbol{x}_t)\otimes\varphi(\boldsymbol{x}_t)]\alpha_{t-1}^{\texttt{X}} + \gamma_t[\Sigma_m - \varphi(\boldsymbol{x}_t)\otimes\varphi(\boldsymbol{x}_t)]\eta_{t-1}^{\texttt{bX}}\,, \tag{35}$$

with $\alpha_0^{\texttt{X}} = 0$ and $\eta_{t-1}^{\texttt{bX}} = \prod_{j=1}^{t-1}(I - \gamma_j\Sigma_m)f^*$. Accordingly, we have

$$\texttt{B1} := \mathbb{E}_{\boldsymbol{X},\boldsymbol{W}}\left[\langle \bar{\eta}_n^{\texttt{bias}} - \bar{\eta}_n^{\texttt{bX}}, \Sigma_m(\bar{\eta}_n^{\texttt{bias}} - \bar{\eta}_n^{\texttt{bX}})\rangle\right] = \mathbb{E}_{\boldsymbol{W}}\left(\mathbb{E}_{\boldsymbol{X}}[\langle \bar{\alpha}_n^{\texttt{X}}, \Sigma_m\bar{\alpha}_n^{\texttt{X}}\rangle]\right)\,.$$

**Proposition 4.** *Under Assumption 1, 2, 3, 4 with $r' \geqslant 1$, if the step-size $\gamma_t := \gamma_0 t^{-\zeta}$ with $\zeta \in [0, 1)$ satisfies*

$$\gamma_0 < \min\left\{\frac{1}{r'\mathrm{Tr}(\Sigma_m)}, \frac{1}{c'\mathrm{Tr}(\Sigma_m)}\right\}\,,$$

*where the constant $c'$ is defined in Eq. (14). Then* $\texttt{B1}$ *can be bounded by*

$$\texttt{B1} \lesssim \gamma_0 r' n^{\zeta-1}\|f^*\|^2 \sim \mathcal{O}\left(n^{\zeta-1}\right)\,.$$

To prove Proposition 4, we need a lemma on stochastic recursions based on $\mathbb{E}[\alpha_t^{\texttt{X}}|\alpha_{t-1}^{\texttt{X}}] = (I - \gamma_t\Sigma_m)\alpha_{t-1}^{\texttt{X}}$, that shares the similar proof fashion with [29, Lemma 1] and [31, Lemma 11].

**Lemma 7.** *Under Assumption 1, 2, 3, 4 with $r' \geqslant 1$, denoting $H_{t-1} := [\Sigma_m - \varphi(\boldsymbol{x}_t)\otimes\varphi(\boldsymbol{x}_t)]\eta_{t-1}^{\texttt{bX}}$, if the step-size $\gamma_t := \gamma_0 t^{-\zeta}$ with $\zeta \in [0, 1)$ satisfies*

$$\gamma_0 < \frac{1}{r'\mathrm{Tr}(\Sigma_m)}\,,$$

*we have*

$$\mathbb{E}_{\boldsymbol{X}}[\langle\bar{\alpha}_n^{\texttt{X}}, \Sigma_m\bar{\alpha}_n^{\texttt{X}}\rangle] \leqslant \frac{1}{2n[1 - \gamma_0 r'\mathrm{Tr}(\Sigma_m)]}\left(\sum_{k=1}^{n-1}\mathbb{E}\|\alpha_k^{\texttt{X}}\|^2(\frac{1}{\gamma_{k+1}} - \frac{1}{\gamma_k}) + 2\sum_{t=0}^{n-1}\gamma_{t+1}\mathbb{E}_{\boldsymbol{X}}\|H_t\|^2\right)\,.$$

**Remark:** We require $\|\Sigma_m\|_2 \neq \frac{1}{r'\gamma_0}$ to avoid the denominator to be zero, which naturally holds as the probability measure of the continuous random variable $\|\Sigma_m\|_2$ at a point is zero.

*Proof.* According to the definition of $\alpha_t^{\texttt{X}}$ in Eq. (35), define $H_{t-1} := [\Sigma_m - \varphi(\boldsymbol{x}_t)\otimes\varphi(\boldsymbol{x}_t)]\eta_{t-1}^{\texttt{bX}}$, we have

$$\begin{aligned}
\|\alpha_t^{\texttt{X}}\|^2 &= \|\alpha_{t-1}^{\texttt{X}} - \gamma_t([\varphi(\boldsymbol{x}_t)\otimes\varphi(\boldsymbol{x}_t)]\alpha_{t-1}^{\texttt{W}} - H_{t-1})\|^2 \\
&= \|\alpha_{t-1}^{\texttt{X}}\|^2 + \gamma_t^2\|H_{t-1} - [\varphi(\boldsymbol{x}_t)\otimes\varphi(\boldsymbol{x}_t)]\alpha_{t-1}^{\texttt{X}}\|^2 + 2\gamma_t\langle\alpha_{t-1}^{\texttt{W}}, H_{t-1} - [\varphi(\boldsymbol{x}_t)\otimes\varphi(\boldsymbol{x}_t)]\alpha_{t-1}^{\texttt{X}}\rangle \\
&\leqslant \|\alpha_{t-1}^{\texttt{X}}\|^2 + 2\gamma_t^2\left(\|H_{t-1}\|^2 + \|[\varphi(\boldsymbol{x}_t)\otimes\varphi(\boldsymbol{x}_t)]\alpha_{t-1}^{\texttt{X}}\|^2\right) + 2\gamma_t\langle\alpha_{t-1}^{\texttt{X}}, H_{t-1} - [\varphi(\boldsymbol{x}_t)\otimes\varphi(\boldsymbol{x}_t)]\alpha_{t-1}^{\texttt{X}}\rangle\,,
\end{aligned}$$

which implies (by taking the conditional expectation)

$$\begin{aligned}
\mathbb{E}_{\boldsymbol{X}}[\|\alpha_t^{\texttt{W}}\|^2|\alpha_{t-1}^{\texttt{W}}] &\leqslant \|\alpha_{t-1}^{\texttt{X}}\|^2 + 2\gamma_t^2\|H_{t-1}\|^2 + 2\gamma_t^2\langle\alpha_{t-1}^{\texttt{X}}, \mathbb{E}_{\boldsymbol{X}}[\varphi(\boldsymbol{x}_t)\otimes\varphi(\boldsymbol{x}_t)\otimes\varphi(\boldsymbol{x}_t)\otimes\varphi(\boldsymbol{x}_t)]\alpha_{t-1}^{\texttt{X}}\rangle \\
&\quad - 2\gamma_t\langle\alpha_{t-1}^{\texttt{X}}, \Sigma_m\alpha_{t-1}^{\texttt{X}}\rangle \\
&\leqslant \|\alpha_{t-1}^{\texttt{X}}\|^2 + 2\gamma_t^2\|H_{t-1}\|^2 + 2\gamma_t^2 r'\mathrm{Tr}(\Sigma_m)\langle\alpha_{t-1}^{\texttt{X}}, \Sigma_m\alpha_{t-1}^{\texttt{X}}\rangle - 2\gamma_t\langle\alpha_{t-1}^{\texttt{X}}, \Sigma_m\alpha_{t-1}^{\texttt{X}}\rangle \\
&= \|\alpha_{t-1}^{\texttt{X}}\|^2 + 2\gamma_t^2\|H_{t-1}\|^2 - 2\gamma_t[1 - \gamma_t r'\mathrm{Tr}(\Sigma_m)]\langle\alpha_{t-1}^{\texttt{X}}, \Sigma_m\alpha_{t-1}^{\texttt{X}}\rangle\,.
\end{aligned} \tag{36}$$

where the first inequality holds by $\mathbb{E}_{\boldsymbol{X}}[H_{t-1}] = 0$, and the second inequality satisfies by Assumption 4.

By taking the expectation of Eq. (36), we have

$$\mathbb{E}_{\boldsymbol{X}}[\|\alpha_t^{\texttt{X}}\|^2] \leqslant \mathbb{E}_{\boldsymbol{X}}[\|\alpha_{t-1}^{\texttt{X}}\|^2] + 2\gamma_t^2\mathbb{E}_{\boldsymbol{X}}[\|H_{t-1}\|^2] - 2\gamma_t[1 - \gamma_t r'\mathrm{Tr}(\Sigma_m)]\mathbb{E}_{\boldsymbol{X}}\langle\alpha_{t-1}^{\texttt{X}}, \Sigma_m\alpha_{t-1}^{\texttt{X}}\rangle\,,$$

which indicates that

$$\mathbb{E}_{\boldsymbol{X}}\left[\langle\bar{\alpha}_n^{\mathtt{X}}, \Sigma_m \bar{\alpha}_n^{\mathtt{X}}\rangle\rangle\right] \leqslant \frac{1}{n}\sum_{t=0}^{n-1}\mathbb{E}_{\boldsymbol{X}}\langle\alpha_t^{\mathtt{W}}, \Sigma_m \alpha_t^{\mathtt{W}}\rangle \leqslant \frac{1}{2n[1-\gamma_0 r'\mathrm{Tr}(\Sigma_m)]}\left(\sum_{k=1}^{n-1}\mathbb{E}_{\boldsymbol{X}}\|\alpha_k^{\mathtt{X}}\|^2(\frac{1}{\gamma_{k+1}}-\frac{1}{\gamma_k})\right.$$

$$+ \frac{1}{2\gamma_1}\mathbb{E}_{\boldsymbol{X}}\|\alpha_0^{\mathtt{X}}\|^2 - \frac{1}{2\gamma_t}\mathbb{E}_{\boldsymbol{X}}\|\alpha_t^{\mathtt{X}}\|^2 + \sum_{t=0}^{n-1}\gamma_{t+1}\mathbb{E}_{\boldsymbol{X}}\|H_t\|^2\bigg)$$

$$\leqslant \frac{1}{2n[1-\gamma_0 r'\mathrm{Tr}(\Sigma_m)]}\left(\sum_{k=1}^{n-1}\mathbb{E}_{\boldsymbol{X}}\|\alpha_k^{\mathtt{X}}\|^2(\frac{1}{\gamma_{k+1}}-\frac{1}{\gamma_k}) + 2\sum_{t=0}^{n-1}\gamma_{t+1}\mathbb{E}_{\boldsymbol{X}}\|H_t\|^2\right),$$

due to $\alpha_0^{\mathtt{W}} = 0$. $\qquad\square$

In the next, we present the error bounds for two respective terms in Lemma 7.

**Lemma 8.** *Based on the definition of $\alpha_t^{\mathtt{X}}$ in Eq. (37), under Assumption 1, 2, 3, 4 with $r' \geqslant 1$, if the step-size $\gamma_t := \gamma_0 t^{-\varsigma}$ with $\varsigma \in [0, 1)$ satisfies*

$$\gamma_0 < \min\left\{\frac{1}{r'\mathrm{Tr}(\Sigma_m)}, \frac{1}{c'\mathrm{Tr}(\Sigma_m)}\right\},$$

*where the constant $c'$ is defined in Eq. (14). Then, we have*

$$\sum_{k=1}^{n-1}\mathbb{E}\|\alpha_k^{\mathtt{X}}\|^2(\frac{1}{\gamma_{k+1}}-\frac{1}{\gamma_k}) \lesssim \frac{\gamma_0 r'\mathrm{Tr}(\Sigma_m)}{1-\gamma_0 r'\mathrm{Tr}(\Sigma_m)}(n^\varsigma - 1)\|f^*\|^2.$$

*Proof.* Based on the definition of $\alpha_t^{\mathtt{X}}$ in Eq. (35), it can be reformulated as

$$\alpha_t^{\mathtt{X}} = [I - \gamma_t\varphi(\boldsymbol{x}_t)\otimes\varphi(\boldsymbol{x}_t)]\alpha_{t-1}^{\mathtt{X}} + \gamma_t[\Sigma_m - \varphi(\boldsymbol{x}_t)\otimes\varphi(\boldsymbol{x}_t)]\prod_{j=1}^{k-1}(I-\gamma_j\Sigma_m)f^*$$

$$= \sum_{s=1}^t \gamma_s \prod_{i=s+1}^t [I-\gamma_i\varphi(\boldsymbol{x}_i)\otimes\varphi(\boldsymbol{x}_i)][\Sigma_m - \varphi(\boldsymbol{x}_s)\otimes\varphi(\boldsymbol{x}_s)]\prod_{j=1}^{s-1}(I-\gamma_j\Sigma_m)f^*.$$

$$(37)$$

and accordingly

$$C_t^{\mathtt{b-X}} := \mathbb{E}_{\boldsymbol{X}}[\alpha_t^{\mathtt{X}}\otimes\alpha_t^{\mathtt{X}}] = (I-\gamma_t T^{\mathtt{W}})\circ C_{t-1}^{\mathtt{b-X}} + \gamma_t^2(S^{\mathtt{W}} - \widetilde{S}^{\mathtt{W}})\circ[\eta_{t-1}^{\mathtt{bX}}\otimes\eta_{t-1}^{\mathtt{bX}}]$$

$$\preccurlyeq (I-\gamma_t T^{\mathtt{W}})\circ C_{t-1}^{\mathtt{b-X}} + \gamma_t^2 S^{\mathtt{W}}\circ[\eta_{t-1}^{\mathtt{bX}}\otimes\eta_{t-1}^{\mathtt{bX}}]$$

$$\preccurlyeq (I-\gamma_t T^{\mathtt{W}})\circ C_{t-1}^{\mathtt{b-X}} + \gamma_t^2 r'\mathrm{Tr}\left[\prod_{s=1}^{t-1}(I-\gamma_s\Sigma_m)^2\Sigma_m\right]\Sigma_m(f^*\otimes f^*) \quad [\text{using Assumption 4}]$$

$$\preccurlyeq (I-\gamma_t T^{\mathtt{W}})\circ C_{t-1}^{\mathtt{b-X}} + \gamma_t^2 r'\mathrm{Tr}(\Sigma_m)\Sigma_m(f^*\otimes f^*) \quad [\text{using } \exp(-2\lambda_i\gamma_0\frac{t^{1-\varsigma}-1}{1-\varsigma})\leqslant 1]$$

$$= r'\mathrm{Tr}(\Sigma_m)\sum_{s=1}^t\prod_{i=s+1}^t(I-\gamma_i T^{\mathtt{W}})\circ\gamma_s^2\Sigma_m(f^*\otimes f^*)$$

$$\preccurlyeq \frac{\gamma_0 r'\mathrm{Tr}(\Sigma_m)}{1-\gamma_0 r'\mathrm{Tr}(\Sigma_m)}(f^*\otimes f^*). \quad [\text{using Lemma 5}]$$

$$(38)$$

Accordingly, we have

$$\sum_{t=1}^{n-1}\mathbb{E}_{\boldsymbol{X}}\|\alpha_t^{\mathtt{X}}\|^2(\frac{1}{\gamma_{t+1}}-\frac{1}{\gamma_t}) = \sum_{t=1}^{n-1}\|C_t^{\mathtt{b-X}}\|_2\left(\frac{1}{\gamma_{t+1}}-\frac{1}{\gamma_t}\right) \quad [\text{using Eq. (38)}]$$

$$\leqslant \sum_{t=1}^{n-1}\frac{\gamma_0 r'\mathrm{Tr}(\Sigma_m)}{1-\gamma_0 r'\mathrm{Tr}(\Sigma_m)}[(t+1)^\varsigma - t^\varsigma]\|f^*\|^2$$

$$\lesssim \frac{\gamma_0 r'\mathrm{Tr}(\Sigma_m)}{1-\gamma_0 r'\mathrm{Tr}(\Sigma_m)}(n^\varsigma - 1)\|f^*\|^2,$$

which concludes the proof. $\qquad\square$

**Lemma 9.** *Denote* $H_{t-1} := [\Sigma_m - \varphi(\boldsymbol{x}_t) \otimes \varphi(\boldsymbol{x}_t)]\eta_{t-1}^{\mathtt{bX}}$, *Assumption 1, 2, 3, 4 with* $r' \geqslant 1$, *if the step-size* $\gamma_t := \gamma_0 t^{-\zeta}$ *with* $\zeta \in [0, 1)$ *satisfies*

$$\gamma_0 \leqslant \frac{1}{\mathrm{Tr}(\Sigma_m)} \,,$$

*we have*

$$\sum_{t=0}^{n-1} \gamma_{t+1} \mathbb{E}_{\boldsymbol{X}} \|H_t\|^2 \leqslant \frac{1}{2} \|f^*\|^2 r' \mathrm{Tr}(\Sigma_m) \,.$$

*Proof.*

$$
\begin{aligned}
\sum_{t=0}^{n-1} \gamma_{t+1} \mathbb{E}_{\boldsymbol{X}} \|H_t\|^2 &= \sum_{t=0}^{n-1} \gamma_{t+1} \left\langle f^*, \prod_{j=1}^{t-1}(I - \gamma_j \Sigma_m) \mathbb{E}_{\boldsymbol{X}}[\Sigma_m - \varphi(\boldsymbol{x}_t) \otimes \varphi(\boldsymbol{x}_t)]^2 \prod_{j=1}^{t-1}(I - \gamma_j \Sigma_m) f^* \right\rangle \\
&\leqslant \sum_{t=0}^{n-1} \gamma_{t+1} \left\langle f^*, r' \mathrm{Tr}(\Sigma_m) \Big[ \prod_{j=1}^{t-1}(I - \gamma_j \Sigma_m) \Big]^2 \Sigma_m f^* \right\rangle \quad \text{[using Assumption 4]} \\
&\leqslant \|f^*\|^2 r' \mathrm{Tr}(\Sigma_m) \left\| \sum_{t=0}^{n-1} \gamma_{t+1} \Big[ \prod_{j=1}^{t-1}(I - \gamma_j \Sigma_m) \Big]^2 \Sigma_m \right\|_2 \\
&= \|f^*\|^2 r' \mathrm{Tr}(\Sigma_m) \max_{i \in \{1,2,\dots,m\}} \sum_{t=0}^{n-1} \gamma_{t+1} \prod_{j=1}^{t-1}(1 - \gamma_j \lambda_i)^2 \lambda_i \\
&\leqslant \|f^*\|^2 r' \mathrm{Tr}(\Sigma_m) \max_{i \in \{1,2,\dots,m\}} \gamma_0 \lambda_i \int_0^n u^{-\zeta} \exp\left( -2\gamma_0 \lambda_i \frac{u^{1-\zeta} - 1}{1 - \zeta} \right) \mathrm{d}u \\
&\leqslant \frac{1}{2} \|f^*\|^2 r' \mathrm{Tr}(\Sigma_m) \,, \quad \text{[using Eq. (20)]}
\end{aligned}
$$

which concludes the proof. $\qquad\square$

Based on the above results, we are ready to prove Proposition 4.

*Proof.* According to Lemma 8, we have

$$
\begin{aligned}
\mathbb{E}_{\boldsymbol{W}} \frac{\sum_{k=1}^{n-1} \mathbb{E}\|\alpha_k^{\mathtt{X}}\|^2 \left( \frac{1}{\gamma_{k+1}} - \frac{1}{\gamma_k} \right)}{2n[1 - \gamma_0 r' \mathrm{Tr}(\Sigma_m)]} &\lesssim \mathbb{E}_{\boldsymbol{W}} \frac{\gamma_0 r' \mathrm{Tr}(\Sigma_m)}{2n[1 - \gamma_0 r' \mathrm{Tr}(\Sigma_m)]^2} (n^\zeta - 1) \|f^*\|^2 \\
&\lesssim \gamma_0 r' n^{\zeta - 1} \|f^*\|^2 \\
&\sim \mathcal{O}(n^{\zeta - 1}) \,,
\end{aligned}
$$

where we use the condition on the step-size regarding $\gamma_0$.

According to Lemma 9, we have

$$
\begin{aligned}
\mathbb{E}_{\boldsymbol{W}} \frac{2 \sum_{t=0}^{n-1} \gamma_{t+1} \mathbb{E}_{\boldsymbol{X}} \|H_t\|^2}{2n[1 - \gamma_0 r' \mathrm{Tr}(\Sigma_m)]} &\leqslant \mathbb{E}_{\boldsymbol{W}} \frac{r' \mathrm{Tr}(\Sigma_m)}{2n[1 - \gamma_0 r' \mathrm{Tr}(\Sigma_m)]} \|f^*\|^2 \\
&\lesssim \frac{r'}{n} \|f^*\|^2 \\
&\sim \mathcal{O}\left( \frac{1}{n} \right) \,.
\end{aligned}
$$

Accordingly, combining the above two equations, we have

$$
\begin{aligned}
\mathtt{B1} := \mathbb{E}_{\boldsymbol{W}} \mathbb{E}_{\boldsymbol{X}} [\langle \bar{\alpha}_n^{\mathtt{X}}, \Sigma_m \bar{\alpha}_n^{\mathtt{X}} \rangle] &\leqslant \frac{1}{2n[1 - \gamma_0 r' \mathrm{Tr}(\Sigma_m)]} \mathbb{E}_{\boldsymbol{W}} \left( \sum_{k=1}^{n-1} \mathbb{E}\|\alpha_k^{\mathtt{X}}\|^2 \left( \frac{1}{\gamma_{k+1}} - \frac{1}{\gamma_k} \right) + 2 \sum_{t=0}^{n-1} \gamma_{t+1} \mathbb{E}_{\boldsymbol{X}} \|H_t\|^2 \right) \\
&\lesssim \gamma_0 r' n^{\zeta - 1} \|f^*\|^2 \,,
\end{aligned}
$$

which concludes the proof. $\qquad\square$

### F.4 Proof of Theorem 1

*Proof.* Combining the above results for three terms B1, B2, B3, if

$$\gamma_0 < \min \left\{ \frac{1}{\mathrm{Tr}(\widetilde{\Sigma}_m)}, \frac{1}{r'\mathrm{Tr}(\Sigma_m)}, \frac{1}{c'\mathrm{Tr}(\Sigma_m)} \right\} \sim \mathcal{O}(1), \tag{39}$$

where the constant $c$ is defined in Eq. (14). Then the Bias can be upper bounded by

$$\mathtt{Bias} \leqslant \left( \sqrt{\mathtt{B1}} + \sqrt{\mathtt{B2}} + \sqrt{\mathtt{B3}} \right)^2 \leqslant 3(\mathtt{B1} + \mathtt{B2} + \mathtt{B3})$$
$$\lesssim \gamma_0 r' n^{\zeta-1} \|f^*\|^2 \,.$$

$\square$

## G  Proof for Variance

In this section, we present the error bound for Variance. Recall the definition of $\eta_t^{\mathtt{vX}}$ in Eq. (8) and $\eta_t^{\mathtt{vXW}}$ in Eq. (9), and

$$\bar{\eta}_n^{\mathtt{vX}} := \frac{1}{n} \sum_{t=0}^{n-1} \bar{\eta}_t^{\mathtt{vX}}, \qquad \bar{\eta}_n^{\mathtt{vXW}} := \frac{1}{n} \sum_{t=0}^{n-1} \bar{\eta}_t^{\mathtt{vXW}},$$

by virtue of Minkowski inequality, Variance can be further decomposed as

$$\left( \mathbb{E}_{\boldsymbol{X},\boldsymbol{W},\boldsymbol{\varepsilon}} \left[ \langle \bar{\eta}_n^{\mathtt{var}}, \Sigma_m \bar{\eta}_n^{\mathtt{var}} \rangle \right] \right)^{\frac{1}{2}} \leqslant \left( \underbrace{\mathbb{E}_{\boldsymbol{X},\boldsymbol{W},\boldsymbol{\varepsilon}} \left[ \langle \bar{\eta}_n^{\mathtt{var}} - \bar{\eta}_n^{\mathtt{vX}}, \Sigma_m (\bar{\eta}_n^{\mathtt{var}} - \bar{\eta}_n^{\mathtt{vX}}) \rangle \right]}_{\triangleq \mathtt{V1}} \right)^{\frac{1}{2}} + \left( \mathbb{E}_{\boldsymbol{X},\boldsymbol{W},\boldsymbol{\varepsilon}} \left[ \langle \bar{\eta}_n^{\mathtt{vX}}, \Sigma_m \bar{\eta}_n^{\mathtt{vX}} \rangle \right] \right)^{\frac{1}{2}}$$

$$\leqslant (\mathtt{V1})^{\frac{1}{2}} + \left( \underbrace{\mathbb{E}_{\boldsymbol{X},\boldsymbol{W},\boldsymbol{\varepsilon}} \left[ \langle \bar{\eta}_n^{\mathtt{vX}} - \bar{\eta}_n^{\mathtt{vXW}}, \Sigma_m (\bar{\eta}_n^{\mathtt{vX}} - \bar{\eta}_n^{\mathtt{vXW}}) \rangle \right]}_{\triangleq \mathtt{V2}} \right)^{\frac{1}{2}} + \underbrace{[\mathbb{E}_{\boldsymbol{X},\boldsymbol{W},\boldsymbol{\varepsilon}} \langle \bar{\eta}_n^{\mathtt{vXW}}, \Sigma_m \bar{\eta}_n^{\mathtt{vXW}} \rangle]^{\frac{1}{2}}}_{\triangleq \mathtt{V3}} \,.$$

$$(40)$$

Accordingly, the Variance can be decomposed as $\mathtt{Variance} \lesssim \mathtt{V1} + \mathtt{V2} + \mathtt{V3}$, and in the next we give the error bounds for them, respectively.

### G.1  Bound for V3

In this section, we aim to bound $\mathtt{V3} := \mathbb{E}_{\boldsymbol{X},\boldsymbol{W},\boldsymbol{\varepsilon}} \langle \bar{\eta}_n^{\mathtt{vXW}}, \Sigma_m \bar{\eta}_n^{\mathtt{vXW}} \rangle$. Note that $\mathbb{E}_{\boldsymbol{X},\boldsymbol{\varepsilon}}[\eta_t^{\mathtt{vXW}} | \eta_{t-1}^{\mathtt{vXW}}] = (I - \gamma_t \widetilde{\Sigma}_m)\eta_{t-1}^{\mathtt{vXW}}$, similar to Appendix F.2 for B2, we have the following expression for V3

$$\mathtt{V3} := \mathbb{E}_{\boldsymbol{X},\boldsymbol{W},\boldsymbol{\varepsilon}} \langle \bar{\eta}_n^{\mathtt{vXW}}, \Sigma_m \bar{\eta}_n^{\mathtt{vXW}} \rangle = \mathbb{E}_{\boldsymbol{W}} \left[ \mathbb{E}_{\boldsymbol{X},\boldsymbol{\varepsilon}} \langle \Sigma_m, \bar{\eta}_n^{\mathtt{vXW}} \otimes \bar{\eta}_n^{\mathtt{vXW}} \rangle \right]$$

$$= \frac{1}{n^2} \mathbb{E}_{\boldsymbol{W}} \left( \left\langle \Sigma_m, \sum_{0 \leqslant k \leqslant t \leqslant n-1} \mathbb{E}_{\boldsymbol{X},\boldsymbol{\varepsilon}}[\eta_t^{\mathtt{vXW}} \otimes \eta_k^{\mathtt{vXW}}] + \sum_{0 \leqslant k < t \leqslant n-1} \mathbb{E}_{\boldsymbol{X},\boldsymbol{\varepsilon}}[\eta_t^{\mathtt{vXW}} \otimes \eta_k^{\mathtt{vXW}}] \right\rangle \right)$$

$$\leqslant \frac{1}{n^2} \mathbb{E}_{\boldsymbol{W}} \left( \left\langle \Sigma_m, \sum_{0 \leqslant k \leqslant t \leqslant n-1} \mathbb{E}_{\boldsymbol{X},\boldsymbol{\varepsilon}}[\eta_t^{\mathtt{vXW}} \otimes \eta_k^{\mathtt{vXW}}] + \sum_{0 \leqslant k \leqslant t \leqslant n-1} \mathbb{E}_{\boldsymbol{X},\boldsymbol{\varepsilon}}[\eta_t^{\mathtt{vXW}} \otimes \eta_k^{\mathtt{vXW}}] \right\rangle \right) \tag{41}$$

$$= \frac{2}{n^2} \sum_{t=0}^{n-1} \sum_{k=t}^{n-1} \mathbb{E}_{\boldsymbol{W}} \left\langle \prod_{j=t}^{k-1} (I - \gamma_j \widetilde{\Sigma}_m) \Sigma_m, \underbrace{\mathbb{E}_{\boldsymbol{X},\boldsymbol{\varepsilon}}[\eta_t^{\mathtt{vXW}} \otimes \eta_t^{\mathtt{vXW}}]}_{:=C_t^{\mathtt{vXW}}} \right\rangle,$$

and thus we have the following error bound for V3.

**Proposition 5.** *Under Assumption 1, 3, 5 with $\tau > 0$, if the step-size $\gamma_t := \gamma_0 t^{-\zeta}$ with $\zeta \in [0, 1)$ satisfies $\gamma_0 \leqslant \frac{1}{\mathrm{Tr}(\widetilde{\Sigma}_m)}$, then V3 can be bounded by*

$$\mathtt{V3} \lesssim \begin{cases} \gamma_0 \tau^2 \dfrac{m}{n^{1-\zeta}}, & \text{if } m \leqslant n \\ \gamma_0 \tau^2 \left( n^{\zeta-1} + \dfrac{n}{m} \right), & \text{if } m > n \,. \end{cases}$$

To prove Proposition 5, we need the following lemma.

**Lemma 10.** *Denote* $C_t^{\texttt{vXW}} := \mathbb{E}_{\boldsymbol{X},\boldsymbol{\varepsilon}}[\eta_t^{\texttt{vXW}} \otimes \eta_t^{\texttt{vXW}}]$, *under Assumptions 1, 3, 5 with* $\tau > 0$, *if* $\gamma_0 \leqslant 1/\text{Tr}(\widetilde{\Sigma}_m)$, *we have*

$$C_t^{\texttt{vXW}} \preccurlyeq \tau^2 \sum_{k=1}^{t} \gamma_k^2 \prod_{j=k+1}^{t} (I - \gamma_j \widetilde{\Sigma}_m)^2 \Sigma_m \,.$$

*Proof.* Recall the definition of $\eta_t^{\texttt{vXW}}$ in Eq. (9), it can be further represented as

$$\eta_t^{\texttt{vXW}} = (I - \gamma_t \widetilde{\Sigma}_m)\eta_{t-1}^{\texttt{vXW}} + \gamma_t \varepsilon_k \varphi(\boldsymbol{x}_k) = \sum_{k=1}^{t} \prod_{j=k+1}^{t} (I - \gamma_j \widetilde{\Sigma}_m)\gamma_k \varepsilon_k \varphi(\boldsymbol{x}_k) \quad \text{with } \eta_0^{\texttt{vXW}} = 0 \,.$$

Accordingly, $C_t^{\texttt{vXW}}$ admits (with $C_0^{\texttt{vXW}} = 0$)

$$C_t^{\texttt{vXW}} = \sum_{k=1}^{t} \prod_{j=k+1}^{t} (I - \gamma_j \widetilde{\Sigma}_m)^2 \gamma_k^2 \Xi \preccurlyeq \tau^2 \sum_{k=1}^{t} \gamma_k^2 \prod_{j=k+1}^{t} (I - \gamma_j \widetilde{\Sigma}_m)^2 \Sigma_m \quad \text{[using Assumption 5]}$$

where we use $\mathbb{E}[\varepsilon_i \varepsilon_j] = 0$ for $i \neq j$. $\qquad\square$

In the next, we are ready to bound V3 in Proposition 5.

*Proof of Proposition 5.* Note that $\widetilde{\lambda}_1 \sim \mathcal{O}(1)$ and $\widetilde{\lambda}_2 \sim \mathcal{O}(1/m)$ in Lemma 2, we take the upper bound of the integral in Eq. (22) to $\frac{n^\zeta}{\widetilde{\lambda}_1 \gamma_0}$ for $\widetilde{\lambda}_1$. However, according to the order of $\widetilde{\lambda}_2$, if $\widetilde{\lambda}_2 \lesssim 1/n$, the exact upper bound is tight. Based on this, we first consider that $m \leqslant n$ case such that $\widetilde{\lambda}_2 \gtrsim 1/n$, and then focus on the $m \geqslant n$ case. Taking $\frac{n^\zeta}{\widetilde{\lambda}_i \gamma_0}$ in Eq. (22) and $\frac{\gamma_0}{\widetilde{\lambda}_i}$ in Eq. (24), we have

$$\texttt{V3} := \mathbb{E}_{\boldsymbol{X},\boldsymbol{W},\boldsymbol{\varepsilon}} \langle \bar{\eta}_n^{\texttt{vXW}}, \Sigma_m \bar{\eta}_n^{\texttt{vXW}} \rangle = \mathbb{E}_{\boldsymbol{X},\boldsymbol{W},\boldsymbol{\varepsilon}} \langle \Sigma_m, \bar{\eta}_n^{\texttt{vXW}} \otimes \bar{\eta}_n^{\texttt{vXW}} \rangle$$

$$\leqslant \frac{2}{n^2} \sum_{t=0}^{n-1} \sum_{k=t}^{n-1} \mathbb{E}_{\boldsymbol{W}} \left\langle \prod_{j=t}^{k-1} (I - \gamma_j \widetilde{\Sigma}_m)\Sigma_m, \underbrace{\mathbb{E}_{\boldsymbol{X},\boldsymbol{\varepsilon}}[\bar{\eta}_t^{\texttt{vXW}} \otimes \bar{\eta}_t^{\texttt{vXW}}]}_{:=C_t^{\texttt{vXW}}} \right\rangle \quad \text{[using Eq. (41)]}$$

$$\leqslant \frac{2\tau^2}{n^2} \sum_{t=0}^{n-1} \sum_{k=t}^{n-1} \mathbb{E}_{\boldsymbol{W}} \left\langle \prod_{j=t}^{k-1} (I - \gamma_j \widetilde{\Sigma}_m)\Sigma_m, \sum_{s=1}^{t} \gamma_s^2 \prod_{j=s+1}^{t} (I - \gamma_j \widetilde{\Sigma}_m)^2 \Sigma_m \right\rangle \quad \text{[using Lemma 10]}$$

$$\leqslant \frac{2\tau^2}{n^2} \sum_{t=0}^{n-1} \sum_{k=t}^{n-1} \left\| \prod_{j=t}^{k-1} (I - \gamma_j \widetilde{\Sigma}_m)\widetilde{\Sigma}_m \sum_{s=1}^{t} \gamma_s^2 \prod_{j=s+1}^{t} (I - \gamma_j \widetilde{\Sigma}_m)^2 \right\|_2 \text{Tr}\left( \mathbb{E}_{\boldsymbol{W}}[\Sigma_m^2 \widetilde{\Sigma}_m^{-1}] \right)$$

$$\lesssim \frac{2\tau^2}{n^2} \sum_{t=0}^{n-1} \sum_{k=t}^{n-1} \max_{i \in \{1,2,\dots,m\}} \left\| \prod_{j=t}^{k-1} (1 - \gamma_j \widetilde{\lambda}_i)\widetilde{\lambda}_i \sum_{s=1}^{t} \gamma_s^2 \prod_{j=s+1}^{t} (1 - \gamma_j \widetilde{\lambda}_i)^2 \right\|_2 \quad \text{[using Lemma 4]}$$

$$\leqslant \frac{2\tau^2}{n^2} \sum_{t=0}^{n-1} \sum_{k=t}^{n-1} \max_{i \in \{1,2,\dots,m\}} \left\| \widetilde{\lambda}_i \exp\left( -\widetilde{\lambda}_i \gamma_0 \frac{k^{1-\zeta} - t^{1-\zeta}}{1 - \zeta} \right) \sum_{s=1}^{t} \gamma_s^2 \exp\left( -2\widetilde{\lambda}_i \gamma_0 \frac{(t+1)^{1-\zeta} - (s+1)^{1-\zeta}}{1 - \zeta} \right) \right\|_2$$

$$\lesssim \frac{\tau^2}{n^2} \sum_{t=0}^{n-1} \max_{i \in \{1,2,\dots,m\}} \left[ \widetilde{\lambda}_i \frac{n^\zeta}{\widetilde{\lambda}_i \gamma_0} \left( \frac{\gamma_0}{\widetilde{\lambda}_i} + \gamma_t^2 \right) \right] \quad \text{[using Eqs. (22), (24)]}$$

$$\leqslant \frac{\tau^2}{n^2} \left[ n^{1+\zeta} m + n^\zeta \text{Tr}(\widetilde{\Sigma}_m)\gamma_0 \int_0^n t^{-2\zeta} \mathrm{d}t \right]$$

$$\lesssim \gamma_0 \tau^2 \frac{m}{n^{1-\zeta}} \,, \quad \text{[using Lemma 2]}$$

$$(42)$$

where the last equality holds that $\int_0^n t^{-2\zeta} \mathrm{d}t \leqslant n$ for any $\zeta \in [0, 1)$.

If $\widetilde{\lambda}_2 \lesssim 1/n$, that means, $m > n$ in the over-parameterized regime, we have

$$
\begin{aligned}
\texttt{V3} &\lesssim \frac{2\tau^2}{n^2} \sum_{t=0}^{n-1} \left[ \widetilde{\lambda}_1 \frac{n^\zeta}{\widetilde{\lambda}_1 \gamma_0} \left( \frac{\gamma_0}{\widetilde{\lambda}_1} + \gamma_t^2 \right) + \widetilde{\lambda}_2 (n-t)t \right] \\
&\lesssim \frac{\gamma_0 \tau^2}{n^2} \left( n^{1+\zeta} + \widetilde{\lambda}_2 \frac{n(n-1)(n+1)}{6} \right) \quad \text{[since $\lambda_1 \sim \mathcal{O}(1)$]} \\
&\lesssim \gamma_0 \tau^2 \left( n^{\zeta-1} + \frac{n}{m} \right),
\end{aligned}
$$

which concludes the proof.

$\square$

### G.2 Bound for V2

Here we aim to bound $\texttt{V2}$

$$
\texttt{V2} := \mathbb{E}_{\boldsymbol{X},\boldsymbol{W},\boldsymbol{\varepsilon}} \left[ \langle \bar{\eta}_n^{\texttt{vX}} - \bar{\eta}_n^{\texttt{vXW}}, \Sigma_m (\bar{\eta}_n^{\texttt{vX}} - \bar{\eta}_n^{\texttt{vXW}}) \rangle \right].
$$

Recall the definition of $\eta_t^{\texttt{vX}}$ and $\eta_t^{\texttt{vXW}}$ in Eqs. (8) and (9), we have

$$
\eta_t^{\texttt{vXW}} = (I - \gamma_t \widetilde{\Sigma}_m) \eta_{t-1}^{\texttt{vXW}} + \gamma_t \varepsilon_k \varphi(\boldsymbol{x}_k) = \sum_{k=1}^{t} \prod_{j=k+1}^{t} (I - \gamma_j \widetilde{\Sigma}_m) \gamma_k \varepsilon_k \varphi(\boldsymbol{x}_k) \quad \text{with } \eta_0^{\texttt{vXW}} = 0,
$$

and accordingly, we define

$$
\begin{aligned}
\alpha_t^{\texttt{vX}-\texttt{W}} &:= \eta_t^{\texttt{vX}} - \eta_t^{\texttt{vXW}} = (I - \gamma_t \Sigma_m) \alpha_{t-1}^{\texttt{vX}-\texttt{W}} + \gamma_t (\widetilde{\Sigma}_m - \Sigma_m) \eta_{t-1}^{\texttt{vXW}}, \quad \text{with } \alpha_0^{\texttt{vX}-\texttt{W}} = 0 \\
&= \sum_{s=1}^{t} \prod_{i=s+1}^{t} (I - \gamma_i \Sigma_m) \gamma_s (\widetilde{\Sigma}_m - \Sigma_m) \sum_{k=1}^{s-1} \prod_{j=k+1}^{s-1} (I - \gamma_j \widetilde{\Sigma}_m) \gamma_k \varepsilon_k \varphi(\boldsymbol{x}_k).
\end{aligned}
$$

**Proposition 6.** *Under Assumptions 1, 3, 5 with $\tau > 0$, if the step-size $\gamma_t := \gamma_0 t^{-\zeta}$ with $\zeta \in [0, 1)$ satisfies*

$$
\gamma_0 \leqslant \frac{1}{\text{Tr}(\Sigma_m)}, \tag{43}
$$

*then $\texttt{V2}$ can be bounded by*

$$
\texttt{V2} \lesssim \begin{cases} \gamma_0 \tau^2 \dfrac{m}{n^{1-\zeta}}, & \text{if } m \leqslant n \\ \gamma_0 \tau^2, & \text{if } m > n. \end{cases}
$$

To prove Proposition 6, we need the following lemma.

**Lemma 11.** *Denote $C_t^{\texttt{vX}-\texttt{W}} := \mathbb{E}_{\boldsymbol{X},\boldsymbol{\varepsilon}} [\alpha_t^{\texttt{vX}-\texttt{W}} \otimes \alpha_t^{\texttt{vX}-\texttt{W}}]$, under Assumptions 1, 3, 5 with $\tau > 0$, if the step-size $\gamma_t := \gamma_0 t^{-\zeta}$ with $\zeta \in [0, 1)$ satisfies*

$$
\gamma_0 \leqslant \min \left\{ \frac{1}{\text{Tr}(\Sigma_m)}, \frac{1}{\text{Tr}(\widetilde{\Sigma}_m)} \right\},
$$

*we have*

$$
\| C_t^{\texttt{vX}-\texttt{W}} \|_2 \lesssim \tau^2 \gamma_0^2 \left( \gamma_0 \| \Sigma_m \|_2 + 1 \right) \left( \gamma_0 \| \widetilde{\Sigma}_m \|_2 + 1 \right).
$$

*Proof.* According to the definition of $C_t^{\texttt{vX}-\texttt{W}}$, it admits the following expression

$$
\begin{aligned}
C_t^{\texttt{vX}-\texttt{W}} &= \sum_{s=1}^{t} \prod_{i=s+1}^{t} (I - \gamma_i \Sigma_m) \gamma_s^2 (\widetilde{\Sigma}_m - \Sigma_m) \sum_{k=1}^{s-1} \prod_{j=k+1}^{s-1} (I - \gamma_j \widetilde{\Sigma}_m)^2 \gamma_k^2 \Xi (\widetilde{\Sigma}_m - \Sigma_m)(I - \gamma_i \Sigma_m) \\
&\preccurlyeq \sum_{s=1}^{t} \prod_{i=s+1}^{t} (I - \gamma_i \Sigma_m) \gamma_s^2 (\widetilde{\Sigma}_m - \Sigma_m) \sum_{k=1}^{s-1} \prod_{j=k+1}^{s-1} (I - \gamma_j \widetilde{\Sigma}_m)^2 \gamma_k^2 \Xi (\widetilde{\Sigma}_m - \Sigma_m)(I - \gamma_i \Sigma_m) \\
&\preccurlyeq \tau^2 \sum_{s=1}^{t} \prod_{i=s+1}^{t} (I - \gamma_i \Sigma_m) \gamma_s^2 (\widetilde{\Sigma}_m - \Sigma_m) \sum_{k=1}^{s-1} \prod_{j=k+1}^{s-1} (I - \gamma_j \widetilde{\Sigma}_m)^2 \gamma_k^2 \Sigma_m (\widetilde{\Sigma}_m - \Sigma_m)(I - \gamma_i \Sigma_m),
\end{aligned}
$$

where the first equality holds by $\mathbb{E}[\varepsilon_i \varepsilon_j] = 0$ for $i \neq j$ and the second inequality holds by Assumption 5.

Accordingly, $\|C_t^{\mathtt{vX-W}}\|_2$ can be upper bounded by

$$\|C_t^{\mathtt{vX-W}}\|_2 \leqslant \tau^2 \sum_{s=1}^{t} \gamma_s^2 \left\| \prod_{i=s+1}^{t} (I - \gamma_i \Sigma_m)^2 \Sigma_m (\widetilde{\Sigma}_m - \Sigma_m)^2 \sum_{k=1}^{s-1} \gamma_k^2 \prod_{j=k+1}^{s-1} (I - \gamma_j \widetilde{\Sigma}_m)^2 \right\|_2$$

$$\leqslant \tau^2 \sum_{s=1}^{t} \gamma_s^2 \left\| \prod_{i=s+1}^{t} (I - \gamma_i \Sigma_m)^2 \Sigma_m \right\|_2 \left\| \sum_{k=1}^{s-1} \gamma_k^2 \prod_{j=k+1}^{s-1} (I - \gamma_j \widetilde{\Sigma}_m)^2 \widetilde{\Sigma}_m \right\|_2 \left\| \widetilde{\Sigma}_m - 2\Sigma_m + \widetilde{\Sigma}_m^{-1} \Sigma_m^2 \right\|_2$$

$$\lesssim \tau^2 \sum_{s=1}^{t} \max_{q \in \{1,2,\ldots,m\}} \gamma_s^2 \exp\left( -2\lambda_q \sum_{i=s+1}^{t} \gamma_i \right) \lambda_q \sum_{k=1}^{s-1} \gamma_k^2 \max_{p \in \{1,2\}} \exp\left( -2\widetilde{\lambda}_p \sum_{j=k+1}^{s-1} \gamma_j \right) \widetilde{\lambda}_p$$

$$\left\| \widetilde{\Sigma}_m - 2\Sigma_m + \widetilde{\Sigma}_m^{-1} \Sigma_m^2 \right\|_2 .$$

Similar to Eq. (23), we have the following estimation

$$\sum_{k=1}^{s-1} \gamma_k^2 \prod_{j=k+1}^{s-1} (1 - \gamma_j \widetilde{\lambda}_p)^2 \leqslant \sum_{k=1}^{s-1} \gamma_k^2 \exp\left( -2\widetilde{\lambda}_p \sum_{j=k+1}^{s-1} \gamma_j \right)$$

$$\leqslant \gamma_{s-1}^2 + \gamma_0^2 \int_1^{s-1} u^{-2\zeta} \exp\left( -2\widetilde{\lambda}_p \gamma_0 \frac{s^{1-\zeta} - (u+1)^{1-\zeta}}{1-\zeta} \right) du$$

$$\leqslant \gamma_0^2 + \left( \frac{\gamma_0}{\widetilde{\lambda}_p} \wedge \gamma_0^2 s \right) ,$$

which implies

$$\max_{p=1,2} \widetilde{\lambda}_p \sum_{k=1}^{s-1} \gamma_k^2 \prod_{j=k+1}^{s-1} (1 - \gamma_j \widetilde{\lambda}_p)^2 \leqslant \gamma_0^2 \widetilde{\lambda}_1 + \gamma_0 \leqslant \gamma_0^2 \widetilde{\Sigma}_m + \gamma_0 . \tag{44}$$

Similar to Eq. (23), we have the following estimation

$$\sum_{s=1}^{t} \gamma_s^2 \exp\left( -2\lambda_q \sum_{i=s+1}^{t} \gamma_i \right) \leqslant \sum_{s=1}^{t} \gamma_s^2 \exp\left( -2\lambda_q \gamma_0 \frac{(t+1)^{1-\zeta} - (s+1)^{1-\zeta}}{1-\zeta} \right)$$

$$\leqslant \gamma_t^2 + \gamma_0^2 \int_1^{t} u^{-2\zeta} \exp\left( -2\lambda_q \gamma_0 \frac{(t+1)^{1-\zeta} - (u+1)^{1-\zeta}}{1-\zeta} \right) du$$

$$\leqslant \gamma_0^2 + \left( \frac{\gamma_0}{\lambda_q} \wedge \gamma_0^2 t \right) ,$$

which implies

$$\max_{q \in \{1,2,\ldots,m\}} \sum_{s=1}^{t} \gamma_s^2 \lambda_q \exp\left( -2\lambda_q \sum_{i=s+1}^{t} \gamma_i \right) = \gamma_0^2 \|\Sigma_m\|_2 + \gamma_0 . \tag{45}$$

Combining the above two equations (44) and (45), we have

$$\|C_t^{\mathtt{vX-W}}\|_2 \lesssim \tau^2 \gamma_0^2 \left( \gamma_0 \|\Sigma_m\|_2 + 1 \right) \left( \gamma_0 \|\widetilde{\Sigma}_m\|_2 + 1 \right) .$$

$\square$

*Proof of Proposition 6.* By virtue of $\mathbb{E}_{\boldsymbol{X},\boldsymbol{\varepsilon}}[\alpha_t^{\mathtt{vX}-\mathtt{W}}|\alpha_{t-1}^{\mathtt{vX}-\mathtt{W}}] = (I - \gamma_t\Sigma_m)\alpha_{t-1}^{\mathtt{vX}-\mathtt{W}}$ and Lemma 11, $\mathtt{V2}$ can be bounded by

$$
\begin{aligned}
\mathtt{V2} &= \mathbb{E}_{\boldsymbol{X},\boldsymbol{W},\boldsymbol{\varepsilon}}\left[\langle \bar{\eta}_n^{\mathtt{vX}} - \bar{\eta}_n^{\mathtt{vXW}}, \Sigma_m(\bar{\eta}_n^{\mathtt{vX}} - \bar{\eta}_n^{\mathtt{vXW}})\rangle\right] = \mathbb{E}_{\boldsymbol{W}}\langle \Sigma_m, \mathbb{E}_{\boldsymbol{X},\boldsymbol{\varepsilon}}[\bar{\alpha}_n^{\mathtt{vX}-\mathtt{W}} \otimes \bar{\alpha}_n^{\mathtt{vX}-\mathtt{W}}]\rangle \\
&\leqslant \frac{2}{n^2}\sum_{t=0}^{n-1}\sum_{k=t}^{n-1}\mathbb{E}_{\boldsymbol{W}}\left\langle \prod_{j=t}^{k-1}(I-\gamma_j\Sigma_m)\Sigma_m, \underbrace{\mathbb{E}_{\boldsymbol{X},\boldsymbol{\varepsilon}}[\eta_t^{\mathtt{vX}-\mathtt{W}}\otimes\eta_t^{\mathtt{vX}-\mathtt{W}}]}_{:=C_t^{\mathtt{vX}-\mathtt{W}}}\right\rangle \\
&\lesssim \frac{\tau^2\gamma_0^2}{n^2}\|\widetilde{\Sigma}_m\|_2\mathbb{E}_{\boldsymbol{W}}\left(\left\|\widetilde{\Sigma}_m - 2\Sigma_m + \widetilde{\Sigma}_m^{-1}\Sigma_m^2\right\|_2 [\|\Sigma_m\|_2\gamma_0 + 1]\,\mathrm{Tr}\left[\sum_{t=0}^{n-1}\sum_{k=t}^{n-1}\prod_{j=t}^{k-1}(I-\gamma_j\Sigma_m)\Sigma_m\right]\right) \\
&\lesssim \frac{\tau^2\gamma_0^2}{n^2}\|\widetilde{\Sigma}_m\|_2\mathbb{E}_{\boldsymbol{W}}\left[\|\Sigma_m\|_2\left\|\widetilde{\Sigma}_m - 2\Sigma_m + \widetilde{\Sigma}_m^{-1}\Sigma_m^2\right\|_2\sum_{i=1}^{m}\sum_{t=0}^{n-1}\lambda_i\left(\frac{n^\zeta}{\lambda_i\gamma_0}\wedge(n-t)\right)\right]. \quad \text{[using Eq. (22)]}
\end{aligned}
$$

In the $m \leqslant n$ case, we choose $n^\zeta/(\lambda_i\gamma_0)$, and thus

$$
\begin{aligned}
\mathtt{V2} &\lesssim \frac{\tau^2 m\gamma_0^2}{n^2}\|\widetilde{\Sigma}_m\|_2\mathbb{E}_{\boldsymbol{W}}\left[\|\Sigma_m\|_2\left\|\widetilde{\Sigma}_m - 2\Sigma_m + \widetilde{\Sigma}_m^{-1}\Sigma_m^2\right\|_2\right]\frac{n^{1+\zeta}}{\gamma_0} \\
&\leqslant \tau^2\gamma_0\frac{m\|\widetilde{\Sigma}_m\|_2}{n^{1-\zeta}}\sqrt{\mathbb{E}_{\boldsymbol{W}}\|\Sigma_m\|_2^2}\sqrt{\mathbb{E}_{\boldsymbol{W}}\left\|\widetilde{\Sigma}_m - 2\Sigma_m + \widetilde{\Sigma}_m^{-1}\Sigma_m^2\right\|_2^2} \quad \text{[using Cauchy–Schwarz inequality]} \\
&\lesssim \tau^2\gamma_0\frac{m}{n^{1-\zeta}}. \quad \text{[using Lemma 2 and 4]}
\end{aligned}
$$

If $m > n$, we have

$$
\begin{aligned}
\mathtt{V2} &\lesssim \frac{2\tau^2\gamma_0^2}{n^2}\|\widetilde{\Sigma}_m\|_2\mathbb{E}_{\boldsymbol{W}}\left([\mathrm{Tr}(\Sigma_m)]^2\left\|\widetilde{\Sigma}_m - 2\Sigma_m + \widetilde{\Sigma}_m^{-1}\Sigma_m^2\right\|_2\right)\sum_{t=0}^{n-1}t \\
&\leqslant \tau^2\gamma_0\|\widetilde{\Sigma}_m\|_2\sqrt{\mathbb{E}_{\boldsymbol{W}}[\mathrm{Tr}(\Sigma_m)]^2}\sqrt{\mathbb{E}_{\boldsymbol{W}}\left\|\widetilde{\Sigma}_m - 2\Sigma_m + \widetilde{\Sigma}_m^{-1}\Sigma_m^2\right\|_2^2} \\
&\lesssim \tau^2\gamma_0, \quad \text{[using Lemmas 2 and 4]}
\end{aligned}
$$

which concludes the proof. $\qquad\square$

### G.3 Bound for $\mathtt{V1}$

Here we aim to bound $\mathtt{V1}$

$$
\mathtt{V1} := \mathbb{E}_{\boldsymbol{X},\boldsymbol{W},\boldsymbol{\varepsilon}}\left[\langle \bar{\eta}_n^{\mathtt{var}} - \bar{\eta}_n^{\mathtt{vX}}, \Sigma_m(\bar{\eta}_n^{\mathtt{var}} - \bar{\eta}_n^{\mathtt{vX}})\rangle\right].
$$

Recall the definition of $\eta_t^{\mathtt{var}}$ in Eq. (6) and $\eta_t^{\mathtt{vX}}$ in Eq. (8), we define

$$
\alpha_t^{\mathtt{v}-\mathtt{X}} := \eta_t^{\mathtt{var}} - \eta_t^{\mathtt{vX}} = [I - \gamma_t\varphi(\boldsymbol{x}_t)\otimes\varphi(\boldsymbol{x}_t)]\alpha_{t-1}^{\mathtt{v}-\mathtt{X}} + \gamma_t[\Sigma_m - \varphi(\boldsymbol{x}_t)\otimes\varphi(\boldsymbol{x}_t)]\eta_{t-1}^{\mathtt{vX}}, \quad \text{with } \alpha_0^{\mathtt{v}-\mathtt{X}} = 0.
$$

$$
\begin{aligned}
&= [I - \gamma_t\varphi(\boldsymbol{x}_t)\otimes\varphi(\boldsymbol{x}_t)]\alpha_{t-1}^{\mathtt{v}-\mathtt{X}} + \gamma_t[\Sigma_m - \varphi(\boldsymbol{x}_t)\otimes\varphi(\boldsymbol{x}_t)]\sum_{k=1}^{t-1}\prod_{j=k+1}^{t-1}(I-\gamma_j\Sigma_m)\gamma_k\varepsilon_k\varphi(\boldsymbol{x}_k) \\
&= \sum_{s=1}^{t}\prod_{i=s+1}^{t}\gamma_s[I - \gamma_i\varphi(\boldsymbol{x}_i)\otimes\varphi(\boldsymbol{x}_i)][\Sigma_m - \varphi(\boldsymbol{x}_t)\otimes\varphi(\boldsymbol{x}_t)]\sum_{k=1}^{s-1}\prod_{j=k+1}^{s-1}(I-\gamma_j\Sigma_m)\gamma_k\varepsilon_k\varphi(\boldsymbol{x}_k),
\end{aligned}
$$

and thus the error bound for $\mathtt{V1}$ is given by the following proposition.

**Proposition 7.** *Under Assumption 1, 2, 3, 4 with $r' \geqslant 1$, and Assumption 5 with $\tau > 0$, if the step-size $\gamma_t := \gamma_0 t^{-\zeta}$ with $\zeta \in [0,1)$ satisfies*

$$
\gamma_0 < \min\left\{\frac{1}{r'\mathrm{Tr}(\Sigma_m)}, \frac{1}{2\mathrm{Tr}(\Sigma_m)}\right\},
$$

*then $\mathtt{V1}$ can be bounded by*

$$
\mathtt{V1} \lesssim \tau^2 r'\gamma_0^2\begin{cases} \dfrac{m}{n^{1-\zeta}}, & \text{if } m \leqslant n \\ 1, & \text{if } m > n. \end{cases}
$$

To prove Proposition 7, we need the following lemma. Define $C_t^{\mathtt{v-x}} := \mathbb{E}_{\boldsymbol{X},\boldsymbol{\varepsilon}}[\alpha_t^{\mathtt{v-x}} \otimes \alpha_t^{\mathtt{v-x}}]$, we have the following lemma that is useful to bound $C_t^{\mathtt{v-x}}$.

**Lemma 12.** *Denote $C_t^{\mathtt{v-x}} := \mathbb{E}_{\boldsymbol{X},\boldsymbol{\varepsilon}}[\alpha_t^{\mathtt{v-x}} \otimes \alpha_t^{\mathtt{v-x}}]$, under Assumptions 1, 2, 3, 4 with $r' \geqslant 1$, and Assumption 5 with $\tau > 0$, if the step-size $\gamma_t := \gamma_0 t^{-\zeta}$ with $\zeta \in [0,1)$ satisfies*

$$\gamma_0 < \min\left\{ \frac{1}{r'\mathrm{Tr}(\Sigma_m)}, \frac{1}{c'\mathrm{Tr}(\Sigma_m)} \right\},$$

*where $c'$ is defined in Eq. (14). Then, we have*

$$C_t^{\mathtt{v-x}} \preccurlyeq \frac{\gamma_0^2 r' \tau^2 [\mathrm{Tr}(\Sigma_m) + \gamma_0 \mathrm{Tr}(\Sigma_m^2)]}{1 - \gamma_0 r' \mathrm{Tr}(\Sigma_m)} I \,.$$

*Proof.* According to the definition of $C_t^{\mathtt{v-x}}$, it admits the following expression

$$C_t^{\mathtt{v-x}} = \sum_{s=1}^{t} \prod_{i=s+1}^{t} \gamma_s^2 \mathbb{E}_{\boldsymbol{x}}[I - \gamma_i \varphi(\boldsymbol{x}_i) \otimes \varphi(\boldsymbol{x}_i)]^2 \mathbb{E}_{\boldsymbol{x}}[\Sigma_m - \varphi(\boldsymbol{x}_t) \otimes \varphi(\boldsymbol{x}_t)]^2 \sum_{k=1}^{s-1} \prod_{j=k+1}^{s-1} (I - \gamma_j \Sigma_m)^2 \gamma_k^2 \Xi$$

$$= (I - \gamma_t T^{\mathtt{w}}) \circ C_{t-1}^{\mathtt{v-x}} + \gamma_t^2 (S^{\mathtt{w}} - \widetilde{S}^{\mathtt{w}}) \circ \sum_{k=1}^{t-1} \prod_{j=k+1}^{t-1} (I - \gamma_j \Sigma_m)^2 \gamma_k^2 \Xi \quad \text{[using PSD operators]}$$

$$\preccurlyeq (I - \gamma_t T^{\mathtt{w}}) \circ C_{t-1}^{\mathtt{v-x}} + \gamma_t^2 S^{\mathtt{w}} \circ \sum_{k=1}^{t-1} \prod_{j=k+1}^{t-1} (I - \gamma_j \Sigma_m)^2 \gamma_k^2 \Xi \quad \text{[using } S^{\mathtt{w}} \succcurlyeq \widetilde{S}^{\mathtt{w}}]$$

$$\preccurlyeq (I - \gamma_t T^{\mathtt{w}}) \circ C_{t-1}^{\mathtt{v-x}} + \tau^2 \gamma_t^2 S^{\mathtt{w}} \circ \sum_{k=1}^{t-1} \prod_{j=k+1}^{t-1} (I - \gamma_j \Sigma_m)^2 \gamma_k^2 \Sigma_m \quad \text{[using Assumption 5]}$$

$$\preccurlyeq (I - \gamma_t T^{\mathtt{w}}) \circ C_{t-1}^{\mathtt{v-x}} + \tau^2 \gamma_t^2 r' \mathrm{Tr}\left[ \sum_{k=1}^{t-1} \prod_{j=k+1}^{t-1} (I - \gamma_j \Sigma_m)^2 \gamma_k^2 \Sigma_m^2 \right] \Sigma_m \,. \quad \text{[using Assumption 4]}$$

$$\tag{46}$$

Similar to Eq. (23), we have the following estimation

$$\mathrm{Tr}\left[ \sum_{k=1}^{t-1} \prod_{j=k+1}^{t-1} (I - \gamma_j \Sigma_m)^2 \Sigma_m^2 \gamma_k^2 \right] = \sum_{i=1}^{m} \lambda_i^2 \sum_{k=1}^{t-1} \gamma_k^2 \prod_{j=k+1}^{t-1} (1 - \gamma_j \lambda_i)^2 \leqslant \sum_{i=1}^{m} \lambda_i^2 \sum_{k=1}^{t-1} \gamma_k^2 \exp\left( -2\lambda_i \sum_{j=k+1}^{s-1} \gamma_j \right)$$

$$\leqslant \gamma_0^2 \sum_{i=1}^{m} \lambda_i^2 \left[ 1 + \int_1^{t-1} u^{-2\zeta} \exp\left( -2\lambda_i \gamma_0 \frac{t^{1-\zeta} - (u+1)^{1-\zeta}}{1-\zeta} \right) \mathrm{d}u \right]$$

$$\leqslant \gamma_0^2 \mathrm{Tr}(\Sigma_m^2) + \sum_{i=1}^{m} \lambda_i^2 \left( \frac{\gamma_0}{\lambda_i} \wedge \gamma_0^2 t \right) \quad \text{[using Eq. (24)]}$$

$$\leqslant \gamma_0^2 \mathrm{Tr}(\Sigma_m^2) + \gamma_0 \mathrm{Tr}(\Sigma_m) \,,$$

where we use the error bound $\frac{\gamma_0}{\lambda_i}$ instead of the exact one $\gamma_0^2 t$ for tight estimation.

Taking the above equation back to Eq. (46), we have

$$C_t^{\mathtt{v-x}} \preccurlyeq (I - \gamma_t T^{\mathtt{w}}) \circ C_{t-1}^{\mathtt{v-x}} + \gamma_t^2 \tau^2 r' \gamma_0 [\mathrm{Tr}(\Sigma_m) + \gamma_0 \mathrm{Tr}(\Sigma_m^2)] \Sigma_m$$

$$\preccurlyeq \tau^2 r' \gamma_0 [\mathrm{Tr}(\Sigma_m) + \gamma_0 \mathrm{Tr}(\Sigma_m^2)] \sum_{s=1}^{t} \prod_{i=s+1}^{t} (I - \gamma_i T^{\mathtt{w}}) \circ \gamma_s^2 \Sigma_m$$

$$\preccurlyeq \frac{\gamma_0^2 r' \tau^2 [\mathrm{Tr}(\Sigma_m) + \gamma_0 \mathrm{Tr}(\Sigma_m^2)]}{1 - \gamma_0 r' \mathrm{Tr}(\Sigma_m)} I \,, \quad \text{[using Lemma 5]}$$

which concludes the proof. $\qquad\square$

*Proof of Proposition 7.* Accordingly, by virtue of $\mathbb{E}_{\boldsymbol{X},\boldsymbol{\varepsilon}}[\alpha_t^{\mathtt{v}-\mathtt{X}}|\alpha_{t-1}^{\mathtt{v}-\mathtt{X}}] = (I - \gamma_t\Sigma_m)\alpha_{t-1}^{\mathtt{v}-\mathtt{X}}$ and Lemma 12, $\mathtt{V1}$ can be bounded by

$$
\begin{aligned}
\mathtt{V1} &= \mathbb{E}_{\boldsymbol{X},\boldsymbol{W},\boldsymbol{\varepsilon}}\left[\langle \bar{\eta}_n^{\mathtt{var}} - \bar{\eta}_n^{\mathtt{v}-\mathtt{X}}, \Sigma_m(\bar{\eta}_n^{\mathtt{var}} - \bar{\eta}_n^{\mathtt{v}-\mathtt{X}})\rangle\right] = \mathbb{E}_{\boldsymbol{W}}\langle \Sigma_m, \mathbb{E}_{\boldsymbol{X},\boldsymbol{\varepsilon}}[\bar{\alpha}_n^{\mathtt{v}-\mathtt{X}} \otimes \bar{\alpha}_n^{\mathtt{v}-\mathtt{X}}]\rangle \\
&\leqslant \frac{2}{n^2}\sum_{t=0}^{n-1}\sum_{k=t}^{n-1}\mathbb{E}_{\boldsymbol{W}}\left\langle \prod_{j=t}^{k-1}(I - \gamma_j\Sigma_m)\Sigma_m, \underbrace{\mathbb{E}_{\boldsymbol{X},\boldsymbol{\varepsilon}}[\eta_t^{\mathtt{v}-\mathtt{X}} \otimes \eta_t^{\mathtt{v}-\mathtt{X}}]}_{:=C_t^{\mathtt{v}-\mathtt{X}}}\right\rangle \\
&\lesssim \frac{\tau^2\gamma_0^2 r'}{n^2}\mathbb{E}_{\boldsymbol{W}}\left[\frac{[\mathrm{Tr}(\Sigma_m) + \gamma_0\mathrm{Tr}(\Sigma_m^2)]}{1 - \gamma_0 r'\mathrm{Tr}(\Sigma_m)}\sum_{i=1}^{m}\sum_{t=0}^{n-1}\lambda_i\left(\frac{n^\zeta}{\lambda_i\gamma_0} \wedge (n-t)\right)\right], \quad \text{[using Lemma 12]}
\end{aligned}
$$

where the last inequality follows the integral estimation in Eq. (22).

For $m \leqslant n$, we use $\frac{n^\zeta}{\lambda_i\gamma_0}$, and thus

$$
\mathtt{V1} \lesssim \frac{\tau^2\gamma_0 r'm}{n^{1-\zeta}}\mathbb{E}_{\boldsymbol{W}}\left[\frac{[\mathrm{Tr}(\Sigma_m) + \gamma_0\mathrm{Tr}(\Sigma_m^2)]}{1 - \gamma_0 r'\mathrm{Tr}(\Sigma_m)}\right] \lesssim \tau^2 r'\gamma_0\frac{m}{n^{1-\zeta}},
$$

where we use $\mathrm{Tr}(\Sigma_m)$ as a nonnegative sub-exponential random variable with the sub-exponential norm $\mathcal{O}(1)$ in Lemma 2.

For $m > n$, take $n - t$, we have

$$
\mathtt{V1} \lesssim \tau^2\gamma_0^2 r'\mathbb{E}_{\boldsymbol{W}}\left[\frac{[\mathrm{Tr}(\Sigma_m) + \gamma_0\mathrm{Tr}(\Sigma_m^2)]}{1 - \gamma_0 r'\mathrm{Tr}(\Sigma_m)}\right] \lesssim \tau^2 r'\gamma_0^2 \sim \mathcal{O}(1).
$$

$\square$

### G.4 Proof of Theorem 2

*Proof.* Combining the above results for three terms $\mathtt{V1}$, $\mathtt{V2}$, $\mathtt{V3}$, we can directly obtain the result for $\mathtt{Variance}$.

$$
\begin{aligned}
\mathtt{Variance} &\leqslant \left(\sqrt{\mathtt{V1}} + \sqrt{\mathtt{V2}} + \sqrt{\mathtt{V3}}\right)^2 \leqslant 3(\mathtt{V1} + \mathtt{V2} + \mathtt{V3}) \\
&\lesssim \gamma_0 r'\tau^2\begin{cases} mn^{\zeta-1}, & \text{if } m \leqslant n \\ 1 + n^{\zeta-1} + \dfrac{n}{m}, & \text{if } m > n \end{cases} \\
&\sim \begin{cases} \mathcal{O}\left(mn^{\zeta-1}\right), & \text{if } m \leqslant n \\ \mathcal{O}\left(1 + n^{\zeta-1} + \dfrac{n}{m}\right), & \text{if } m > n \end{cases}
\end{aligned}
$$

which concludes the proof. $\square$

## H More experiments

In this section, we provide additional experimental results to support our theory.

### H.1 Results on a regression dataset

We conduct the RF regression via averaged SGD and minimum solution under different initialization schemes and different epochs on a synthetic regression dataset across the Gaussian kernel.

**data generation:** Apart from the commonly used MNIST in the double descent topic [13, 53], we also add a synthetic regression dataset via normalized MSE in Figure 3(a) for fully supporting our work. The data are generated from a normal Gaussian distribution with the training data ranging from $n = 10$ to $n = 400$, the test data being 200, and the feature dimension $d = 50$. The label is generated by $y = f_\rho(\boldsymbol{x}) + \epsilon$, where the $\epsilon$ is a Gaussian noise with the variance 0.01. The target function $f^*$ is generated by a Laplace kernel $k(\boldsymbol{x}, \boldsymbol{x}') = \exp\left(-\frac{\|\boldsymbol{x}-\boldsymbol{x}'\|_2}{d}\right)$, to ensure $f^* \in \mathcal{H}$. To be specific, for any a data point $\boldsymbol{x} \in \mathbb{R}^d$, its target function is $f^*(\boldsymbol{x}) = [k(\boldsymbol{x}, \boldsymbol{x}_1), k(\boldsymbol{x}, \boldsymbol{x}_2), \cdots, k(\boldsymbol{x}, \boldsymbol{x}_n)]\boldsymbol{w}$, where

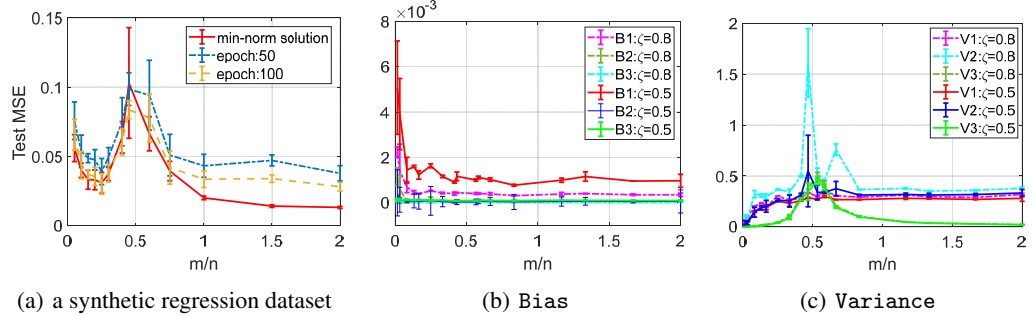

| (a) a synthetic regression dataset | (b) `Bias` | (c) `Variance` |
|---|---|---|

Figure 3: Normalized MSE (mean±std.) of RF regression with different epochs on a synthetic regression dataset across the Gaussian kernel in (a); trends of `Bias` and `Variance` under different step-size are empirically given in (b) and (c), respectively.

$w \in \mathbb{R}^n$ is a standard random Gaussian vector as a sign. We remark that the reason why we do not choose the Gaussian kernel as the target function is to avoid the data and model induced by a same (type) kernel.

**experimental settings:** We follow Figure 2(a) with the same experiment settings, i.e., conducting RF regression via averaged SGD and minimum-norm solution under the Gaussian kernel. In our experiment, the initial step-size is set to $\gamma_0 = 1$ with $\zeta = 0.5$. Nevertheless, we take *constant initialization* (i.e., set the initialization point as a constant vector) and *different epochs* (i.e., 50 and 100) for broad comparison.

Fig. 3(a) shows that, first, under this regression dataset with *constant initialization*, we still observe a phase transition between the two sides of the interpolation threshold at $2m = n$ when min-norm solution and averaged SGD are employed, which leads to the double descent phenomenon. Second, averaged SGD with more epochs result in a better generalization performance, but is still slightly inferior to that with min-norm solution. We need remark that, when employing gradient descent, under mild conditions, the solution converges to the minimum norm solution, as suggested by [4]. Nevertheless, whether this result holds for SGD is unclear, depending on the choice of the ground truth, step-size, etc [64, 65]. Studying the property of converged solution is indeed beyond the scope of this paper.

## H.2 Different step-size on Bias and Variance

Following Section 5.2, we also evaluate our error bounds for `Bias` and `Variance` under different step-sizes on the MNIST dataset. Figure 3(b) on bias and 3(c) on variance coincides with the results of Section 5.2: monotonically decreasing bias and unimodal variance (phase transition of `V3` and non-decreasing `V1` and `V2`) under different step-size. We remark that, the estimated error bounds are normalized for better illustration, and accordingly we cannot directly compare the value of these components under different step-size.