# OpenReview forum: "On the Double Descent of Random Features Models Trained with SGD"
_NeurIPS.cc/2022/Conference — NeurIPS 2022 Accept_

### Official Review · Reviewer_zi7u · 2022-07-11

**Rating:** 6
**Confidence:** 4
**Soundness:** 4 excellent
**Presentation:** 3 good
**Contribution:** 2 fair

**Summary:**

This paper derives the excess risk bounds for random feature (RF) model optimized by SGD, with polynomial-decay stepsize (including constant stepsize). Excess risk is decomposed into bias and variance errors, and it is shown that bias error decreases monotonically with the increase of the number of random features, while the variance error demonstrates unimodal behavior.

**Questions:**

1. The key finding of Zou et al. (2021) is that an effective dimension is established. How is the results of this paper related to the effective dimension?

**Limitations:**

The limitations are well evaluated by the authors.

**Strengths And Weaknesses:**

Stengths:
1. The double descent phenomenon is theoretically characterized.

Weaknesses:
1. The paper lacks discussion apart from the descent phenomenon. For example, how is RF different from the setting of linear regression? How does the input dimension $d$ affect the result ($d$ does not explicitly appear in the main theorems)?
2. The technique of the paper is quite similar to Zou et al. (2021), except that other sources of randomness are considered. Therefore, the paper is not technically strong.
3. The main theorems still have terms like $\mathbb{E}[1-\gamma_0r'\mathrm{Tr}(\Sigma_m)]^2$ and $\mathbb{E}[1-\gamma_0r'\mathrm{Tr}(\Sigma_m)]^4$ which require further analysis.

References
Difan Zou, Jingfeng Wu, Vladimir Braverman, Quanquan Gu, and Sham M Kakade. Benign overfitting of constant-stepsize sgd for linear regression. In Conference on Learning Theory, 2021.

---

> ### Author Response · Authors · 2022-08-01
> **Response to Reviewer zi7u**
>
> We thank the reviewer's constructive comments.
>
> **Q1:** The paper lacks discussion apart from the descent phenomenon. How is RF different from the setting of linear regression?
>
> **A1:** In our previous version, we had provided a detailed discussion in Appendix A.2 (see line 622) due to page limit. According to your suggestions, we briefly discuss it here.
>
> When taking $m=d$, our setting falls into least squares. In this case, our upper bound result is
>
> $\text{excess risk} \lesssim \gamma_0 \tau^2 \bigg(\frac{1}{n} + \frac{d}{n} \bigg)$,  if $d \leq n$;
>
> $\text{excess risk} \lesssim \gamma_0 \tau^2 \bigg(1 + \frac{1}{n} + \frac{n}{d} \bigg)$, if $d > n$,
>
>
> which matches the same order with Corollary 1 in [S1].
>
> When compared to Zou et al. (2021), if taking the effective dimension $k^* = \min (n,d )$ (we don't add any data spectrum here), we can recover their result. Regarding to the effective dimension issue, we reply this in **Q5**.
>
> [S1] H. Trevor, A. Montanari, S. Rosset, and R. Tibshirani. "Surprises in high-dimensional ridgeless least squares interpolation." *Annals of Statistics*, 2022.
>
> ---
>
> **Q2:** How does the input dimension $d$ affect the result ($d$ does not explicitly appear in the main theorems)?
>
> **A2:**  Random feature models in fact map the data from $\mathbb{R}^d$ to $\mathbb{R}^m$ (normally $m \geq d$), and thus the input dimension $d$ is included into $m$. In this case, it's enough to consider/tune $m$ in practice and theory. Even if $d \rightarrow \infty$ (i.e., $m \rightarrow \infty$), our result can still handle it.
>
> ---
>
> **Q3:** The technique of the paper is quite similar to Zou et al. (2021), except that other sources of randomness are considered. Therefore, the paper is not technically strong.
>
> **A3:** Handling other sources of randomness is precisely the challenge, which in fact also leads to different ways of tackling the effective dimension that the reviewer may concern. Indeed, coping with **multiple randomness sources** and **decaying step-size setting** (no longer a homogeneous markov chain) make Zou et al. (2021) intractable and largely different from our work. We take the estimation for the variance in Zou et al. (2021) as an example (where $H = \mathbb{E}[xx^{\top}]$):
>
> $$
> \text{Variance} \lesssim \sum_{t=0}^{n-1} \Big\langle  I - (I-\gamma H)^{N-t}, I - (I - \gamma H)^t  \Big\rangle ~~~~~[\text{Eq. (4.10) in Zou~et al. (2021)}]$$
> - In our setting, with multiple randomness sources, the covariance operator/matrix $\widetilde{\Sigma}_m$ is demonstrated to has only two distinct eigenvalues at $\mathcal{O}(1)$ and $\mathcal{O}(1/m)$ order in Lemma 1. In this case, we don't need to introduce the effective dimension to tackle $I - (I-\gamma H)^{N-t}$.
>
> This is used for obtaining a sharp convergence rate and accordingly we do not need to make any spectral decay assumption in Zou et al.(2021).
>
> - We use operators in Hilbert spaces from an approximation theory perspective to analyse the statistical properties of covariance operators. This technique does not exist in their work. Decomposition based on various randomness soruces leads to estimation on various terms, e.g., B1, B2, B3, V1, V2, V3, which makes our proof technique different from them. For example,
>
> Apart from multiple randomness sources, **decaying step-size setting** also makes our analysis largely different from their work as they exploit the property of constant step-size as a homogeneous markov chain. This is naturally invalid in our setting. For example, we cannot use $(I-\gamma H)^{N-t}$ under an adaptive step-size $\gamma$ in the above equation.
> Instead, we introduce integral estimation techniques to tackle adaptive step-size, see Appendix E for details.
>
> We  may kindly ask the reviewer to reconsider this point, which affects the evaluation of our work and score.

---

> > ### Author Response · Authors · 2022-08-01
> > **Response to Reviewer zi7u (continued)**
> >
> > **Q4:** The main theorems still have terms like $\mathbb{E}[1−\gamma_0r′ \mathrm{tr}(\Sigma_m)]^2$ and $\mathbb{E}[1−\gamma_0r′ \mathrm{tr}(\Sigma_m)]^4$ which require further analysis.
> >
> > **A4:** This is still in O(1) order, and we leave it here just to show how to obtain the condition on the step-size. According to your suggestions, we have already moved it in the proof, see line 909 for details.
> > Here we take $\mathbb{E}[1−\gamma_0r′ \mathrm{tr}(\Sigma_m)]^4$ as an example to show it’s in an $\mathcal{O}(1)$ order. By Jensen inequality, $\mathbb{E}(X^4) \geq (\mathbb{E} X)^4$ for a r.v. $X$, we have
> >
> > $$
> > \frac{1}{\mathbb{E}[1−\gamma_0r′ \mathrm{tr}(\Sigma_m)]^4} \leq \frac{1}{[1−\gamma_0r′ \mathbb{E}\mathrm{tr}(\Sigma_m)]^4} = \frac{1}{[1−\gamma_0r′ \mathrm{tr}(\widetilde{\Sigma}_m)]^4}  \sim \mathcal{O}(1)
> > $$
> >
> > by virtue of $\mathrm{tr}(\Sigma_m)$ with an $\mathcal{O}(1)$ sub-exponential norm.
> >
> >
> > **Q5:** The key finding of Zou et al. (2021) is that an effective dimension is established. How is the results of this paper related to the effective dimension?
> >
> > **A5:** In our work, we do not require the effective dimension as we prove that the covariance operator $\widetilde{\Sigma}_m$ is demonstrated to has only two distinct eigenvalues at $\mathcal{O}(1)$ and $\mathcal{O}(1/m)$ order in Lemma 1 (as our Contribution 1 in the introduction). Such fast eigenvalue decay does not need to separate the entire space into a “head” subspace where the error decays more quickly than the complement “tail” subspace, e.g., to bound $I - (I-\gamma H)^{N-t}$ in Variance ( see Eq. (4.10) in Zou et al. (2021) in the above).
> > In fact, in the double descent of random features setting, the minimal eigenvalue of covariance matrix/operator is more important, see the discussion in [S2, Section 2].
> >
> > [S2] K. Ilja, C. Szepesvári, O. Rivasplata, A. Rannen-Triki, and R. Pascanu. "On the role of optimization in double descent: A least squares study. NeruIPS2021.
> >
> >
> > We hope that our response has addressed the concerns of the reviewer on our paper; we would be happy to elaborate further if something remains unclear.

---

> > > ### Comment · Reviewer_zi7u · 2022-08-08
> > > **Thanks for your response**
> > >
> > > Dear authors,
> > >
> > > Thanks for your response, which has addressed most of my remaining questions. I am also happy to see the presentation of the paper improved.

---

> > > > ### Author Response · Authors · 2022-08-08
> > > > **thanks for your feedback and positive support**
> > > >
> > > > Dear Reviewer zi7u,
> > > >
> > > > We are thankful to your feedback and positive support on our work.
> > > >
> > > > We will polish this paper according to your constructive suggestions in the final version.
> > > >
> > > > Best regards,
> > > >
> > > > Authors

---

> > ### Author Response · Authors · 2022-08-07
> > **further feedback based on our rebuttal**
> >
> > Dear Reviewer zi7u,
> >
> > We are thankful for your constructive feedback.
> >
> > Based on your comments, we have already provided clarifications and discussion on **our derived results** and **the technical tools**.
> > We are currently unaware of the extent to which our responses have clarified your concerns, and would greatly appreciate it if you could provide your feedback and evaluation on our responses.
> >
> > We are willing to provide further clarifications to the reviewers if that helps.
> >
> > Best regards,
> >
> > Authors

---

### Official Review · Reviewer_NMXc · 2022-07-11

**Rating:** 6
**Confidence:** 3
**Soundness:** 3 good
**Presentation:** 3 good
**Contribution:** 3 good

**Summary:**

This paper studies the generalization properties of random features (RF) regression in high dimensions using SGD and derived non-asymptotic error bounds of RF regression under both constant and polynomial-decay step-size SGD settings. The analysis can be used to theoretically study the double descent phenomenon and does not restricted to Gaussian/spherical data assumption.
In addition, the authors also showed that the constant step-size SGD setting incurs no loss in convergence rate when compared to the exact minimal-norm interpolator, as a theoretical justification for using SGD in practice.


**Questions:**

Currently, almost all experiments have been conducted on the simple MNIST dataset or synthetic dataset. What prevents extending experiments to more difficult datasets such as CIFAR-10 or ImageNet?


**Limitations:**

Overall, this is an interesting paper trying to explain the double descent phenomenon theoretically. I feel the numerical validation is a little bit weak, considering it was conducted on MNIST. However, this looks like a standard-setting that has been used in many previous papers.


**Strengths And Weaknesses:**

This paper studies a very important phenomenon: the double descent for the over-parameterized model. They provided both theoretical analysis as well as empirical validation to show that the error bounds for variance and bias can be unimodal and monotonically decreasing, respectively, which is able to recover the double descent phenomenon. The presented analysis requires a weaker assumption on data distribution and holds for more general data distribution under the SGD setting. It provides a useful tool to understand the highly over-parameterized deep neural networks in general.

---

> ### Author Response · Authors · 2022-08-01
> **Response to Reviewer NMXc**
>
> We thank the reviewer’s efforts and the positive support.
>
> Q1: Currently, almost all experiments have been conducted on the simple MNIST dataset or synthetic dataset. What prevents extending experiments to more difficult datasets such as CIFAR-10 or ImageNet? I feel the numerical validation is a little bit weak, considering it was conducted on MNIST. However, this looks like a standard-setting that has been used in many previous papers.
>
> A1: We focused more on the theoretical aspects of the double descent for which we thought MNIST demonstration was sufficient. However, according to your suggestions, we have conducted our experiments on ImageNet16 (feature dimension is $16 \times 16 \times 3$, a downsampled version of ImageNet due to computational efficiency) in two classes, see Appendix G for details, highlighted in red for convenience.
>
> Our experimental results (also displayed on https://imgur.com/sIHpSxO in an anonymous link) also demonstrate the existence of the double descent phenomenon, and SGD requires more epoch to converge in the over-parameterized regime.

---

### Official Review · Reviewer_2MfL · 2022-07-13

**Rating:** 4
**Confidence:** 2
**Soundness:** 3 good
**Presentation:** 2 fair
**Contribution:** 2 fair

**Summary:**

This paper studies the variance and bias induced by using SGD to fit a linear model on random features from a theoretical point of view. They allow the data distribution to be non-Gaussian, but require bounded 4th moments. The main contributions appear to be some proof techniques for the analysis of SGD.

**Questions:**

There are other issues with the writing, that hide the potential significance of the work and make the paper difficult to read. For example, many technical lemmas are given before the main result, but it is unclear to the reader at the time why these may be important. The paper does not adequately help the reader digest the results and their significance. What is the significance of these results, and what is the purpose of having them before the main results in the paper?

**Limitations:**

N/A, the paper is theoretical in nature and does not have any significant direct social impact.

**Strengths And Weaknesses:**

## Strengths

The paper appears to provide some new results on SGD for models linear in the parameters, generalizing recent work [36], although I am not an expert on these proof techniques to critically assess their originality.

## Weaknesses

The paper is not particularly clear, sweeping a lot of details under the rug in the main paper. This is unfortunate, as the devil is in the details for this topic. For example, while most papers on random feature models study the bias and variance of the model itself, usually assuming that training finds the empirical loss minimizer, this paper focuses on the bias and variance of the SGD procedure itself. For example, they consider the approximation error with respect to the best model _conditional on the RKHS selected by the random features_, as opposed to considering the RKHS that the random features are approximating.

This makes it more comparable to other papers on SGD, which there are many, but which I am not appropriately qualified to assess the significance of this work in the context of. It's unclear what the significance of random feature model is for the paper. Additionally, while the paper specifically mentions that the results are similar in nature to those in [36], they are able to generalize to the stochastic setting with changing learning rate. To me, this suggests that the main contributions are in the proof techniques. However, these are all deferred to the appendix.

Updates following author discussion: The writing has improved slightly, but the overall point of the paper still does not come through, and is mostly buried among technical points of unclear significance. I am increasing my rating slightly, if the other reviewers feel strongly about the paper, but overall, think that it is a bit lacking in clarity of writing or significance.

---

> ### Author Response · Authors · 2022-08-01
> **Response to Reviewer 2MfL**
>
> We thank the reviewer's effort and feedback on this work despite not researching this topic directly.
>
> **Q1:** The paper is not particularly clear, sweeping a lot of details under the rug in the main paper.
>
> **A1:** As the reviewer could imagine, we are a bit conflicted on the quality of writing remarks as other reviewers mention that the paper is **clear and is well-written** and **significant**.  In particular, we adopt a standard technical style of writing, which introduces technical results in terms of lemmas before providing the main result, as they are used to build the main result.
>
> It appears the reviewer prefers the reverse way, providing a rough/informal main result, then building the technical lemmas and then providing the technical main result itself. While this can also be attractive, the space limitations make it quite a bit difficult. I hope the reviewer will excuse us and allow us to adopt the classical style and reconsider our score.
>
> ---
>
> Here we clarify some key points you concern by the following folds:
>
> **Q1-1:** most papers on random feature models study the bias and variance of the model itself, this paper focuses on the bias and variance of the SGD procedure itself.
>
> **A1-1:** In fact, the bias-variance decomposition in all of these works depends on the structure of the obtained solution. The only difference is, previous works (random features regression with the squared loss)  directly employ the closed-form solution for analysis. In our setting, the obtained solution is yielded by the SGD procedure without the closed-form structure. In this case, it is natural to conduct bias-variance decomposition of SGD, which is quite common and standard in learning theory, e.g., [S1,S2,S3]
>
> ---
>
> **Q1-2:** they consider the approximation error with respect to the best model  *conditional on the RKHS selected by the random features* , as opposed to considering the RKHS that the random features are approximating.
>
> **A1-2:** The hypothesis space (RKHS) in our work is spanned by random features (i.e., hidden neurons), see line 117 in Eq. (2), which is common and standard in the double descent community, e.g., see Eq. (1.1) in [S4] and page 2 in [S5].
>
> ---
>
> **Q1-3:** It's unclear what the significance of random feature model is for the paper.
>
> **A1-3:** Analysing neural networks in a simple version, e.g., random features (or even linear regression), NTK, is the first step to understanding the mystery of neural networks in terms of double descent/benign overfitting, see recent refs [S4,S6,S7,S8,S9].
>
>
> ---
> **Refs**
>
> [S1] A. Dieuleveut and F. Bach. Nonparametric stochastic approximation with large
> step-sizes. *Annals of Statistics*, 2016.
>
> [S2] P. Jain, S. Kakade, R. Kidambi, P. Netrapalli, and A. Sidford. Parallelizing stochastic gradient descent for least squares regression: mini-batching, averaging, and model misspecification. JMLR2018.
>
> [S3] A. Varre; N. Flammarion. Accelerated SGD for Non-Strongly-Convex Least Squares. COLT2022.
>
> [S4] S. Mei and A. Montanari.The generalization error of random features regression: Precise asymptotics and double descent curve. *Communications on Pure and Applied Mathematics*,  2022.
>
> [S5] B. Ghorbani, S. Mei, Theodor Misiakiewicz, and Andrea Montanari.Linearized two-layers neural networks in high dimension. *Annals of Statistics*, 2021.
>
> [S6] P. Bartlett, P. Long, G. Lugosi, and A. Tsigler. Benign overfitting in linear regression. PNAS2020.
>
> [S7] Z. Liao, R. Couillet, and M. Mahoney. A random matrix analysis of random fourier features: beyond the gaussian kernel, a precise phase transition, and the corresponding double descent. NerIPS2020.
>
> [S8] K. Ilja, C. Szepesvári, O. Rivasplata, A. Rannen-Triki, and R. Pascanu. "On the role of optimization in double descent: A least squares study. NeruIPS2021.
>
> [S9] G. Federica, B. Loureiro, F. Krzakala, M. Mézard, and L. Zdeborová. Generalisation error in learning with random features and the hidden manifold model. ICML2020.

---

> > ### Author Response · Authors · 2022-08-01
> > **Response to Reviewer 2MfL (continued)**
> >
> > **Q1-4:** the results are similar in nature to those in [36], they are able to generalize to the stochastic setting with changing learning rate. To me, this suggests that the main contributions are in the proof techniques. However, these are all deferred to the appendix.
> >
> > **A1-4:** The technical contributions have already been discussed in Section 4 of the manuscript in both plain language and in technical terminologies. Apart from this, we also give some theoretical findings/insights that cannot be easily obtained with the technical approach proposed in [36]. For example,
> >
> > - we show that $\widetilde{\Sigma}_m$ admits a very fast eigenvalue decay (use for sharp convergence rate), i.e., has only two distinct eigenvalues at $\mathcal{O}(1)$ and $\mathcal{O}(1/m)$ order, instead of making an assumption of data matrix in [36]. See contribution 1 in the introduction.
> > - we show that our result is able to capture the double descent behavior. This phenomenon cannot be obtained from their theoretical results.
> > - We also show that the constant step-size SGD setting incurs no loss in convergence rate when compared to the exact closed-form solution. Note that in practice, depending on the size of the problem, we might have to use SGD as opposed to obtaining the closed form solution due to memory issues.
> >
> > **Q2:** many technical lemmas are given before the main result, but it is unclear to the reader at the time why these may be important. What is the significance of these results, and what is the purpose of having them before the main results in the paper?
> >
> > **A2:** These key lemmas on statistical properties of covariance operators are helpful to understand how and why to obtain the main results. For example, in Lemma 1, we demonstrate that $\widetilde{\Sigma}_m$ admits a very fast eigenvalue decay, i.e., has only two distinct eigenvalues at $\mathcal{O}(1)$ and $\mathcal{O}(1/m)$ order.
> >
> > This is used for obtaining a sharp convergence rate and the reader can naturally understand why we do not need the data decay assumption in [36] for proof.

---

> > > ### Author Response · Authors · 2022-08-07
> > > **further feedback based on our rebuttal**
> > >
> > > Dear Reviewer 2MfL,
> > >
> > > Thanks for your time and effort on reviewing our work.
> > >
> > > Based on your comments, we have provided clarifications on problem setting and discussion on the writing style in the rebuttal.
> > > We  would greatly appreciate it if you could provide your feedback on our responses and re-think your final evaluations.
> > >
> > > Best regards,
> > >
> > > Authors

---

> > > > ### Comment · Reviewer_2MfL · 2022-08-08
> > > > **Missed main point**
> > > >
> > > > The authors seem to have missed the main points of the critique.
> > > >
> > > > Regarding clarity and writing: While it is common in theoretical papers to state lemmas before the proof of the main theory, this is usually done to highlight and discuss key technical ideas that enable the proof of the main theorem. In the case of this paper, my concern is that the results are provided with little to no discussion, and as the reader, I am left wondering what the main point of the paper is. Given that space is so limited, it must all be used to help the reader understand and believe the key ideas. Any theoretical lemmas that aren't important insights or interesting new proof ideas should be deferred to the supplementary materials to make room for much needed discussion of the theoretical results that do remain in the main text.
> > > >
> > > > Regarding the random feature model, I do not dispute the significance of the random feature model in the NTK, and its use as a potential approximation to neural networks. What I question is the significance to this paper. It seems like the main results about the statistical properties of SGD conditioned on the random mapping from inputs to features could just as easily be presented under a fixed featurization model. Therefore, the paper seems to be more about SGD than about random features. It is not clear what the delta is between this setting and the fixed featurization setting, and how that changes the result and/or proof technique.

---

> > > > > ### Author Response · Authors · 2022-08-09
> > > > > **Response to Reviewer 2MfL on the presentation style and difference with the least squares setting**
> > > > >
> > > > > Dear Reviewer 2MfL,
> > > > >
> > > > > Thanks for your effort and feedback on our response.
> > > > >
> > > > > According to your suggestions, in the revised version, we have moved several lemmas to Appendix C and included more discussions on our key results for better understanding.
> > > > > Besides, we also clarified the difference with the fixed featurization setting, and mentioned it in the revised version.
> > > > >
> > > > > ---
> > > > >
> > > > > **Q1:** the results are provided with little to no discussion
> > > > >
> > > > > **A1:** According to your suggestions, in the revised version, we have moved Lemmas 3 and 4 Appendix C and remained Lemmas 1 and 2 that are fundamental results in our work.
> > > > > Besides, more discussion on our main results and the used techniques are added in the revised version.
> > > > >
> > > > > - For **better understanding Lemmas 1 and 2**, we have added more discussion on motivation and influence. For example, before introducing Lemma 1, we briefly introduce the commonly-used effective dimension in least squares setting, which aims to separate the entire space into a “head” subspace where the error decays more quickly than the complement “tail” subspace.
> > > > > Then we explain our Lemma 1 to avoid this quantity with a clear motivation, see line 170 for details.
> > > > >
> > > > > - For **discussion on the obtained results**, we compared our obtained error bound with previous work: e.g., comparison to random features with closed form solution on bias, see line 207; comparison to least squares and random features on variance, see line 218. Our results can recover their results under some specific settings.
> > > > >
> > > > > - For **discussion on the used techniques** of our main results, we also took the estimation for the variance in Zou et al. (2021) as an intuitive example to show why the effective dimension is not needed and the estimation under the constant step-size setting is invalid, see line 275 for details.
> > > > >
> > > > > ---
> > > > >
> > > > > **Q2:** It is not clear what the delta is between this setting and the fixed featurization setting, and how that changes the result and/or proof technique.
> > > > >
> > > > > **A2:** We suspect that “the fixed featurization model” (the reviewer mentioned) indicates “linear regression" under our setting when compared to random features. Our response is based on these. We briefly discuss **the influence of introducing random feature mapping on the obtained result and proof techniques** when compared to least squares in the double descent community. We hope we have a good understanding of the reviewer’s question and would appreciate it if the reviewer could make this more clear.
> > > > >
> > > > > 1). **Effective dimension and covariance operator**: In the least squares setting, the effective dimension is often needed to obtain the double descent curve. Instead, in our random features mapping, we do not require the effective dimension as we prove that the covariance operator $\widetilde{\Sigma}_m$ is demonstrated to has only two distinct eigenvalues at $\mathcal{O}(1)$ and $\mathcal{O}(1/m)$ order in Lemma 1 (as our Contribution 1 in the introduction). Our analysis is based on the minimum eigenvalue of covariance matrix/operator instead of effective dimension.
> > > > >
> > > > > We take the estimation for the variance in Zou et al. (2021) as an example (where $H = \mathbb{E}[xx^{\top}]$):
> > > > >
> > > > > $$
> > > > > \text{Variance} \lesssim \sum_{t=0}^{n-1} \Big\langle  I - (I-\gamma H)^{n-t}, I - (I - \gamma H)^t  \Big\rangle ~~~~~[\text{Eq. (4.10) in Zou~et al. (2021)}]
> > > > > $$
> > > > >
> > > > > In their result, the effective dimension is introduced to separate the entire space into a “head” subspace where the error decays more quickly than the complement “tail” subspace, which aims to tackle $I - (I-\gamma H)^{n-t}$. Instead, in our setting the covariance operator with fast eigenvalue decay can avoid this separation. We have clarified this in the revised version, see line 275 for details.
> > > > >
> > > > >
> > > > > 2). Introducing random features leads to **multiple randomness sources**, which makes our analysis intractable. To this end,  we introduce a new error decomposition technique to (partly) disentangle the multiple randomness sources on the data $\mathbf{X}$, the random features matrix $\mathbf{W}$, the noise $\epsilon$. This also makes full use of statistical properties of covariance operators. Here we take the decomposition of Bias as an example:
> > > > >
> > > > > $$
> > > > > \text{Bias} \lesssim B1 + B2 + B3,
> > > > > $$
> > > > >
> > > > > where B3 is a deterministic quantity that is closely connected to model (intrinsic) bias without any randomness; while B1 and B2 evaluate the effect of randomness from $\mathbf{X}$ and $\mathbf{W}$ on the bias, respectively. This is different from the least squares setting.
> > > > >
> > > > > ---
> > > > >
> > > > > We hope that our response as well as the revised version would address the reviewer’s main concern on the presentation and difference with the least squares setting. We would be happy to elaborate more if the reviewer has any further questions.
> > > > >
> > > > > Thank you again for your effort on improving the readability of this paper.
> > > > >
> > > > > Best regards,
> > > > >
> > > > > Authors

---

### Official Review · Reviewer_XDPQ · 2022-07-18

**Rating:** 7
**Confidence:** 2
**Ethics Flag:** Yes
**Soundness:** 3 good
**Presentation:** 3 good
**Contribution:** 3 good

**Summary:**

This work gives a novel analysis of the excess risk of a random features regression which is solved by SGD. The result is relevant in the overparmeterized high dimensional setup. The analysis demonstrates the existence of the double decent phenomena in this setup and is more general than existence theory with regard to the data distribution. The theory is supported by experiments on real data.

**Questions:**

1) In line 120 $\Sigma_m$ is said to be an operator $\mathbb{R}^m \to \mathbb{R}^m$ but operates on $\mathcal{H}$ which is maybe isomorphic to $\mathbb{R}^m$ but it is not mentioned.


2) It would be helpful to define ⊗ in the notations section.

3) Even though the work deals with finite dimensional regression is uses a functional terminology. Is that really helpful for the representation of the results? why wouldn't we describe everything in the parameters space, for example, use covariance of features terminology instead of covariance operator. isn't it over-sophistication?

**Limitations:**

There is a quite significant discussion on neural networks in the introduction. However, as the random features method consists only a single trainable layer it is essentially linear and the theoretical result has a small contribution to the understanding of the excess risk in neural networks.

**Strengths And Weaknesses:**

Originality: the work expands the exiting known theory regarding expected test error of an SGD learner by requiring weaker assumption on the data.
It also expands the exiting known theory  regarding the expected test error of random feature regression to non closed form solution e.g. SGD optimization.

Quality: the work is technically sound in my views. I did not check the proofs, however.

Clarity: the paper is clear and well written.

Significance: The result helps to understand the relation between the numerical SGD solution and the closed form minimal norm solution of the regression problem and is significant.

---

> ### Author Response · Authors · 2022-08-01
> **Response to Reviewer XDPQ**
>
> **Q1:** $\Sigma_m$ is said to be an operator $\mathbb{R}^m \rightarrow \mathbb{R^m}$ but operates on $\mathcal{H}$ may be isomorphic to $\mathbb{R}^m$ but is not mentioned.
>
> **A1:** The covariance operator $\Sigma_m$ is self-adjoint in the Hilbert space $\mathcal{H}$ that is isomorphic to $\mathbb{R}^m$. We have made this clear in our revised version, see line 119, highlighted in red for convenience.
>
>
> **Q2:** It would be helpful to define $\otimes$ in the notations section.
>
> **A2:** The $\otimes$ notation denotes the tensor product and can be found in the revised version, see line 71.
>
> **Q3:** Even though the work deals with finite dimensional regression is uses a functional terminology. Is that really helpful for the representation of the results? why wouldn't we describe everything in the parameters space, for example, use covariance of features terminology instead of covariance operator. isn't it over-sophistication?
>
> **A3:** The use of functional terminology allows us to focus on the empirical risk functional and the expected risk functional in the $ L_2(\mathrm{d} \rho) $ space. This is standard and commonly used in learning theory, e.g., [S1,S2] in the under-parameterized regime and in the interpolation or over-parameterized regime [S3]. Besides, using this terminology covers the infinite dimensional setting, i.e., our result is able to handle the asymptotic case $m \rightarrow \infty$.
>
> [S1] A. Dieuleveut and F. Bach. Nonparametric stochastic approximation with large
> step-sizes. *Annals of Statistics*, 2016.
> [S2] Carratino, Luigi, Alessandro Rudi, and Lorenzo Rosasco. "Learning with SGD and random features." NeurIPS2018.
> [S3] Liang, Tengyuan, and Alexander Rakhlin. Just interpolate: Kernel “ridgeless” regression can generalize. *The Annals of Statistics* 2020.
>
> **Limitations:** There is a quite significant discussion on neural networks in the introduction. However, as the random features method consists of only a single trainable layer it is essentially linear and the theoretical result has a small contribution to the understanding of the excess risk in neural networks.
>
> **A4:** We agree with the reviewer that there appears to be a gap in analysis between random features and neural neural networks. Nevertheless, analysing neural networks in a simple version, e.g., random features, NTK, is the first step to understanding the mystery of neural networks in terms of double descent/benign overfitting. According to your suggestions, we revised the introduction, see line 16-18.

---

### Author Response · Authors · 2022-08-08
**discussion period**

Dear Reviewers,

We are thankful for the reviewers' time and effort on our work with thoughtful and constructive comments and appreciate Reviewer zi7u's feedback on our rebuttal.

Since the author-reviewer discussion period is coming to end, we are currently unaware of the extent to which our responses have clarified the reviewers' concerns, and would greatly appreciate it if you could provide your feedback on our responses.

We are willing to provide further clarifications to the reviewers, including revisions, if it helps.

Best regards,

Authors

---

### Author Response · Authors · 2022-08-09
**General response on paper revision during the author-reviewer discussion period**

Dear Reviewers,

We thank all the reviewers' effort and constructive feedback on this work during the author-reviewer discussion period.

According to suggestions from all the reviewers, we have revised this paper for better presentation and added more discussions on the obtained result and techniques.

- We have moved some examples and lemmas that are not significantly important to the appendix, and clarified the motivation behind our lemmas in the main text for **better presentation**.
- We have added more discussions on the difference from least squares in the over-parameterized regime with regard to the derived error bounds and the used techniques for **better clarifications**.

We hope that our response and the revised version would further bolster the Reviewers XDPQ, NMXc, zi7u positive opinion of the work as well as address Reviewer 2MfL’s main concern.

Best regards,

Authors

---

### Meta-Review · Area_Chair_SXmJ · 2022-08-26

**Recommendation:** Accept
**Confidence:** Less certain

**Metareview:**

While the reviewers showed some disagreement, the majority of them considered the paper novel and interesting. Moreover, during the discussion period the authors improved the clarify of the presentation and of the proved results. Overall, the paper makes a good contribution to the conference.

**Award:**

No

---

### Decision · Program_Chairs · 2022-09-14

Accept